# AtCHE1, the Arabidopsis homolog of mammalian AATF/Che-1 protein, is involved in safeguarding genome stability
Fang Liu [1,2,11] ✉, Bingshan Wang[2,11], Xiangyang Wang[1,3], Daofeng Dong[4], Lieven De Veylder [5,6], Shengdong Qi[1], Beatrix M. Horvath[7,8], Klaus Palme [2,9,10] ✉ & Xugang Li[1,9,12]

Both endogenous and exogenous genotoxins can inflict damage on cellular DNA, leading to reduced genomic stability in plants, which adversely affects development. Apoptosis Antagonizing Transcription Factor (AATF), also referred to as Che-1, has been identified as a binding protein for RNA polymerase II. It plays a crucial role in various cellular processes, including cell proliferation, transcriptional regulation, apoptosis, DNA damage response, and ribosome biogenesis in mammals. In this study, we identify the *che1* mutant derived from an ethyl methanesulfonate (EMS)-mutagenized Arabidopsis Col-0 population, characterized by a short root and small leaf phenotype. The underlying mutation is a G-to-A transition located at the boundary of the eighth intron and ninth exon of the *AT5G61330* gene, resulting in a misprocessed mRNA transcript. AtCHE1, a homolog of the mammalian AATF/Che-1, contains both the conserved AATF/Che-1 and TRAUB domains in Arabidopsis. Under standard conditions, the *che1* mutant exhibits an accumulation of damaged DNA, cell death, and differentiation defects at the root tip. Collectively, these findings underscore the importance of AtChe-1 in meristem maintenance and genome stability.

In higher plants, new organs are generated from stem cells located in the root and shoot meristems. Within the root apical meristem (RAM), stem cells surround the rarely dividing quiescent center (QC) cells, which express the *WUSCHEL-RELATED HOMEOBOX5* (*WOX5*) transcription factor gene[1,2]. Stem cell niche maintenance is regulated by several transcription factors, including the AP2/PLT (APETALA2/PLETHORA) transcription factors, providing an apical-basal patterning signal. They are regulated by auxin, and form together a gradient of *PLT* expression[3]. On the other hand, the GRAS (derived from GAI: gibberellic acid insensitive, RGA: repressor of GA1-3 mutant, and SCR: scarecrow), SHORTROOT (SHR), and SCARE-CROW (SCR) transcription factors provide a radial patterning signal[4,5]. Besides, meristem maintenance requires genome integrity. Members of the MAIN (MAINTENANCE OF MERISTEMS) and MAIN-like family, the three homologs: MAIN-LIKE1 (MAIL1, AT2G25010), MAIN-LIKE2 (MAIL2, AT2G04865), and MAIN-LIKE3 (MAIL3, AT1G48120) are involved in maintaining genome stability. Their mutants show short primary roots, disorganized RAM, and accumulated DNA breaks[6,7]. AtMMS21 (Methyl Methanesulfonate Sensitivity gene 21), a subunit of the STRUC-TURAL MAINTENANCE OF CHROMOSOME5/6 complex, is involved in ameliorating DNA double-strand breaks (DSBs) and maintaining the stem cell niche during root development[8]. The *m56-1fas2-4* triple mutant lacking the H3/H4 histone chaperone CHROMATIN ASSEMBLY FACTOR-1 (CAF-1) and the H2A/H2B histone chaperone NAP1-RELATED PROTEIN1/2 (NRP1/2), which function synergistically in chromatin maintenance and genome maintenance, exhibit programmed cell death and severe short-root phenotype[9].

Plants, as any other living organisms, suffer from various endogenous and exogenous stresses, such as ultraviolet radiation, chemical mutagens,

[1]State Key Laboratory of Crop Biology, College of Life Sciences, Shandong Agricultural University, Tai'an, China. [2]Institute of Biology II, Faculty of Biology, Albert-Ludwigs-University of Freiburg, Schänzlestrasse 1, Freiburg, Germany. [3]Key Laboratory of Herbage & Endemic Crop Biology of Ministry of Education, Inner Mongolia Key Laboratory of Herbage & Endemic Crop Biotechnology, School of Life Sciences, Inner Mongolia University, Hohhot, China. [4]Shandong Key Laboratory of Greenhouse Vegetable Biology, Shandong Branch of National Vegetable Improvement Center, Institute of Vegetables, Shandong Academy of Agricultural Sciences, Jinan, China. [5]Department of Plant Biotechnology and Bioinformatics, Ghent University, Gent, Belgium. [6]Center for Plant Systems Biology, VIB, Gent, Belgium. [7]School of Biological Science, Centre for Systems and Synthetic Biology, Royal Holloway, University of London, Egham Hill, Egham, UK. [8]Molecular Life Sciences, Wageningen University, Stippenweg 2, Wageningen, The Netherlands. [9]Sino-German Joint Research Center on Agricultural Biology, College of Life Sciences, Shandong Agricultural University, Tai'an, China. [10]ScreenSYS GmbH, Biotechpark, Engesserstr. 4a, Freiburg, Germany. [11]These authors contributed equally: Fang Liu, Bingshan Wang. [12]Deceased. ✉e-mail: 903656896@qq.com; klaus.palme@biologie.uni-freiburg.de

and reactive oxygen species, which can damage the integrity of their genomes[10,11]. To prevent the transmission of damaged DNA to daughter cells, the ATM/RAD53-RELATED (ATR) and ATAXIA-TELANGIECTASIA MUTATED (ATM) kinases are activated in response to replication defects and DSBs, respectively[12,13], ultimately leading to DNA damage repair, programmed cell death, or endoreduplication. Unlike animals, there is no orthologue of Chk1, Chk2, or CDC25, which act downstream of ATM or ATR. In plants, the p53 functional homolog SUPPRESSOR OF GAMMA RESPONSE 1 (SOG1) and the cell cycle regulatory kinase, WEE1 play a central role in signaling. In response to DNA damage, SOG1 is phosphorylated by ATM[14–16]. The activated SOG1 transcriptionally regulates hundreds of DNA damage response genes involved in cell cycle regulation, endocycle, cell death, and DNA repair. While the WEE1 controls cell cycle progression via the WEE1-RPL1-CYCDs and the WEE1-FBL17/SKP2-CKI-CDKs axis in response to replication stress[17,18]. Furthermore, as recent studies have shown, RETINOBLASTOMA RELATED (RBR), the Arabidopsis retinoblastoma (Rb) homolog, in addition to its well-known cell cycle control and stem cell maintenance functions, also maintains genome integrity of root meristematic cells. It mediates the localization of the repair protein RAD51 to DNA lesions and accumulates together with E2FA and AtBRCA1 at the damaged, γH2AX-labeled foci in Arabidopsis[19,20]. Besides cell death, DSBs also induce endoreplication through a series of cell cycle regulators controlled by the ATM-SOG1 and ATR-SOG1 pathways[21].

The human AATF/Che-1, identified as a subunit of RNA polymerase II, interacts with Rb to repress its function[22]. In response to DNA damage, Che-1 is stabilized by phosphorylation by ATM/ATR and Chk2[23]. Phosphorylated Che-1 activates p53 and p21 transcription to maintain the G2/M checkpoint[23]. As a cofactor, phosphorylated Che-1 binds to p53 and forms a ternary complex with Brca1 in the first hours of DNA damage, resulting in transcription of growth arrest genes. Consequently, when cells accumulate an excess of damaged DNA, p53 is released from the Che-1/p53/Brca1 complex to promote transcription of pro-apoptotic genes[24]. AATF/Che has a wide variety of functions. In general, AATF/Che-1 plays an important role in the regulation of proliferation and survival in the DNA damage response pathway in mammals. Besides this role, AATF/Che-1 is also a key component of ribosome biogenesis. It has been reported to interact with neuroguidin (NGDN) and NOL10 to form a complex called the AATF-NGDN-NOL10 (ANN) complex. And this nucleolar complex is involved in the synthesis of 40S ribosomal subunits[25]. AATF/Che-1 also interacts with several RNA species and proteins such as 45S pre-rRNA, snoRNAs, ribosome biogenesis mRNAs, rRNA processing proteins, and ribosomal proteins[26]. In addition, Che-1 affects rDNA transcription by binding to the RNA polymerase I machinery[27].

Although AATF/Che-1 has been well studied in animals, little is known about its plant homolog. Here, we describe the isolation of the mutant, named che1, which exhibits severe aberrations in root architecture, root growth inhibition and cell death in the root meristem. Based on the phenotypic changes, we propose that the nuclear-localized AtChe1, the Arabidopsis AATF/Che-1 homolog, is involved in root development and genome stability. Our analysis of sog1-1;che1 double mutant shows that AtCHE1 and SOG1 play a synergistic role in genome safeguarding.

## Results
### The che1 mutant exhibits a short primary root and a disorganized stem cell niche (SCN)

Isolated from an EMS-mutagenized Arabidopsis Col-0 population, the che1 mutant demonstrates retarded growth, characterized by shorter primary roots and smaller cotyledons compared to Col-0 (Fig. 1A). To rule out potential background mutations, we backcrossed the homozygous che1 mutant with Col-0. Analysis of the backcrossed lines revealed a 3:1 (221:68, Chi-squared test, $p = 0.56$), confirming that che1 is a recessive allele. At 4 days after germination (DAG), the root meristem length in the backcrossed che1 mutant was already reduced and remained significantly shorter than that in Col-0 at 8 DAG (Fig. 1B, C, and E). The number of transit amplifying

cells in the cortex layer was also significantly decreased ($P < 0.01$) in the che1 mutant at 4 DAG (Fig. 1D). While Col-0 showed a slight increase in cell numbers at 6 DAG, the che1 mutant continued to exhibit a decline (Fig. 1B, D), indicating a progressive retardation of the meristem, and that shortening of the mitotic zone is often observed in roots undergoing genomic stress[6]. These findings suggest that transit-amplifying cells in the che1 mutant differentiate at an earlier developmental stage.

To determine whether the observed phenotypic changes in root meristem development correlate with alterations in the stem cell niche, specifically the QC, we analyzed its structure (Fig. 1F) and monitored the expression of stem cell-specific markers (Fig. 1G, H). Propidium-iodide (PI) staining revealed disorganized QC architecture in the che1 mutant (Fig. 1F), which was further confirmed by altered expression pattern of the $WOX5_{pro}$:GFP (Fig. 1G).

We also examined the distal stem cell niche in the che1 mutant using the J2341 enhancer trap line, which indicates expression in the columella initials in Col-0. In the che1 mutant, J2341 expression expanded into additional cells surrounding the QC, unlike in Col-0 (Fig. 1H). To delve into the changes in the proximal meristem, we analyzed the expression of $CO2_{pro}$:H2B-YFP, $CO3_{pro}$:H2B-YFP, and $SCR_{pro}$:GFP, which are specifically expressed in cortical and endodermal layers[28]. In the che1 root meristem, all three markers exhibited abnormal localization and cell division patterns within the cortex and endodermis (Fig. 1I, J, and K). Collectively, these results indicate that AtCHE1 is essential for maintaining the stem cell niche and proper organization of the root meristem.

### AtCHE1 encodes the Arabidopsis homolog of the mammalian AATF/Che-1

To identify the AtCHE1 gene, we conducted map-based cloning. Preliminary mapping indicated that the mutated site is located at the tail of chromosome 5, between markers MTE17 and MSN2[29].

Using simple sequence length polymorphism (SSLP) markers and derived cleaved amplified polymorphic sequence (CAPS) markers, we fine-mapped the mutation and sequenced all the genes in this region, comparing them to the wild-type control. The mutated base pair was identified in the gene annotated as AT5G61330, at the junction of its eighth intron and ninth exon, where a G-to-A transition led to RNA mis-splicing (Fig. 2A, Supplementary data 1). To assess the effects of this alternative splicing, we compared the sequences of mutant and wild-type cDNAs via reverse transcription-polymerase chain reaction (RT-PCR) using total RNA extracted from both mutant and wild-type seedlings. The mis-splicing event appears to result in premature translation termination 870 bp from the translation start site (Fig. 2A), likely producing a truncated protein of 290 amino acids that still retains the part of the domain (Fig. 2B).

To confirm that the mutation in the AtCHE1 gene was solely responsible for the growth defects observed in the che1 mutants, we transformed homozygous che1 plants with the full-length genomic region of AtCHE1, expressed under its own promoter (The sequence spanning 2197 bp between AtCHE1 and the adjacent gene AT5G61340, along with portions of both exon and the 3'UTR of AT5G61340. This design aims to encompass any potential distal control elements.) and including its 3'-UTR ($AtCHE1_{pro}$-$AtCHE1_g$-$AtCHE1_{3'-UTR}$). The wild-type AtCHE1 protein fully complemented the short primary root phenotype of the che1 mutant (Supplementary Fig. 1), confirming that the mutation in the AtCHE1 gene was indeed responsible for the observed phenotypic alterations.

According to the public database (TAIR), AtCHE1 is predicted to encode an rRNA processing protein, with its coding sequence showing 41.7% sequence similarity and 26.5% sequence identity with the human AATF/Che-1 protein, which encodes the Apoptosis Antagonizing Transcription Factor (AATF) (https://www.ebi.ac.uk/Tools/psa/emboss_needle/). AtCHE1 contains two conserved protein domains: the N-terminal leucine zipper region of AATF/Che-1 (from amino acids 142 to 271) and the C-terminal TRAUB domain (from amino acids 350 to

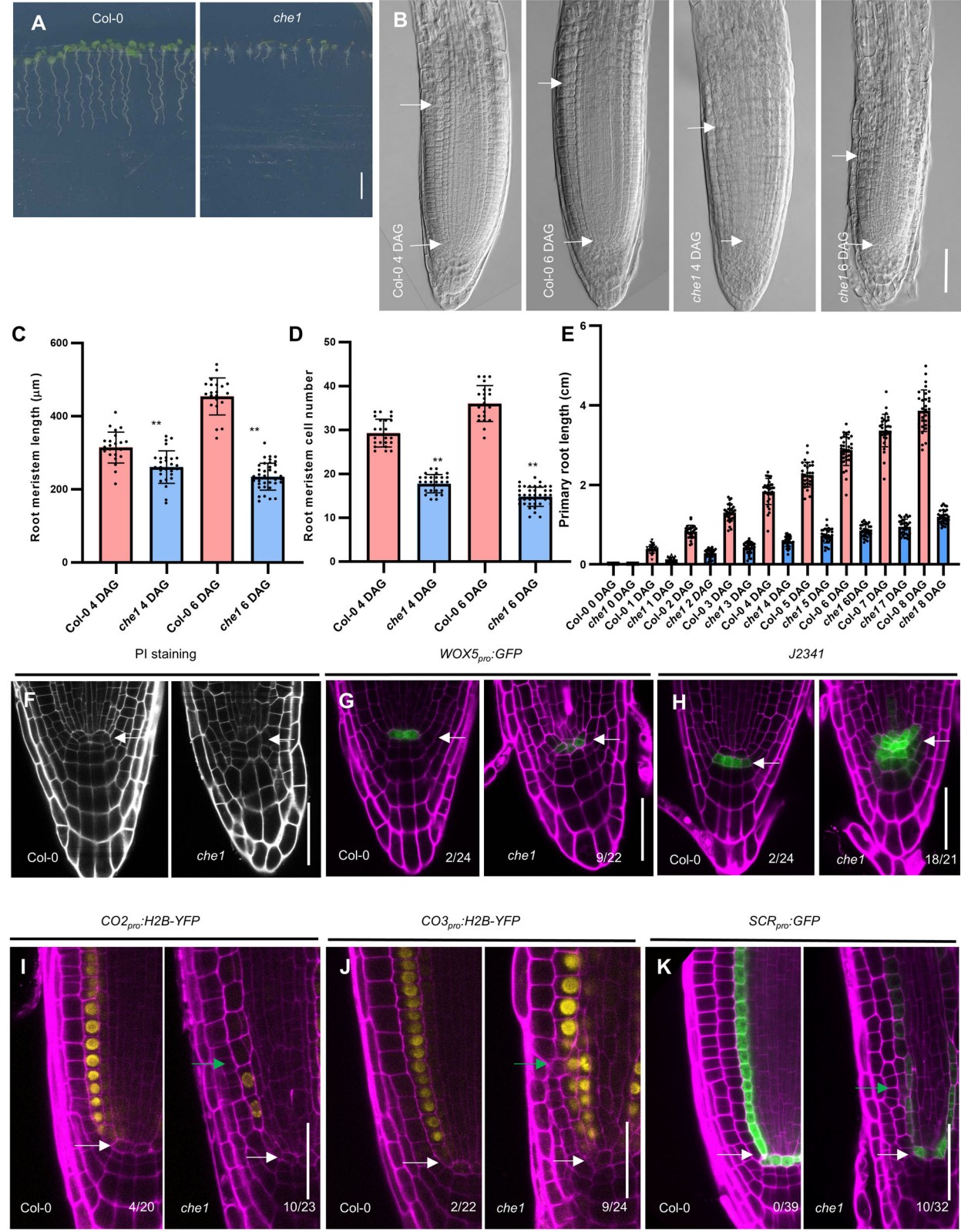

425) (Fig. 2B). Notably, the truncated AtCHE1 protein retains the AATF domain. Our comparative analysis indicates that AtCHE1 is the only AATF/Che-1 homolog identified in Arabidopsis that demonstrates evolutionary conservation between the animal and plant kingdoms (Supplementary Fig. 2 and Supplementary Fig. 3).

**AtCHE1 is ubiquitously expressed and localized in the nucleus**
To investigate *AtCHE1* expression in Arabidopsis, we generated *AtCHE1pro:GUS* reporter lines and analyzed three independent transgenic lines. We observed GUS activity in young roots, shoots, leaves, flowers, and siliques (Fig. 3A–G), with particularly high

**Fig. 1 | Primary root growth and SCN maintenance are inhibited in the *che1* mutant. A** Growth habit of the *che1* mutant compared to the control, Col-0 at 6 DAG under normal growth conditions. Scale bar = 1 cm. **B** Representative images of differential interference contrast (DIC) microscopy showing the root meristem of Col-0 and *che1* (4 and 6 DAG). Scale bar = 50 µm; arrowheads point to QC (White) and first elongated cortical cell position. **C** Meristem length was measured from the QC to the first elongating cortex cell and (**D**) the number of meristem cells in the cortex of Col-0 and *che1* at 4 and 6 DAG was counted. The values and error bars in (**C**) and (**D**) represent means and ±SD, n > 20. Double asterisks indicate highly significant differences (*P* < 0.01) between *che1* and Col-0 analyzed by two-tailed Student's *t* tests. **E** The graph shows the root growth of Col-0 and *che1* between 0-8 days after germination. Data represent the mean with ± SD; *n* = 30. Representative confocal images of propidium iodide (PI)-stained Col-0 and *che1* mutant root tips (**F**) PI only, (**G**) showing expression of *WOX5pro:GFP* while (**H**) of columella stem cell marker *J2341*. Scale bars = 50 µm, arrowheads point to QC. Col-0 and *che1* roots expressing *CO2pro:H2B-YFP* (**I**), *CO3pro:H2B-YFP* (**J**), and *SCRpro:GFP* (**K**). The ratios indicate the number of seedlings showing abnormal expression patterns compared to the total number of seedlings examined. White arrowheads indicate to QC and green arrowheads point to abnormal cell division. Scale bars = 50 µm. Magenta, PI staining; yellow, YFP and green, GFP.

**Fig. 2 | *AtCHE1* encodes AATF/CHE-1 in Arabidopsis. A** Sequence comparison of the wild type (*AtCHE1*) and mutant (*che1*) DNA, CDS, and predicted proteins. The red frame shows the last base of the eighth intron, in which G is mutated to A by EMS mutagenesis. The green frame and black frame show the missing sequence 'AACGTTAG' and the formation of a stop codon by mis-splicing. The green arrow indicates the last amino acid of the predicted *che1* mutant protein. The CDS sequences were obtained by reverse transcription-polymerase chain reaction (RT-PCR) using total RNA isolated from mutant and wild-type seedlings. **B** Comparison of the conserved protein boxes between the human AATF/Che-1 (hAATF) and AtCHE1. Light blue box: AATF/Che-1 domain, lila box: TRAUB, orange box: Nuclear localization signal (NLS), yellow box: Nucleolar localization sequence (NoLS). Red arrowhead: position of the last amino acid of *che1*. Blue letters and numbers indicate the potential phosphorylation sites on serine and threonine residues in hAATF and AtCHE1, respectively.

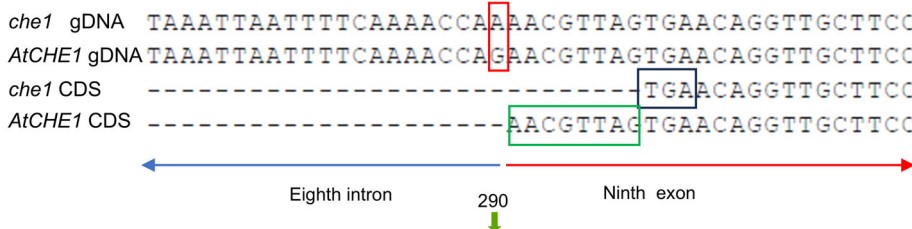

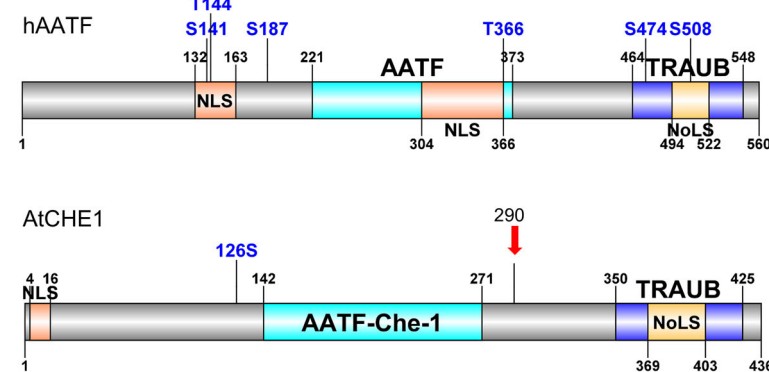

expression in the root apical meristem, aligning with the observed root phenotypes of the *che1* mutant.

To explore the cellular and subcellular localization of the AtCHE1 protein, we created translational reporter lines (*AtCHE1pro:AtCHE1g-GFP*) expressing the AtCHE1-GFP fusion protein under its own promoter. Consistent with the expression patterns of the *AtCHE1pro:GUS* lines, the AtCHE1-GFP fusion protein exhibited strong expression in stem cells and transit amplifying cells, with weaker expression in the QC, columella cells, and the elongation zone, where cell division is less active (Fig. 3H).

The AtCHE1-GFP fusion protein localizes to the nucleus (Fig. 3H). To further examine this localization, we introgressed the nucleoplasm marker *35S:H2B-tdTomato* into the *AtCHE1pro:AtCHE1g-GFP* transgenic line. The H2B labeling allowed for a more precise assessment of nuclear morphology, including the nucleolus and surrounding nucleoplasm[30]. Confocal images revealed that the peaks of tdTomato and GFP signals did not overlap (Fig. 3I, K). Additionally, the fluorescent dye 4',6-diamidino-2-phenylindole (DAPI) was used to visualize nuclear DNA, highlighting the nucleolus as a darker region due to its low DNA concentration[31]. DAPI staining in the *AtCHE1pro:AtCHE1g-GFP* transgenic line confirmed the nuclear presence of AtCHE1; however, the peak of the GFP signal was located differently than that of DAPI (Fig. 3J, L).

Collectively, these observations demonstrate that AtCHE1 is predominantly localized in the nucleus of dividing cells.

### Expression of the root master regulator genes is altered in the *che1* roots

Given the retarded root growth and the expression of AtCHE1 in the root meristem, we investigated whether these developmental changes were linked to alterations in the auxin/PLT pathway. To analyze auxin response in the *che1* mutant, we examined the expression of the *DR5rev:GFP* reporter. The auxin response maxima in the QC and the distribution in columella cells in the *che1* mutant were comparable to those in Col-0 (Fig. 4A).

We further assessed the transcriptional changes of root master regulators by crossing the *PLT1pro:CFP* and *PLT2pro:CFP* marker lines into the *che1* mutant background. Both PLT1 and PLT2 are essential for maintaining stem cell identity, promoting the mitotic activity of stem cell daughters, and facilitating cell differentiation[32]. In the Col-0 background, *PLT1pro:CFP* and *PLT2pro:CFP* displayed high expression in the stem cell niche, intermediate levels in rapidly dividing cells, and low levels in elongated cells. In contrast, the transcriptional expression of *PLT1* and *PLT2* in the *che1* mutant was reduced and restricted primarily to the stem cell niche (Fig. 4B, C; Supplementary Fig. 4A, B). Quantitative RT-PCR (qRT-PCR) analysis of *PLT1* and *PLT2* transcript levels in *che1* root tips confirmed these findings (Fig. 4F). Similarly, expression levels of the GRAS family genes *SCR* and *SHR* were also decreased in the *che1* mutant compared to Col-0 (Fig. 4D, E, and G; Supplementary Fig. 4C, D). Collectively, these results suggest that the aberrant root formation and stem cell niche maintenance in

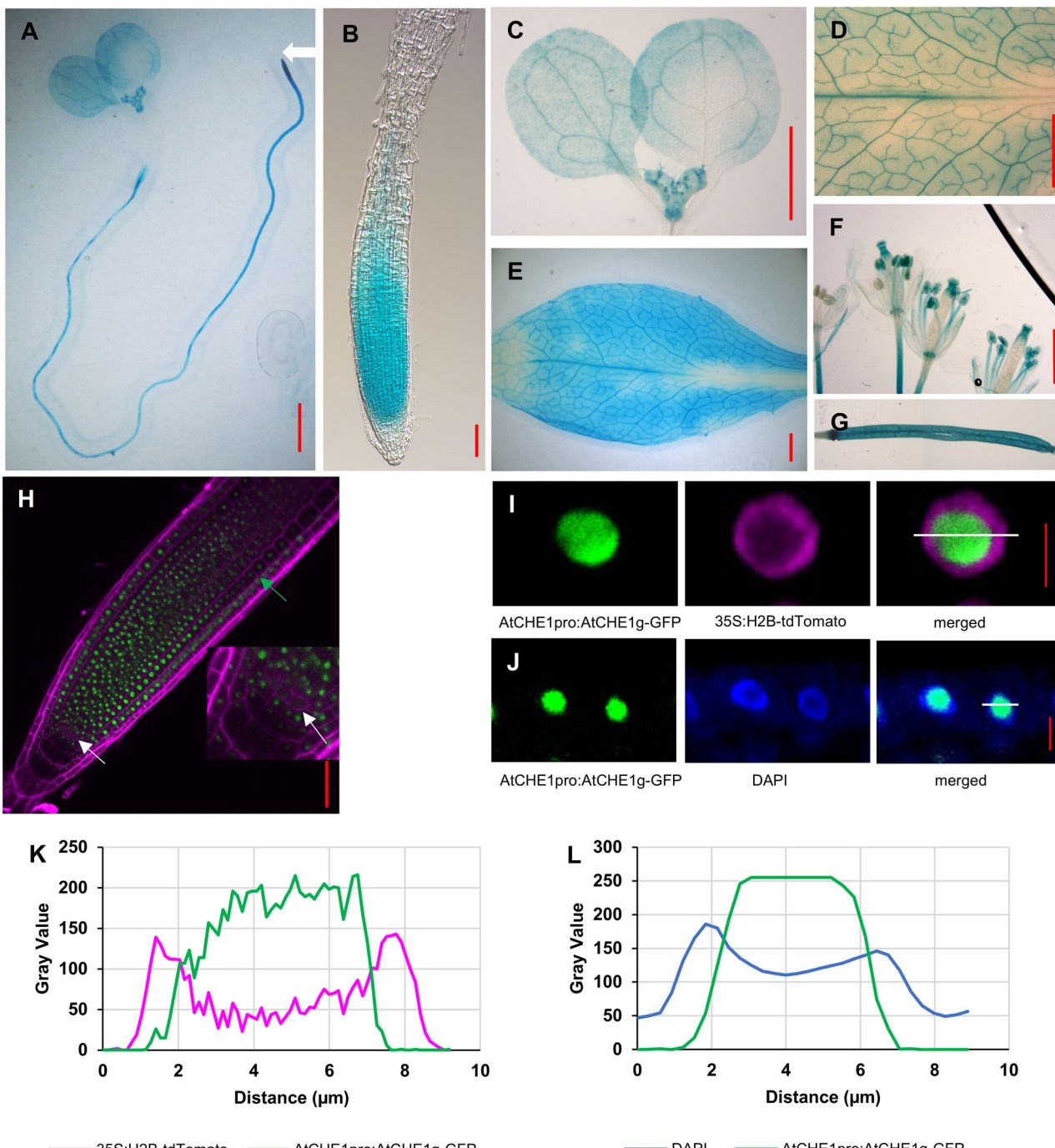

**Fig. 3 | AtCHE1 is widely expressed and localized in nucleus. A–G** represent GUS expressional studies in **A** young seedling, **B** root, **C** young shoot, **D** enlarged image of mature rosette leaf, **E** mature rosette leaf, **F** flowers and **G** silique. The white arrowhead in (**A**) points to the root apical meristem. **H** The representative confocal image illustrates the expression of AtCHE1 in the root meristem followed in the *AtCHE1_{pro}:AtCHE1_g-GFP* line (6 DAG). The enlarged section highlights its reduced level of expression in the QC compared to the surrounding stem cells. Magenta: PI staining; green, GFP fluorescence. Arrowheads (White) point to QC and the first elongated cortex cell (Green). Scale bars =1000 μm in (**A, C, D–G**), scale bars = 50 μm in (**B**) and (**H**). **I** Confocal images of the nuclear localization of the AtCHE1-GFP in the *AtCHE1_{pro}:AtCHE1_g-GFP* line transformed with the *35 s:H2B-tdTomato* marker. Magenta: H2B-tdTomato; green, AtCHE1-GFP. Scale bar = 5 μm. **J** DAPI-stained nuclei of the *AtCHE1_{pro}:AtCHE1_g-GFP* line. Blue: DAPI; green, GFP. Scale bar = 5 μm. **K** and **L** Gray value distribution of the total number of pixels at the position of the white lines marked in (**I**) and (**J**).

the *che1* mutants are associated with the disrupted distribution of PLT1, PLT2, SCR, and SHR.

### *che1* mutant shows cell death and DNA damage

Similar to the mutations in genes, such as *FAS1* (subunit of the counterpart of chromatin assembly factor-1)[33], *MMS21-1*[8] and *RBR*[19] involved in stem cell niche and meristem maintenance and critical for genome integrity, we also observed occasional cell death response in the *che1* mutant (42%, Fig. 5A control and Supplementary Fig. 5A).

To investigate whether the absence of AtCHE1 leads to genome instability, first we used the neutral comet assay, and afterwards we followed the accumulation of the phosphorylated histone variant, γH2AX to detect

**Fig. 4 | Auxin signaling and expression patterns of root developmental regulator genes are altered in *che1*. A** represents the expression of *DR5$_{rev}$:GFP*. **B** of *PLT1$_{pro}$:CFP* and **C** *PLT2$_{pro}$:CFP* in the root tips of 5 DAG Col-0 and *che1*. **D** Expression pattern of *SCR$_{pro}$:GFP* and **E** *SHR$_{pro}$:SHR-GFP* in the root tips of 5 DAG Col-0 and *che1*. **A–E** In each image, scale bar = 50 μm; arrowhead points to QC, Magenta, PI staining; Green, GFP, Cyan, CFP. **F** Relative mRNA levels of *PLT1* and *PLT2* quantified by qRT-PCR in Col-0 and *che1*. **G** *SCR* and *SHR* relative mRNA levels (gene vs reference gene) determined by qRT-PCR in Col-0 and *che1*. For the analysis, both in (**F** and **G**), root tips (2 mm long) were collected, values and error bars represented means and ±SD from three independent biological replicates. Asterisks indicate significant differences compared with Col-0. (***P* value < 0.01, **P* value < 0.05, two-tailed Student's t test).

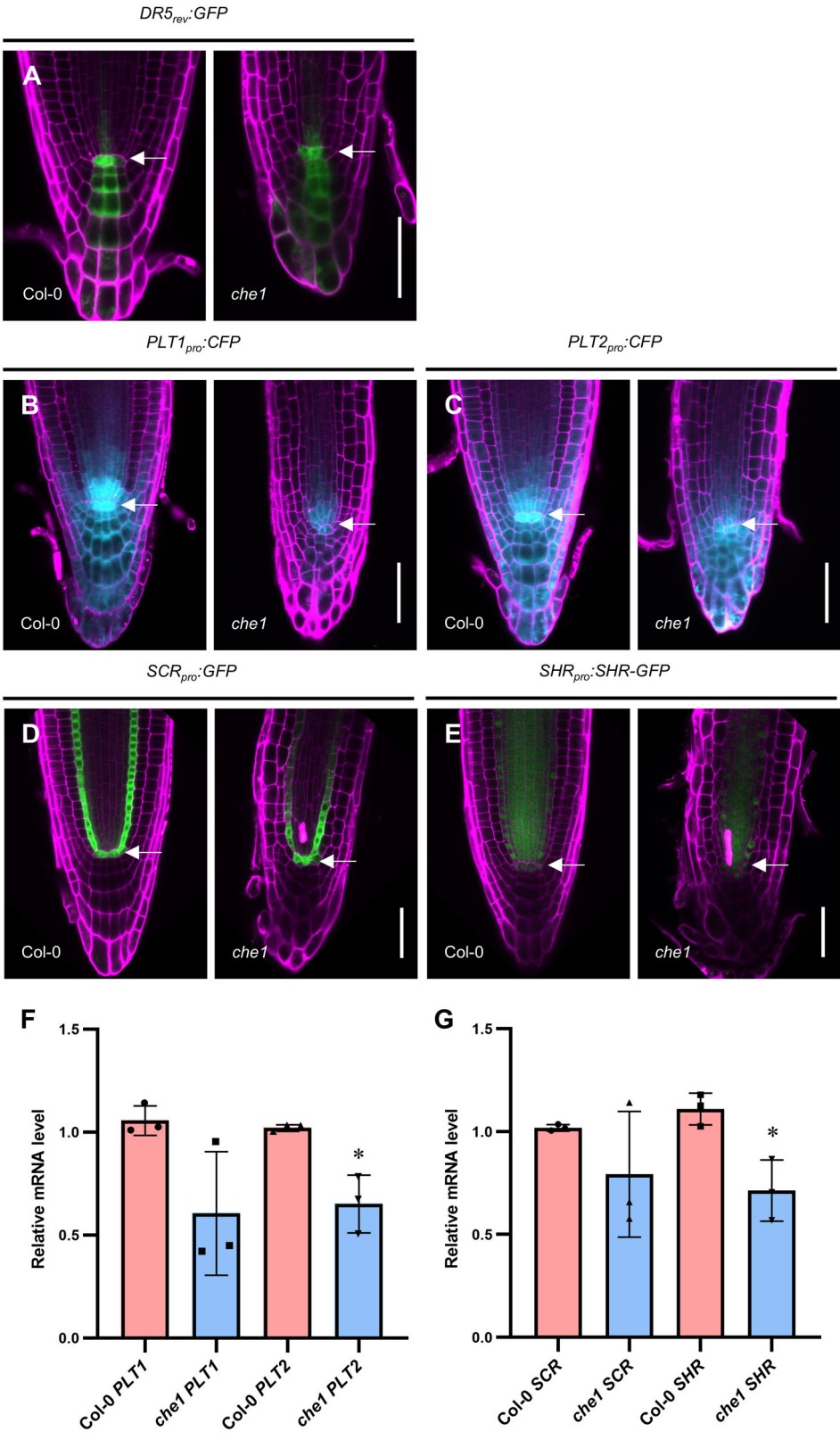

DNA damage. The comet assay showed that the level of damaged DNA was higher in the *che1* mutant compared to the wild type (Fig. 5B, C). With the use of immunofluorescence staining with the γH2AX antibody, we showed that *che1* had increased DNA damage compared to the control under normal growth conditions (Fig. 5D, E). The number of γH2AX foci in the *che1* mutant was similar to that of Col-0 treated with 10 μM zeocin, a radiomimetic drug that causes DSBs (Fig. 5A, D, E). The increased expression level of DNA damage marker genes, such as *AtBRCA1* and *AtPARP2*, in the *che1* mutant further supported our hypothesis that the absence of *AtCHE1* leads to DNA damage (Fig. 5F, G).

In mammals, AATF/Che1 plays an essential role in DNA damage response pathway, contributes to cell cycle arrest and apoptosis

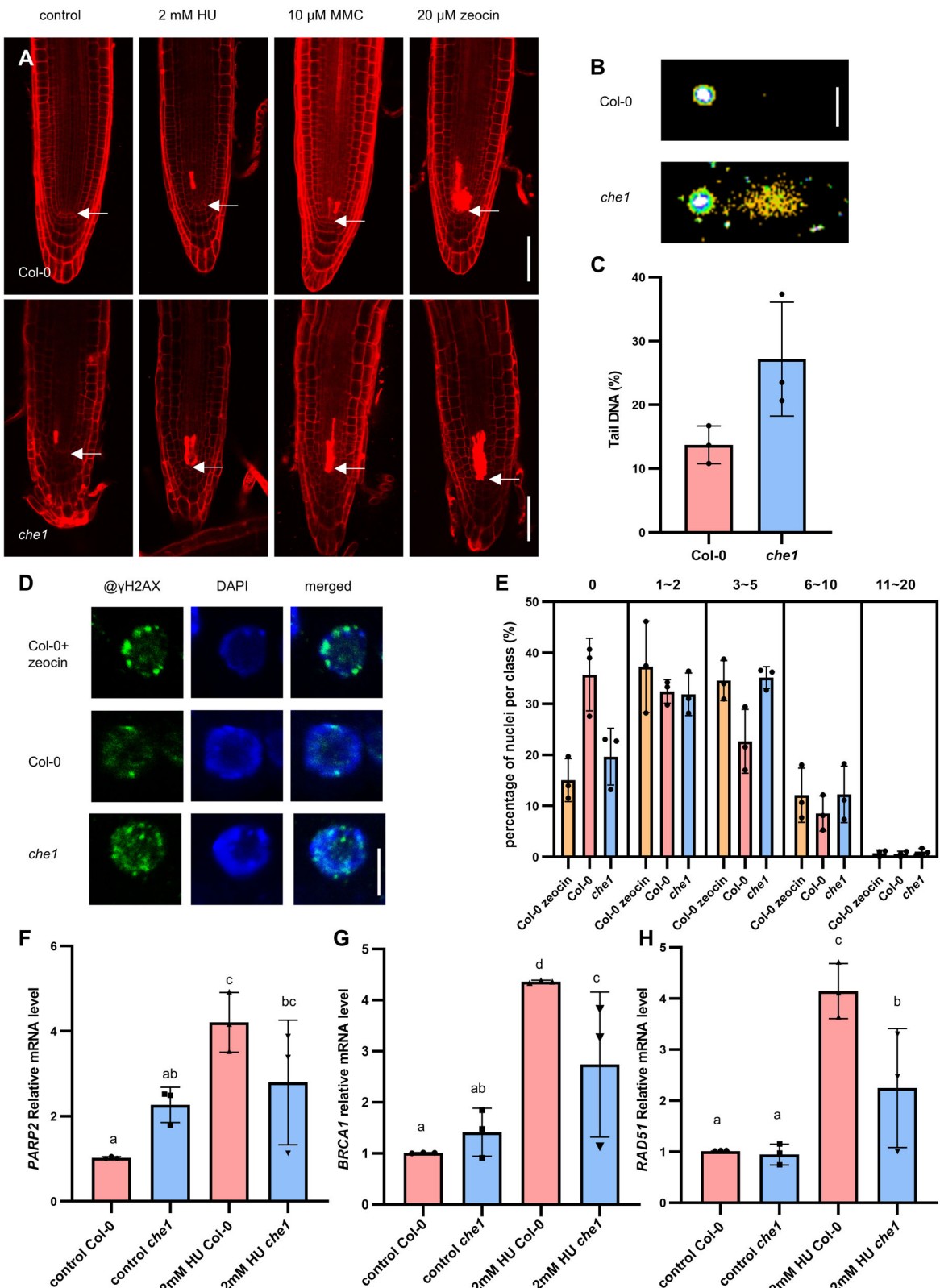

depending on the DNA damage level[34]. To study whether *che1* is sensitive to genotoxic agents, we tested its response besides upon hydroxyurea (HU, 2 mM), also upon mitomycin C (MMC, 10 μM) and zeocin (20 μM) treatment (4 DAG seedling, 24 h) to induce replication stress, DNA cross-linking, and DSBs, respectively. PI-stained roots, *che1*, and Col-0 as a control, were imaged by confocal microscopy. By measuring the area of dead cells in the proximal meristem, we could conclude that after HU, MMC, and zeocin treatments, the *che1* mutant was more sensitive to HU and MMC than Col-0 (Fig. 5A; Supplementary Fig. 5A, B). Furthermore, the increased expression levels of the DNA damage marker genes *AtBRCA1*, *AtRAD51*, and *AtPARP2*, induced by HU, were found to be lower in the *che1* mutant compared to Col-0

**Fig. 5 | che1 mutant develops DNA damage response. A** Confocal images of PI-stained root tips of Col-0 and *che1* under normal growth conditions, treated with 2 mM HU treatment, 10 μM MMC and 20 μM zeocin. Seedlings (4 DAG) were transferred to 1/2 MS medium with or without DNA-damaging agents. Scale bars = 50 μm; arrowheads point to QC position. **B** Representative comet assay image of nuclei from Col-0 and *che1* (7 DAG). Scale bars = 50 μm. **C** Levels of DNA fragments measured by the percentage of DNA in the tail of comets in the comet assay. The values and error bars represent means and ±SD, from three independent experiments. **D** γH2AX accumulation in the root tips of 5 DAG Col-0 on 1/2 MS medium containing 10 μM zeocin, Col-0 and *che1* grown under normal conditions measured by immunofluorescence staining. Blue, DAPI staining; green, γH2AX. Scale bar = 5 μm. **E** Quantification of γH2AX foci in Col-0 treated with 10 μM zeocin, Col-0 and *che1*. At least 100 nuclei were analyzed and grouped into 5 classes according to the number of γH2AX foci/nuclei. The data are from three independent experiments. Error bars indicate the SD. **F–H** illustrate relative expression level (gene vs reference gene) of DNA damage response genes quantified by qRT-PCR. Seedlings, 4 DAG, were grown in normal growth conditions and treated for 24 h with the relevant genotoxic agents. 2 mm Root tips (2 mm long) were collected. The values and error bars in (**F-H**) represent means and ±SD from three independent biological replicates. Columns with different letters indicate significantly differences at $P < 0.05$ (Duncan's multiple range means comparisons).

(Fig. 5F, G, and H). Based on these observations, we concluded that AtCHE1 protects genome integrity and is involved in the DNA damage response to replication stress.

Since the DNA damage responses, both cell death and DNA damage response (DDR) gene induction, are different in the *AtCHE1* mutant normal growth conditions and upon treatment, we asked whether *AtCHE1* gene expression itself is also triggered by genotoxic stress. Upon genotoxic treatment of the transgenic lines *AtCHE1_{pro}:GUS* and *AtCHE1_{pro}:AtCHE1_g-GFP* (5 DAG, 24 h), to 1/2 MS containing 2 mM HU, 10 μM MMC or 20 μM zeocin and grown for 24 hours. following GUS expression, we did not detect significantly changes upon HU compared to the seedlings grown under control conditions (Supplementary Fig. 6A). In contrast, the presence of GUS was significantly reduced upon zeocin treatment (20 μM) due to severe retardation of the root meristem (Supplementary Fig. 6A). qRT-PCR experiments confirmed these observations (Supplementary Fig. 6B). Meanwhile, *AtCHE1_{pro}:AtCHE1_g-GFP* confocal images showed that HU, MMC, and zeocin reduced AtCHE1 fusion protein levels (Supplementary Fig. 6C, D). It is worth to note that under zeocin treatment, the reduction of GFP and GUS expression is likely due to extensive cell death and shrinkage of mitotic zone.

### AtCHE1 plays a synergistic role with SOG1

To clarify the role of AtCHE1 in the DDR signaling pathway, we investigated its interaction with SOG1, the master regulator of DDR. The *sog1-1;che1* double mutant showed enhanced level of cell death under control conditions compared to either single mutant, *che1* and *sog1-1*, as well as Col-0 (Fig. 6). This elevated response indicates a synergistic interaction, with AtCHE1 and SOG1 maintaining meristematic cell integrity.

Given the importance of ATR kinase in the cellular response to replication stress, we also explored the relationship between AtCHE1 and ATR functions in DDR. We introgressed *atr-2* and *che1* mutants for this purpose. The *atr-2* mutant treated with hydroxyurea (HU, 2 mM) demonstrated a significant reduction in root growth (Fig. 7A–D, Supplementary Fig. 7). In contrast, the root length and number of meristematic cells in the *atr-2;che1* double mutant were comparable to those in the *che1* mutant, suggesting that the loss of AtCHE1 partially alleviates the root growth inhibition observed in the *atr-2* mutant under replication stress (Fig. 7A–D).

Lastly, we analyzed the ratio of roots exhibiting cell death and measured the cell death area in the *atr-2;che1* double mutant compared to the controls. Under normal growth conditions, the ratio of roots with cell death was similar between the *atr-2;che1* and *che1* mutants, while significantly lower in the *atr-2* mutant (Fig. 7E, G and H). However, upon HU treatment (24 hours), all genotypes, regardless of genetic background, displayed a cell death response (Fig. 7F, G). Notably, the extent of cell death varied significantly among the genotypes. The *atr-2* mutant displayed a robust cell death response compared to either Col-0 or *che1* (Fig. 7F), which was substantially diminished in the absence of AtCHE1 function (Fig. 7F, G and H). These findings suggest that the loss of AtCHE1 partially rescues impaired root development caused by the deficiency of ATR under HU stress.

## Discussion

AATF/Che-1, an RNA polymerase II-binding protein, plays a critical role in a wide array of cellular processes, including transcriptional regulation, cell proliferation, DNA damage response, apoptosis, and ribosome biogenesis in mammals. In this study, we demonstrate that AtCHE1, the Arabidopsis homolog of AATF/Che-1, is essential for root development and genome integrity.

Accumulating evidence underscores the importance of genome integrity in maintaining the SCN. Mutants such as *teb*, *fas* and *mms21*, which involve helicase and DNA polymerase domain encoding gene by the *TEBICHI*, as well as *FASCIATA* (*FAS*) gene that encodes subunits of the Arabidopsis counterpart of chromatin assembly factor-1 (CAF-1), and the *methyl methanesulfonate sensitivity gene21* functions as a subunit of the STRUCTURAL MAINTENANCE OF CHROMOSOMES5/6 complex, demonstrate severe meristem organization defects alongside cell death at the root tip[8,33,35]. Similar effects were induced by the mutations of *MAIN-related* genes, *MEDIATOR18* and *MERISTEM DISORGANIZATION 1* genes, lead to analogous defects. These genes encode components involved in plant-specific processes, such as aminotransferase-like plant mobile domain, a subunit of mediator complex, and an essential telomere protein in CTC1 (Cdc13)/STN1/TEN1 complex, respectively[6,7,36–38]. The RETINOBLASTOMA-RELATED (RBR) protein is also crucial for both stem cell maintenance and the DNA damage response[19,20,39].

In this study, we report that mutations in the *AtCHE1* gene result in a disorganized SCN and a cell death response at the root tip, similar to the phenotypes observed in *fas* or *mms21* mutants that lack functional telomeres or chromatin-organizing complexes. In addition to the SCN abnormalities, we observed exhaustion of the root meristem and disorganized cell division in the cortex and endodermal cells of the *che1* mutant. Increased DNA content in the comet tail, accumulation of γH2AX foci, and upregulation of DNA damage response genes indicated genomic instability in the *che1* mutant. Similar to the effects of genotoxic agents, the absence of AtCHE1 function led to reduced transcription of critical root development genes, including *WOX5*, *SHR*, *SCR*, *PLT1*, and *PLT2*[7,8].

At this point in our investigation, it remains difficult to determine whether the disrupted expression of root patterning genes leads to aberrant root formation in the *che1* mutant or if genome instability itself drives the DNA damage response, cell death, and premature differentiation. However, given the phenotypic similarities between the *che1* mutant and those observed in chromatin-organizing protein mutants, along with the anti-apoptotic role of AtCHE1's ortholog, AATF/Che-1, in animals, we hypothesize that AtCHE1 plays a primary role in ensuring genome stability, which is critical for maintaining the integrity of the root meristem.

Given the role of AATF/Che-1 in the DNA damage response in animals, we explored whether AtCHE1 serves a similar function in Arabidopsis. Unlike their animal counterparts, the inactivation of plant genes involved in DDR typically results in subtle phenotypes under normal growth conditions, but increases sensitivity to genotoxic stress. The *che1* mutant, akin to *wee1-1* and *atr-2* mutants, exhibits heightened cell death in response to hydroxyurea (HU) treatment[40]. It is noteworthy that the transcription levels of *BRCA1*, *RAD51*, and *PARP2* are elevated in mutants but remain lower than those observed in wild type. We assume either that the

**Fig. 6 | Cell death response in *che1* is independent on SOG1. A** Confocal images of PI-stained root tips of Col-0, *che1*, *sog1-1* and *sog1-1;che1* double mutants. Scale bars = 50 μm; arrowheads point to QC. **B** Measurement of dead cell area and **C** percentage of roots showing dead cells. The values and error bars in (**B**) and (**C**) represent means and ±SD of three independent experiments with at least 6 seedlings each. Columns with different letters indicate significant differences at *P* < 0.05 (Duncan's multiple range means comparisons).

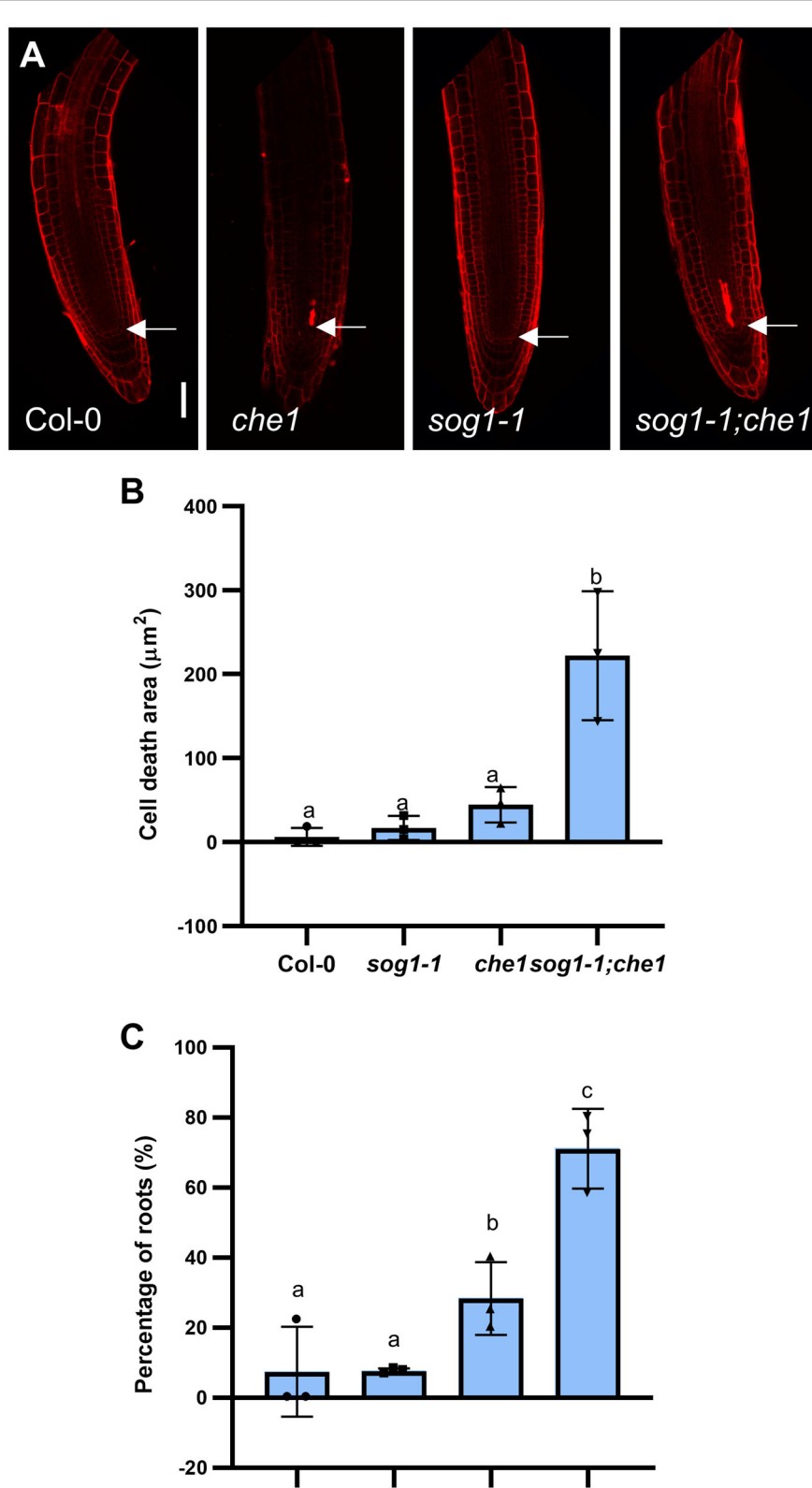

transcription of these genes is affected by AtCHE1 or the accumulation of cell death in the mutant is due to the lack of BRCA1, RAD51 and PARP2 involved in DNA damage repair.

Since the ATR kinase plays a central role in the response to replication stress[13] we examined the relationship between the AtCHE1 and ATR signaling pathways. Upon HU treatment, the *atr-2;che1* double mutant displayed less severe cell death and only moderate growth impairment compared to the *atr-2* mutant alone. This phenomenon could be explained by the followings: either, ATR and AtCHE1 act in the same pathway, ATR is upstream or downstream of AtCHE1 or the retarded growth of the *che1* mutant partly restores the function of ATR, arresting the cell cycle in response to HU treatment. Additional experimentation is required to

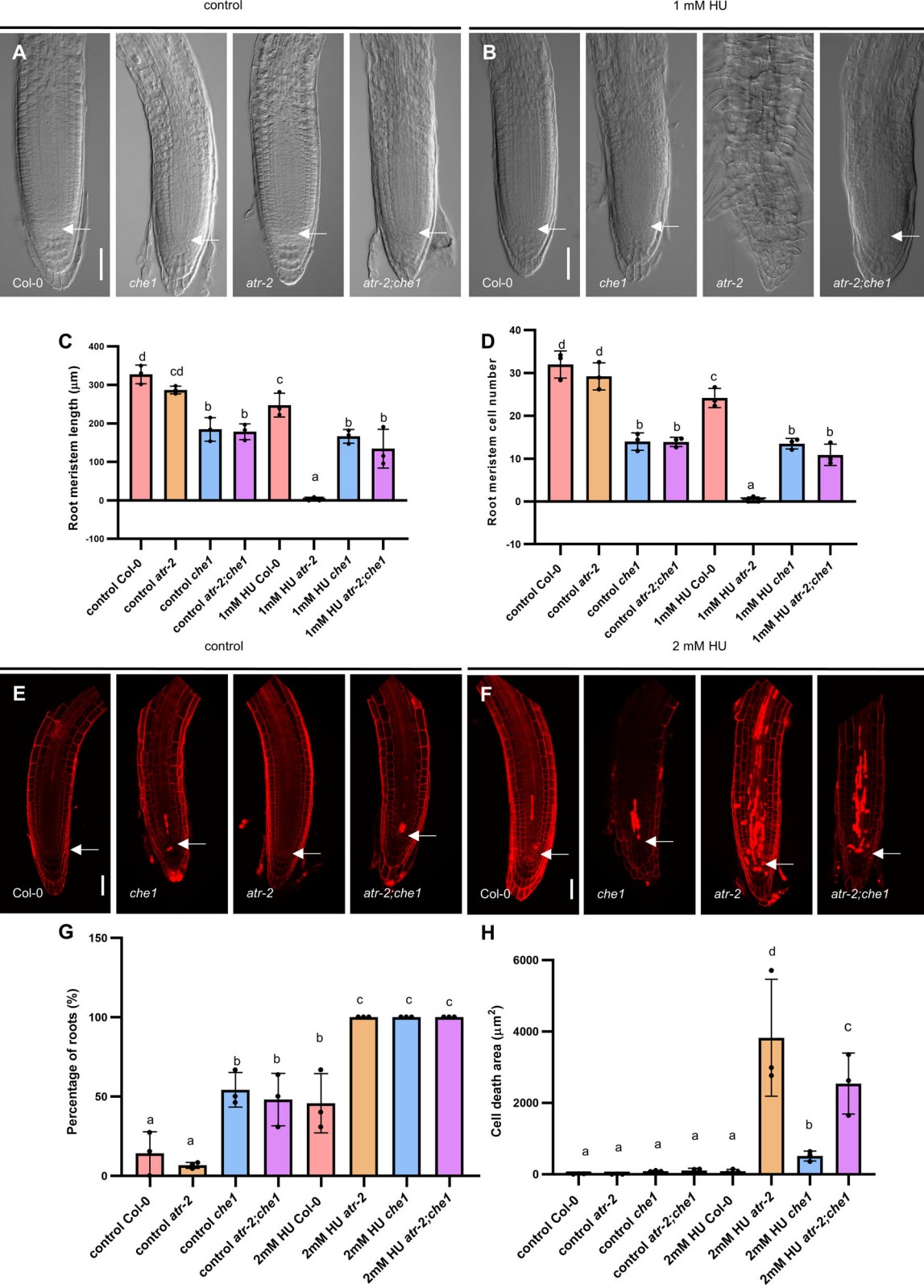

**Fig. 7 | Loss of AtCHE1 partially rescues impaired root development of *atr-2* under HU treatment. A**, **B** Representative images of root meristems of 6 DAG Col-0, *che1*, *atr-2* and *atr-2;che1*. Seedlings grown on 1/2 MS medium with or without 1 mM HU. Scale bars = 50 μm; arrowheads point to QC. **C** Measurement of root meristem length and **D** root meristem cell number of 6 DAG Col-0, *che1*, *atr-2* and *atr-2;che1*. Seedlings grown on 1/2 MS medium with or without 1 mM HU. **E** and **F** Confocal images of PI-stained root tips of Col-0, *che1*, *atr-2* and *atr-2;che1* double

mutants. 4 DAG seedlings were transferred and grown on 1/2 MS medium with or without 2 mM HU for 24 h. Scale bars = 50 μm; arrowheads point to QC. **G** Measurement of the percentage of roots showing dead cells. **H** Measurement of dead cell area from Col-0, *che1*, *atr-2* and *atr-2;che1* PI-stained roots. The values and error bars in (**C**, **D**) and (**G**, **H**) represent means and ±SD from three independent experiments with at least 8 seedlings each. Columns with different letters indicate significant differences at $P < 0.05$ (Duncan's multiple range mean comparisons).

provide evidence to verify these options. Moreover, genetic analysis indicates that cell death in the *che1* mutant operates independently of SOG1, with AtCHE1 and SOG1 playing synergistic roles in promoting cell survival. In mammals, AATF/Che-1 activity is regulated post-translationally through phosphorylation, ubiquitination, poly(ADP-ribosyl)ation, and conformational changes[23,41,42]. Further investigations should provide evidence for post-transcriptional and post-translational regulation of AtCHE1 in response to DNA damage stress. Similar to its homolog AATF, AtCHE1 has a predicted nuclear and a nucleolar localization signal, and is predominantly detected in the nucleus. The cellular localization of AtCHE1 provides important insight into its function, paralleling the role of the mouse AATF/Che-1 homolog Traube (Trb), which is crucial for the growth of pre-implantation embryos. *trb* mutant embryos exhibit reduced total cell numbers, developmental arrest, and a lack of ribosomes, polyribosomes, and rough endoplasmic reticulum[43]. The absence of ribosomes in the *trb* mutant and the nucleolar localization of Traube suggested that its involvement in ribosome synthesis. The research for the refined and dynamic subcellular localization of AtCHE1 is important for studying *AtCHE1* gene function.

The *che1* mutant may retain the complete AATF motif (290 amino acids) while lacking the TRAUB motif, suggesting that it is likely a weak allele. Using a single mutant to study gene function has limitations. For example, if the AtCHE1 knockout causes early lethality, a weak allele might mask its role in embryonic development. To comprehensively investigate gene function, it is essential to obtain a broader variety of mutants, including knockout mutants, knockdown mutants, and single or multiple amino acid substitution mutants.

In conclusion, our results indicate that AtCHE1 plays a critical role in the regulation of genome integrity and stem cell maintenance in Arabidopsis.

## Materials and methods
### Plant materials and growth conditions
In this study, we utilized the *Arabidopsis thaliana* Columbia-0 (Col-0) ectopy is used as the wild type. The marker lines employed have been previously characterized: $WOX5_{pro}$:GFP[44], $SCR_{pro}$:GFP[4,45], $SHR_{pro}$:SHR-GFP[46], $PLT1_{pro}$:CFP[32], $PLT2_{pro}$:CFP[32], J2341[47], $CO2_{pro}$:H2B-YFP[28], $CO3_{pro}$:H2B-YFP[28], and $DR5_{rev}$:GFP[48], all of which are in the Col-0 ecotype background. The *atr-2* (SALK_032841) and *che1* mutants utilized in this research are also based on the Col-0 ecotype. The *sog1-1* mutant has been described previously[49]. Additionally, the *35S:H2B-tdTomato* line was generously provided by Prof. Dr. Thomas Laux. Sequence information for the primers used for genotyping is provided in Supplementary Table 1.

Surface-sterilized seeds were planted on 1/2 Murashige and Skoog (MS) (containing 1% sucrose, 1.3% Agar, pH5.7) and incubated at 4 °C in the dark for 2 days. Subsequently, the seeds were transferred to a growth environment maintained at 22 °C with a photoperiod of 16 hours light and 8 hours dark. For growing in soil, seedlings were transferred to soil between 10–14 days and grown under the aforementioned conditions in a growth chamber.

For DNA damage reagents treatments, MMC (Mitomycin C from *Streptomyces caespitosus*, M0503, Sigma) was prepared by adding 1 vial to 4 ml $H_2O$ as a stock solution, stored stock solution at 4 °C. HU (H8627 Sigma) was dissolved in $H_2O$ to create a 0.5 M stock solution and stored at −20 °C. 100 mg ml$^{-1}$ Zeocin stock solution was bought from Invitrogen (R25001) and stored at −20 °C. For long time (6 days) and short time (24 h) treatment, diluted 100 mg ml$^{-1}$ zeocin to 10 μM and 20 μM, 0.5 M HU to 1 mM and 2 mM and 0.5 g l$^{-1}$ MMC to 2.5 mg l$^{-1}$ and 10 μM in 1/2 MS medium, respectively. Seedlings were grown (long-time treatment) or transferred (short-time treatment) to the medium containing DNA damage reagents and grown under the described conditions.

### Protein structure prediction
Nuclear localization signals were predicted by NLS mapper (http://nls-mapper.iab.keio.ac.jp)[50–52]. For detecting nucleolar localization sequence,

NOD: NuclOlar localization sequence Detector (dundee.ac.uk) was utilized (http://www.compbio.dundee.ac.uk/www-nod/index.jsp)[53,54]. The phosphorylation sites of AtCHE1 protein were predicted via the NetPhos 3.1 server (http://www.cbs.dtu.dk/services/NetPhos/)[55]. Finally, the protein domain structure was generated using DOG software for visualization of the protein domains (version 2.0)[56].

### EMS mutagenesis and map-based cloning of *AtCHE1*
In this study, we conducted EMS mutagenesis on an Arabidopsis population (ecotype: Col-0) and isolated mutants according to the protocols established by Jander et al. and Yu et al.[57,58]. Col-0 seeds were soaked in a 50 ml Falcon tube containing 40 ml of sterile 100 mM phosphate buffer at 4 °C with continuous shaking. Subsequently, the phosphate buffer was removed and replaced with 40 ml of fresh sterile 100 mM phosphate buffer supplemented with 0.4% EMS. The seeds were then incubated at room temperature with gentle shaking. Following incubation, the seeds were washed with sterile water 20 times. Finally, the washed seeds were dried under a hood, resulting in the EMS-mutagenized population (M1). To map the *AtCHE1* locus, we employed standard map-based cloning techniques as detailed by Lukowitz et al.[29]. The *che1* mutant, also in the Col-0 background, was crossed with Landsberg *erecta* to generate an F2 mapping population. Initial coarse mapping localized the *AtCHE1* gene to a region on chromosome 5 between MTE17 and MSN2. For fine mapping, we utilized simple sequence length polymorphisms (SSLPs) and derived CAPS (dCAPS) markers. Sequence analysis revealed a G to A mutation in the AT5G61330 gene, which caused RNA mis-splicing. A 5700 bp genomic fragment from the wild type (Col) AT5G61330 locus, which includes an additional 2197 bp upstream, a 2817 bp transcribed region, and a 686 bp downstream sequence, was introduced into the *che1* mutant. This fragment effectively complemented the short root and dwarf phenotype of the *che1* mutant. The sequence information for the primers used in mapping and the mutation site of the *che1* mutant can be found in Supplementary Table 2 and Supplementary Table 3. Additionally, genomic DNA and coding sequence information for both the wild type and the *che1* mutant are provided in Supplemental Data 1.

### Plasmid construction and generation of transgenic lines
To construct $AtCHE1_{pro}$:GUS, the *AtCHE1* promoter (comprising 2197 bp upstream and 201 bp transcribed region) was amplified from BAC MFB13 using primers 164-6 and 164-7. The Pvu I and Asc I cleavage sites were introduced using these primers, with Pvu I being compatible with Pac I. The PCR product was digested with Pvu I and Asc I, while the *pMDC162* vector was digested by Pac I and Asc I. The *AtCHE1* promoter fragment was then inserted into *pMDC162*.

For phenotypic complementation of the mutant lines, we amplified the native *AtCHE1* promoter along with the full-length genomic DNA (comprising 2197 bp upstream, 2817 bp transcribed region, and 686 bp downstream) from BAC MFB13, using primers 164-35 and 164-11. Sbf I and Sac I cleavage sites were introduced with these primers. The *pCR-Blunt* intermediate vector containing the PCR product was digested with Sbf I and Sac I, as was pMDC85 after the removal of the sequence containing the CaMV35S promoter and GFP tag between these cleavage sites. $AtCHE1_{pro}$:$AtCHE1_g$-$AtCHE1_{3'-UTR}$ construct was generated following this procedure.

To investigate the subcellular localization of the AtCHE1 protein, the native *AtCHE1* promoter and full-length genomic DNA (excluding the stop codon) were amplified from BAC MFB13 using primers 164-35 and 164-12. The resulting PCR product was similarly digested with Sbf I and Asc I, and inserted into *pMDC85* after removing the CaMV35S promoter sequence. This generated the $AtCHE1_{pro}$:$AtCHE1_g$-GFP construct.

All constructs were introduced into the Agrobacterium strain GV3101 and transformed into Arabidopsis through the floral dip method[59]. Sequence information for the primers used to construct the vectors is provided in Supplementary Table 4.

## Histochemical analysis

The activity of the reporter enzyme β-glucuronidase (GUS) was assessed in transgenic plants using X-Gluc as a substrate for enzyme activity. Seedlings were first fixed in 90% acetone for 20 min. Then, rinsed with GUS staining buffer comprised of 50 mg l$^{-1}$ X-Gluc; 100 mM NaH$_2$PO$_4$; 10 mM Na$_2$EDTA; 0.5 mM K ferrocyanide; 0.5 mM K ferricyanide; 1‰ Triton X-100 pH = 7.0. After vacuum infiltration in GUS staining solution for 2~3 min the seedlings were incubated in the solution at 37 °C. Following incubation, they were washed twice with PBS buffer and subsequently in 70% Ethanol. The seedlings were then mounted in 40% glycerol. Tissue samples were photographed using an AxioImager A1 microscope (Carl Zeiss Microimaging) and a SteREO Discovery V20 stereomicroscope (Carl Zeiss Microimaging) connected to an AxionCam MRc digital camera (Carl Zeiss Microimaging).

## Phenotypic analysis

For assessing primary root length, seedlings were scanned, and root lengths were measured using Image J (National Institutes of Health; http://rsb.info.nih.gov/ij). For the root meristem, roots were mounted in a chloral hydrate solution (chloral hydrate: water: glycerol=4:3:1) and photographed with an AxioImager A1 microscope (Carl Zeiss Microimaging). Meristem length and cell number were calculated using Image J, based on the region between the QC and the first elongated cell in the cortical cell file[60,61]. For PI staining, seedlings were rinsed in a 10 μg ml$^{-1}$ PI for 2 min. After a thorough rinse in water, roots were mounted in water for analysis via confocal laser scanning microscopy.

## qRT-PCR

Gene expression was evaluated using quantitative RT-PCR (qRT-PCR). Total RNA was extracted from 2 mm root tips of seedlings at 5 DAG using the RNeasy Plus Micro Kit (cat. No. 74034 QIAGEN). Prior to cDNA synthesis, the RNA samples were incubated with 1 μl DNase I at 37 °C for 30 min (Thermo Scientific), followed by the addition of 1 μl 50 mM EDTA at 65 °C for 10 min to eliminate any residual DNA. RNA concentration was measured using a NanoDrop 1000 spectrophotometer (PRQlab), and RNA integrity was assessed via agarose gel electrophoresis. Reverse transcription was conducted using Thermo Scientific ReverAid Reverse Transcriptase (EP0041). qRT-PCR was performed with Maxima SYBR Green qPCR Mastermix (Thermo Scientific K0222) on an ABI 7300 real-time PCR system (Applied Biosystems). The primers utilized for qRT-PCR are listed in Supplementary Table 5, with *UBIQUITIN10* (*UBQ10*) serving as the reference gene.

## Confocal microscopy

Fluorescence imaging was conducted using a Zeiss LSM 510 microscope (Carl Zeiss Microimaging) and an A1 HD25/A1R HD25 confocal microscope (Nikon). A 488 nm line laser was employed for the excitation of GFP and Alexa Fluor 488 picolyl azide, with emission detected from 505 to 530 nm. For CFP, a 458 nm line laser was used, and emission were collected from 465 to 520 nm. PI and tdTomato fluorescene were excited at 561 nm, with emissions recorded between 590 and 653 nm. DAPI was excited at 408 nm, and emissions were detected at 480 nm. Images were processed using ZEN lite (Carl Zeiss Microimaging) and NIS-Elements Viewer (Nikon). Fluorescence intensity and plot profiles were analyzed using Image J software (National Institutes of Health; http://rsb.info.nih.gov/ij) and Fiji[62].

## Comet assay

The comet assay was performed following established methods[63,64], and [16]. Briefly, 7-day-old seedlings were dissected using a fresh razor blade on ice in PBS buffer containing 50 mM EDTA. The resulting nuclear suspensions were filtered through a 40 μm sieve. This nuclear suspension was mixed with 1% low-melting-point agarose at a 1:10 (v/v) ratio. Two drops, each containing 50 μl of the nuclear suspension, were placed separately on precoated slides and covered with 22 mm × 22 mm coverslips. The slides were incubated at 4 °C for 60 minutes, followed by treatment with a neutral lysis

solution (high salt: 2.5 M NaCl, 10 mM Tris-HCl, pH 7.5, 100 mM EDTA) for 20 minutes at room temperature (N/N protocol). The slides were then equilibrated in 1× TBE buffer for 5 minutes and subjected to electrophoresis in the same buffer for 6 minutes at 25 V (1 V cm$^{-1}$). Excess electrophoresis solution was drained from the slides and the slides were gently rinsed twice in dH$_2$O for 5 minutes each, followed by a 5-minute rinse in 70% ethanol. The slides were then dried on a sterile bench. Dried agarose gels were stained with 10 μg ml$^{-1}$ PI and covered with coverslips. Comets were observed using an AxioImager A1 fluorescence microscope (ZEISS). Comet analysis was conducted using TriTek Comet Score software, measuring at least 100 comet images from each slide while avoiding doublets or comets at the slide edges.

## Immunofluorescence staining

5 DAG plants were incubated for 30 min in fixation buffer (4% paraformaldehyde in 100 ml MTSB, MTSB buffer: 50 mM PIPES, 5 mM MgSO$_4$, 5 mM EGTA, pH 6.9) in the cold room (4 °C). Cell wall digestion, blocking, primary antibody incubation, secondary antibody incubation, and nuclear co-staining of the nucleus were performed as previously described[65] and [20]. For cell wall digestion, 600 μl enzyme mix (2.5% pectinase (Mazerozym R10), 2.5% cellulase (Onozuka RS), and 2.5% pectolyase (Y-23) in buffer MTSB) was added to new wells. Seedlings were then transferred into the enzyme mix and incubated at 37 °C for 8 minutes. Root tips were excised under a binocular microscope and placed into MTSB buffer. Subsequently, root tips were transferred onto the slides and dried on a sterile bench. For blocking, 60 μl of blocking solution (2% BSA in MTSB.) was applied to the area containing the root tips, followed by incubation at room temperature for 1 hour. For primary antibody incubation, the blocking solution was replaced with 60 μl of primary antibody solution (γH2AX antibody), which was incubated for 1 hour at 37 °C. The root tips were washed 3 times with 100 μl MTSB buffer. For secondary antibody incubation, 60 μl of the secondary antibody solution (Goat anti-rabbit Alexa Fluor 488) was added to the root tips and incubated for 1 hour at 37 °C. The root tips underwent 3 washes with MTSB buffer. To co-staining the nucleus, 100 μl of the DAPI solution (0.2 mg l$^{-1}$) was added to the root tips and incubated for 10 minutes before washing 3 times with dH$_2$O for 5 minutes each time. A rabbit anti-plant γH2AX antibody[66] and a goat anti-rabbit IgG (H + L) highly cross-adsorbed secondary antibody, Alexa Fluor 488 (Invitrogen, A-11034), were utilized at a working concentration diluted to 1:500 in an antibody dilution buffer (1% BSA in MTSB).

## Accession numbers

Sequence data from this article is available from the Arabidopsis Genome initiative under the following accession numbers: AT5G61330 (*AtCHE1*), AT3G11260 (*WOX5*), AT4G05320 (*UBQ10*), AT3G20840 (*PLT1*), AT1G51190 (*PLT2*), AT3G54220 (*SCR*), AT4G37650 (*SHR*), AT5G40820 (*ATR*), AT4G21070 (*BRCA1*), AT4G02390 (*PARP2*), AT1G25580 (*SOG1*), and AT5G20850 (*RAD51*). And the accession numbers of protein in phylogenetic analysis (Supplementary Fig. 2 and Supplementary Fig. 3) are provided as follow: NP_001331141.1 (*Arabidopsis thaliana*), XP_003552502 (*Glycine max*), NP_036270 (*Homo sapiens*), NP_062790 XP_919127 (*Mus musculus*), XP_015623660 (*Oryza sativa*), AJU67199 (*Saccharomyces cerevisiae*), NP_001150275 (*Zea mays*).

## Statistics and reproducibility

Statistical analysis was conducted using IBM SPSS statistics 21 and Microsoft Excel. Data are presented as the mean ± standard deviation (SD). The two-tailed Student's *t*-test and one-way ANOVA test were used for comparing two groups and more than two groups, respectively. For the purpose of reproducibility, three independent biological replicates were utilized in qRT-PCR, comet assays, immunofluorescence staining, and the analysis of double mutant phenotypes. Additionally, at least three independent experiments were conducted to compare the responses to DNA damage treatments in *che1* and Col-0. The sample size and the number of experimental repetitions were detailed in the figure legend.

## Data availability

All data supporting the findings of this study are available from the corresponding author upon reasonable request. The source data underlying the graphs presented in this paper can be found in Supplementary Data 2.

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

## Acknowledgements

We thank Prof. Dr. Yan Zhang for comments on the manuscript. This work was supported by the initiation fund from the Shandong Agricultural University, National Natural Science Foundation of China (31570291); Shandong "Foreign experts double hundred" Program (WST2017008), and Natural Science Foundation of Shandong Province (ZR2021MC175). In additional the work was supported by Bundesministerium für Bildung und Forschung [BMBF FKZ 031B0556]; the Excellence Initiative of the German Federal and State Governments [EXC 294, SFB746]; and the Deutsches Zentrum für Luft und Raumfahrt [DLR 50WB1022]. Beatrix. M. Horvath was funded by a Marie-Curie IEF fellowship (FP7-PEOPLE-2012-IEF-330789).

## Author contributions

X.L. and K.P. designed the research and supervised the experiments; X.L., F.L., and K.P. wrote the manuscript; B.M.H. improved the preliminary version of the manuscript; F.L., B.W., and X.W. performed most of the experiments; B.M.H., L.D.V., D.D., and S.Q. contributed reagents, materials, analytical tools, and suggestions.

## Funding

## Competing interests

The authors declare no competing interests.
