## [Transparent Peer Review file · Communications Biology]

AtCHE1, the Arabidopsis homolog of mammalian AATF/Che-1 protein, is involved in safeguarding genome stability

Corresponding Author: Professor Klaus Palme

Version 0:

Reviewer comments:

Reviewer #1

(Remarks to the Author)

This manuscript proposes a new cascade based on molecular genetics and the discovery of a novel factor FAN, based on the premise that reduced genome stability may have detrimental effects on plant development. The fan mutant was identified by screening using ethyl methanesulfonate (EMS) as a mutagen. fan is a homolog of the mammalian AATF/Che-1 protein, which consists of an AATF/Che-1 domain and a TRAUB domain, and may have an impact on animal researchers as well as plant researchers. In the fan mutant, DNA damage and cell death responses were detected in root tips under normal conditions. On the other hand, exogenous administration of hydroxyurea reduced sensitivity compared to Col-0, suggesting that FAN is involved in the DNA damage response (DDR). Furthermore, FAN was shown to be involved in the DDR pathway regulated by ATM/RAD53-RELATED (ATR). These findings suggest that FAN is required for mitotic tissue maintenance and DNA damage response.

In this manuscript, the authors used Arabidopsis mutants for a meticulous phenotypic analysis, which clearly supports the involvement of FAN in DNA damage response through the DDR pathway. The sample size and statistical analysis are also adequate. With the following minor modifications, this manuscript is considered acceptable.

1. Table S2

Change g to G at the end of K11J9 454021_R2.

2. Figure S3

"In Arabidopsis" in the title should be changed.

3. Figure S7

Bars under X-axis are not aligned. Why don't you open the distance between two groups?

Reviewer #2

(Remarks to the Author)

The authors characterize an EMS-induced mutant they call fan, which exhibits slow root growth, a moderately disorganized stem cell niche, and a short mitotic zone. The mutant root tips also display moderately enhanced (significant at $P < 0.05$) spontaneous cell death in what appears to be both the stem cell niche and the mitotic zone (the authors do not distinguish between the two zones). The mutant root tips express slightly elevated (2x) levels of a few tested transcripts that are known to be induced during DNA Damage Response, increased (2x?) spontaneous DSBs, and enhanced numbers of gamma-H2AX foci (also indicative of DSBs). All of the above phenotypes are consistent with a plant that is experiencing higher than normal levels of DNA damage and a constitutive low grade DNA damage response. This a difficult-to-diagnose phenotype that includes (but isn't limited to!) defects in: metabolism (leading to enhanced spontaneous DNA damage), chromosome packaging, transcription, translation, replication, nucleotide pool hygiene, DNA repair, and DNA damage response. The authors also proposed errors in development might induce DNA damage.

The authors identify the defective locus by mapping, and find the loss of a splice acceptor in a homolog of a gene already

described in yeast and mammals as CHE1. This is a renaissance protein that appears to be involved in a wide range of functions in mammals and fungi, from (inhibiting) induction of cell death to construction of ribosomes. The authors clearly have identified a plant homolog of CHE1 (not described before in plants) and the phenotype observed is consistent with some sort of defect in genome maintenance, though, again, the precise nature of that defect is not determined here. The paper is generally well written, well organized, and the data well presented (thank you!). The major problem is the overstatement- in highly visible places like the title and abstract- of poorly supported conclusions. The authors tell us that the protein is inhibited by ATR and required for some aspects of ATR-triggered DDR, but I don't find their evidence convincing. I don't think this paper has impact outside the field of plant genome maintenance, as there are no new or even confirmatory insights into CHE1 function. However, with toning down of the ATR-related conclusions (see line 281- this is the way), I think this paper will be of interest to researchers in the field of plant genome maintenance.

I have a couple of suggestions below, just in the order they appear in the paper:

Abstract: "suggesting that FAN plays a role in DDR". As stated above, there are MANY possible explanations for this phenotype. Given the homology to CHE1, it's fair to lean towards your proposed regulatory role, but not because of the phenotype described here. The fact that the authors push an "ATR inhibits FAN model" without ever determining whether ATR is required for inhibition of transcript, protein, or for phosphorylation (if phosphorylation is occurring at all) is pretty strange- I realize they want to put this into a more developed paper, but that second paper can be the one that draws these conclusions.

L38 FAN isn't hypersensitive to HU. More must of the author's measurements (HU has zero effect on fan, but does have an effect on Col wt). Note the fold increase in cell death in fan with 2 mM HU is not significantly different from the fold increase in wt. There is an approx. 2x increase in root tip cell death in fan, but this is also true for Col.

A very easy measurement of DDR in plants is the induction of certain transcripts (BRCA1, RAD51 etc). These are induced very strongly in plants, in proportion to the DSBs and with dependence on ATM and SOG1. The authors measure a few transcript levels in the absence of exogenous DNA damaging agents, but suggest (not insist) that they also check to see if the mutant is defective in the induction of these transcripts in response to exogenous agents. The hyperexpression of these transcripts in the absence of exogenous damaging agents suggests fan is not defective in this pathway, and would justify their focus on ATR rather than ATM.

L107 and later... It's a troubling that the group has only one mutant, and that mutation affects splicing. This mutant might be a partial loss of function or- a different issue- generate a variety of splice products, creating a variety of proteins. The authors need to tell us whether there is only a single splice product- or at least a strongly predominant product- in the mutant cell. Please also include both your wt and mutant cDNA sequence in the supplement.

I also feel strongly that the authors need to generate a KO and tell us whether that's homozygous lethal or exhibits a phenotype equivalent to fan-1. The allele described here could be a weak allele. Hopefully they are already in the process of getting this line.

L145 add the gene number to the text here, and to figure 2.

L149 You haven't looked at the protein, so maybe say "resulting in a PREDICTED 290 aa truncation"..

L159 *Please take the yeast and mammalian CHE1 sequences and look to see if there are additional homologs in the Arabidopsis genome*

Why is the gene named FAN? It would be really helpful to name it AtCHE1 so readers have one less homolog to memorize.

L170 I don't know- what IS the rate of columellar cell division vs say cortex stem cell division? Give us numbers? Maybe genome maintenance isn't important in columellar (pre-root cap) cells but is in other cells? Expression is rather uniform through the cotyledons, which are not undergoing cell division. Why is expression high in veins? Does your result match published cell type specific transcriptome data? Also is this a description of a single transformed line, or a consensus of many?

L203 I'd say Fig 5A,F

L205 you mean "neutral" comet assay, I'm sure? Add that word.

L213 I'd add the DDR transcript data to Figure 5

L229 say this suggests that FAN suppresses cell death- as in mammals, I think? I really would play up this parallel more.

Could the entire phenotype be due to dysregulation of PCD? That would result in enhanced DSBs too.

DSB-induced SOG1 dependent programmed cells death occurs only in the stem cell niche. I would have liked to see more on stem cell nice vs mitotic zone cell death ("mitotic cell death" doesn't seem to be programmed at all; it's death due to cell division in the absence of functional DDR).

L238 reduced FUSION protein levels- you don't assay the wt protein

L241 The preservation of root meristem structure under HU stress in atr-/- is a VERY nice phenotype. But are you sure it's not simply because fan-/- is growing (and replicating its DNA) so slowly that ATR isn't needed (as much) to slow down growth? I remember the original work demonstrating the existence of programmed cell cycle response to DNA damage, in yeast, showed that cells defective in damage induced cell cycle arrest could be rescued by reducing growth rate through other means. And again, as mentioned above under abstract, fan is not "hypersensitive to HU"- perhaps there's more cell death (though the fold change is like wt) but there's no effect on meristem organization. *I would like to know if there's an effect on root growth or rate of germination- please show us!*

L258 "CHE-1 is essential for both root development and DNA damage response." The term essential is far too strong. Both root development (growth and differentiation) and DNA damage response are perhaps intact (to some degree induction of transcripts (in fan vs wt), though their inducibility by DNA damaging agents should be investigated, and cell death, though the cell death seen in fan hasn't been demonstrated to be programmed- to do this the authors would need to show that this requires regulatory factors like ATM and ATR, or more simply, SOG1).

L281 Nicely put! Yes! Now apply that same caution to the rest of the paper, which reaches unjustified conclusions.

L297 define root growth? *Please DO show the equivalent of figure 1E- root length, but for atr, HU etc. I would love to know how fan root growth in the presence of HU stacks up vs air*

Figure S3- include gene numbers for each of the homologs described here.

Reviewer #3

(Remarks to the Author)

In this paper, Liu et al. describe protein FAN, the putative homolog of transcription factor AATF/Che-1. Although I like the biology presented in this paper and fact that it provides the functional characterization of yet undescribed protein in Arabidopsis, I think that authors should provide stronger support for their hypothesis and there are still some open questions and many points of technical character that need to be addressed. Overall, it is not clear what is the function of the FAN itself, all data are based on indirect evidence.

Apart from the points described below, it is not completely clear, why the mutant was named fan. „The mutant, named fan, which we isolated from the EMS mutagenized Arabidopsis Col-0 population“. The name is quite confusing for the DNA damage community, as it points to the Fanconi anemia pathway genes (FAN1 and FANCON1 Anemia Complex components). I strongly suggest renaming the gene unless there is a clear relationship to existing Fanconi Anemia pathway genes.

The fan was found in the EMS mutagenesis screen. The correct mapping of the causal mutation was confirmed by complementation with the genomic construct. The description of causal mutation is a bit confusing. I suggest improving Figure 2A by alignment of WT and mutated sequences on both gDNA and protein levels. Regarding the protein domains in Figure 2B, it should be better integrated in the text, that the resulting truncated proteins still contain the AATF-Che1 domain. I am surprised by the relatively low homology of the protein as stated on lanes 156 – 157: “41.7% sequence similarity with the human AATF/Che-1 protein, which encodes the Apoptosis Antagonizing Transcription Factor, with 26.5 % sequence identity”. Although the authors provide evidence of a sequence similarity, the presence of conserved domains, and phylogeny (Figure S3), I think this is only limited evidence. Further testing is needed to consider this protein a true AATF/Che-1 homolog in plants. Add sequence alignment of protein domains and comparing the sequence of conserved residues would be helpful. Also, the accession numbers of proteins analyzed in Figure S3 should be provided in the Materials and Methods section. The phylogeny should be complemented with the alignment of the protein sequences.

For true homolog of RNA polymerase binding protein II, one would expect interaction/colocalization of the two proteins. Could the authors provide colocalization analyses, BiFC, or any other evidence that the proteins are localized in the same compartment?

In Figure 3J, the localization of FAN-GFP and DAPI doesn't show the localization of FAN-GFP in the cell nucleus. In fact, it looks like the image is somehow manipulated (different positions of blue objects in DAPI and merged images). I am not convinced that the protein is nucleolar and I suggest performing additional experiments with nucleolar markers (e.g. crosses with fibrillar marker lines or simple immunolabeling with anti-GFP and anti-fibrillar antibodies). The whole discussion should be improved. Is there any predicted function of FAN in Arabidopsis?

Minor points:

- Line 34 of the abstract At5G61330 should be changed either to AT5G61330 or At5g61330
- Abbreviations that needs explanation: AP2/PLT transcription factors, GRAS transcription factors; MAIN and MAIN Like family members; AtMMS21; WEE1;
- Line 50 this sentence is hard to follow „In addition to these important transcription factors, genome integrity in stem cells and their progeny cells is required for meristem maintenance.“
- Line 108 “atr-2;fan double mutant“
- In Figure NetPhos 3.1 prediction of the phosphorylation sites – this database is available only for a limited number of animal kinases. This should be carefully described in the text.
- Line 186 - „To analyze the auxin distribution and levels in fan, we followed the auxin marker DR5rev:GFP“ – the DR5 is used as a proxy for auxin accumulation, please rephrase
- Line 188 - „To follow the expression changes of the root master regulators, PLTs,“ – please rephrase to changes in protein localization/stability or something similar
- Line 221 „Based on these observations we concluded that FAN protects genome integrity and is involved in the DNA damage response upon replication stress and DNA cross-linking.“ – simple cell death assay is only indirect evidence of increased DNA damage

In conclusion, this is a nice story describing a new protein in Arabidopsis. The findings are valuable for the community, the DNA damage assays are well performed. My main concern is the homology with the AATF/che1 which should be toned down or better supported.

Version 1:

Reviewer comments:

Reviewer #2

(Remarks to the Author)

The is a review of a revised ms. The authors have identified a mutant with defect root tip meristem organization and slow growth, which they have named FAN. Map-based cloning reveals that this may be a homolog of the mammalian CHE-1 gene (which performs a wide range of functions) and it is the closest Arabidopsis homolog. The authors characterize many

aspects of the early seedling phenotype (similar to some other genome maintenance-defective mutants, such as *teb*), and come to a more specific conclusion about the function of FAN than I think is justifiable. I do appreciate discovering new-(ish) genes via EMS mutagenesis, and I do think this report will be useful, I just think the function of FAN, and there may be many functions (if it's a CHE-1 homolog), is still unclear. I think the phenotype described is consistent with some defect in genome maintenance- but as I said before, this could be due to defects in metabolism, repair, replication, or regulation of damage response. I don't agree that their results conclusively indicate that FAN operates upstream of ATR in a regulatory pathway. I think their results are fairly described in the Discussion, but over interpreted in the title and abstract.

L118 I might mention here that abbreviation of the mitotic zone is often observed in roots undergoing genomic stress.

L150 tell us briefly how you defined the promoter (maybe just everything up to the next predicted gene?)

Fig 2A Did you accidentally switch the labels for the mutant vs wt gDNA sequences? Didn't wt have a G that EMS turned into an A?

L173 Does GFP localize to the nucleus on its own?

L181 Fig 3 are you sure you have the right DAPI image in the middle of fig. 3J? Maybe the right image but it was flipped?

L215 if FAN is required for the prevention of DSBs under routine growth conditions (and it has a phenotype under routine growth conditions), we'd expect an increase in the expression of these genes. 5FGH assays very responsive DDR genes in the absence of exogenously applied DNA damaging agents. But your statistics in Fig. 5FGH indicate (a little surprisingly!) that they are not significantly different from the level of expression in wt (both columns are labeled a). So your "increased expression is not significant. Or is there an error in the figure?

L217 Be a little more specific here? Does CHE1 play a role in the induction or the suppression of the DDR in mammals.

L225 again, your figure 5F doesn't support the notion that PARP is induced to a level significantly different from wt. This "does transcriptional DDR occur" experiment is a really important experiment if you want to make a case for a role in the induction of damage response, rather than a role in preventing damage occurrence. Also, when you say "relative expression" on the Y axis, I assume you mean vs some control transcript, like actin. Not vs the uninduced level? Just clarify that in the figure legend.

L233 it's not clear that the level of expression between the two OE lines is significantly different, and the two lines don't seem to have the same phenotypes. I'd drop the whole OE section.

L244 The shrinkage of GUS expression is I'm sure due to the shrinkage of the mitotic zone with *zeo*. The diminishing of GFP expression may be due to the extensive cell death.

L258 this is a nice result indicating that the effects of *sog-* and *fan-* are synergistic- that they independently protect the genome via different pathways. The term "work together" is a little vague and suggests a more intimate collaboration, I'd drop that.

L269 again, I suggest that the restoration of HU resistance by FAN may simply because the *fan* defect is slowing growth down by a lot. One of ATR's roles is to arrest the cell cycle in response to the depletion of dNTPs induced by HU. Any mutation that drastically reduces cell division rate should enhance HU resistance. If FAN was truly upstream of ATR- required to induce ATR function in the presence of replicative stress- then its phenotype would be similar to that of the *atr* mutant. If it were instead upstream and required to suppress ATR function in the absence of replicative stress... then, hmmm, that might explain the constitutive slow growth. Would it also explain other phenotypes?

L262 this paragraph seems to ignore the wt phenotypes, just comparing the mutants. It just needs some corrections- for example, not all seedlings display cell death with HU.

Reviewer #3

(Remarks to the Author)

The authors significantly improved the text of the manuscript and sufficiently answered all my concerns and questions. I would still prefer to simply call the gene *AtCHE1* instead of FAN, but I understand the personal reasons why the authors doesn't want to change it. Anyhow, I recommend the article for publication.

Version 2:

Reviewer comments:

Reviewer #2

(Remarks to the Author)

I'm satisfied that this paper is ready for publication.

Dear Editor and Reviewers,

First of all, thank for valuable feedbacks. The reviewers' comments have greatly helped us to revise and improve our manuscript.

We have addressed most of the comments and summarized them in the attached document. Due to certain constraints, we were and are unable to perform new experiments; the first authors, Dr Fang Liu, and Dr Bingshan Wang in the meantime have different positions, and the research group supervised by Prof Dr Klaus Palme and Prof Dr Xugang Li does not exist either. Prof Dr Klaus Palme retired in 2020 and unfortunately Prof Dr Xugang Li has passed away due to cancer.

In line with the 'revised manuscript submission file checklist' provided by Communications Biology, we have retained the referees' original comments without modification in this Response to Reviewers document. The reviewers' comments, together with our responses, are organized numerically in tables for ease of the reviewing of the new version. Please note that the line numbers referenced in our responses correspond to those in the revised article, not those in the original submission manuscript.

In addition, bar graph with overlapping dots plot is provided in the Figures where it is relevant.

Sincerely

Prof. Dr. Klaus Palme

Original comments from reviewer #1 are presented below:

Reviewer #1 (Remarks to the Author):

This manuscript proposes a new cascade based on molecular genetics and the discovery of a novel factor FAN, based on the premise that reduced genome stability may have detrimental effects on plant development. The fan mutant was identified by screening using ethyl methanesulfonate (EMS) as a mutagen. fan is a homolog of the mammalian AATF/Che-1 protein, which consists of an AATF/Che-1 domain and a TRAUB domain, and may have an impact on animal researchers as well as plant researchers. In the fan mutant, DNA damage and cell death responses were detected in root tips under normal conditions. On the other hand, exogenous administration of hydroxyurea reduced sensitivity compared to Col-0, suggesting that FAN is involved in the DNA damage response (DDR). Furthermore, FAN was shown to be involved in the DDR pathway regulated by ATM/RAD53-RELATED (ATR). These findings suggest that FAN is required for mitotic tissue maintenance and DNA damage response.

In this manuscript, the authors used Arabidopsis mutants for a meticulous phenotypic analysis, which clearly supports the involvement of FAN in DNA damage response through the DDR pathway. The sample size and statistical analysis are also adequate. With the following minor modifications, this manuscript is considered acceptable.

1. Table S2

Change g to G at the end of K11J9 454021_R2.

2. Figure S3

"In Arabidopsis" in the title should be changed.

3. Figure S7

Bars under X-axis are not aligned. Why don't you open the distance between two groups?

Reply to reviewer #1

Table 1. Comments from reviewer# 1 and author's responses

Comments	Responses
----------	-----------

1. Table S2: Change g to G at the end of K11J9 454021_R2.	"g" was changed to "G" at the end of K11J9 45021_R2 in Table S2.
2. Figure S3: "In Arabidopsis" in the title should be changed.	"In Arabidopsis" was removed from the title of Figure S3.
3. Figure S7: Bars under X-axis are not aligned. Why don't you open the distance between two groups?	Due to the addition of a new figure labeled as "Figure S4," the original "Figure S7" has been renumbered to "Figure S8." In the updated "Figure S8D," we have new Bar-dot plot.

Original comments from reviewer #1 are presented below:

Reviewer #2 (Remarks to the Author):

The authors characterize an EMS-induced mutant they call fan, which exhibits slow root growth, a moderately disorganized stem cell niche, and a short mitotic zone. The mutant root tips also display moderately enhanced (significant at $P < 0.05$) spontaneous cell death in what appears to be both the stem cell niche and the mitotic zone (the authors do not distinguish between the two zones). The mutant root tips express slightly elevated (2x) levels of a few tested transcripts that are known to be induced during DNA Damage Response, increased (2x?) spontaneous DSBs, and enhanced numbers of gamma-H2AX foci (also indicative of DSBs). All of the above phenotypes are consistent with a plant that is experiencing higher than normal levels of DNA damage and a constitutive low grade DNA damage response. This a difficult-to-diagnose phenotype that includes (but isn't limited to!) defects in: metabolism (leading to enhanced spontaneous DNA damage), chromosome packaging, transcription, translation, replication, nucleotide pool hygiene, DNA repair, and DNA damage response. The authors also proposed errors in development might induce DNA damage.

The authors identify the defective locus by mapping, and find the loss of a splice acceptor in a homolog of a gene already described in yeast and mammals as CHE1. This is a renaissance protein that appears to be involved in a wide range of functions in mammals and fungi, from (inhibiting) induction of cell death to construction of ribosomes. The authors clearly have identified a plant homolog of CHE1 (not described before in plants)

and the phenotype observed is consistent with some sort of defect in genome maintenance, though, again, the precise nature of that defect is not determined here. The paper is generally well written, well organized, and the data well presented (thank you!). The major problem is the overstatement- in highly visible places like the title and abstract- of poorly supported conclusions. The authors tell us that the protein is inhibited by ATR and required for some aspects of ATR-triggered DDR, but I don't find their evidence convincing. I don't think this paper has impact outside the field of plant genome maintenance, as there are no new or even confirmatory insights into CHE1 function. However, with toning down of the ATR-related conclusions (see line 281- this is the way), I think this paper will be of interest to researchers in the field of plant genome maintenance.

I have a couple of suggestions below, just in the order they appear in the paper:

Abstract: "suggesting that FAN plays a role in DDR". As stated above, there are MANY possible explanations for this phenotype. Given the homology to CHE1, it's fair to lean towards your proposed regulatory role, but not because of the phenotype described here. The fact that the authors push an "ATR inhibits FAN model" without ever determining whether ATR is required for inhibition of transcript, protein, or for phosphorylation (if phosphorylation is occurring at all) is pretty strange- I realize they want to put this into a more developed paper, but that second paper can be the one that draws these conclusions.

L38 FAN isn't hypersensitive to HU. More must of the author's measurements (HU has zero effect on fan, but does have an effect on Col wt). Note the fold increase in cell death in fan with 2 mM HU is not significantly different from the fold increase in wt. There is an approx. 2x increase in root tip cell death in fan, but this is also true for Col.

A very easy measurement of DDR in plants is the induction of certain transcripts (BRCA1, RAD51 etc). These are induced very strongly in plants, in proportion to the DSBs and with dependence on ATM and SOG1. The authors measure a few transcript levels in the absence of exogenous DNA damaging agents, but suggest (not insist) that they also check to see if the mutant is defective in the induction of these transcripts in response to exogenous agents. The hyperexpression of these transcripts in the absence of exogenous damaging agents suggests fan is not defective in this pathway, and would justify their focus on ATR rather than ATM.

L107 and later... It's a troubling that the group has only one mutant, and that mutation affects splicing. This mutant might be a partial loss of function or- a different issue- generate a variety of splice products, creating a variety of proteins. The authors need to tell us whether there is only a single splice product- or at least a strongly predominant product- in the mutant cell. Please also include both your wt and mutant cDNA sequence

in the supplement.

I also feel strongly that the authors need to generate a KO and tell us whether that's homozygous lethal or exhibits a phenotype equivalent to fan-1. The allele described here could be a weak allele. Hopefully they are already in the process of getting this line.

L145 add the gene number to the text here, and to figure 2.

L149 You haven't looked at the protein, so maybe say "resulting in a PREDICTED 290 aa truncation"..

L159 *Please take the yeast and mammalian CHE1 sequences and look to see if there are additional homologs in the Arabidopsis genome*

Why is the gene named FAN? It would be really helpful to name it AtCHE1 so readers have one less homolog to memorize.

L170 I don't know- what IS the rate of columellar cell division vs say cortex stem cell division? Give us numbers? Maybe genome maintenance isn't important in columnellar (pre-root cap) cells but is in other cells? Expression is rather uniform through the cotyledons, which are not undergoing cell division. Why is expression high in veins? Does your result match published cell type specific transcriptome data? Also is this a description of a single transformed line, or a consensus of many?

L203 I'd say Fig 5A,F

L205 you mean "neutral" comet assay, I'm sure? Add that word.

L213 I'd add the DDR transcript data to Figure 5

L229 say this suggests that FAN suppresses cell death- as in mammals, I think? I really would play up this parallel more. Could the entire phenotype be due to dysregulation of PCD? That would result in enhanced DSBs too.

DSB-induced SOG1 dependent programmed cells death occurs only in the stem cell niche. I would have liked to see more on stem cell nice vs mitotic zone cell death ("mitotic cell death" doesn't seem to be programmed at all; it's death due to cell division in the absence of functional DDR).

L238 reduced FUSION protein levels- you don't assay the wt protein

L241 The preservation of root meristem structure under HU stress in atr-/- is a VERY nice phenotype. But are you sure it's not simply because fan-/- is growing (and replicating its DNA) so slowly that ATR isn't needed (as much) to slow down growth? I remember the original work demonstrating the existence of programmed cell cycle response to DNA damage, in yeast, showed that cells defective in damage induced cell cycle arrest could be rescued by reducing growth rate through other means. And again, as mentioned above under abstract, fan is not "hypersensitive to HU"- perhaps there's more cell death (though the fold change is like wt) but there's no effect on meristem organization. *I would like to know if there's an effect on root growth or rate of germination- please show us!*

L258 "CHE-1 is essential for both root development and DNA damage response." The

term essential is far too strong. Both root development (growth and differentiation) and DNA damage response are perhaps intact (to some degree induction of transcripts (in fan vs wt), though their inducibility by DNA damaging agents should be investigated, and cell death, though the cell death seen in fan hasn't been demonstrated to be programmed- to do this the authors would need to show that this requires regulatory factors like ATM and ATR, or more simply, SOG1).

L281 Nicely put! Yes! Now apply that same caution to the rest of the paper, which reaches unjustified conclusions.

L297 define root growth? *Please DO show the equivalent of figure 1E- root length, but for atr, HU etc. I would love to know how fan root growth in the presence of HU stacks up vs air*

Figure S3- include gene numbers for each of the homologs described here.

Reply to reviewer #2

Table 2. Comments from reviewer #2 and author's responses

Comments	Responses
1. The major problem is the overstatement- in highly visible places like the title and abstract- of poorly supported conclusions. The authors tell us that the protein is inhibited by ATR and required for some aspects of ATR-triggered DDR, but I don't find their evidence convincing. I don't think this paper has impact outside the field of plant genome maintenance, as there are no new or even confirmatory insights into CHE1 function.	We acknowledge the comments of reviewers 2, but refer to FAN as a plant protein, our results are limited to the observation that the atr-2;fan double mutant has a reduced response to HU treatment. To reflect this more accurately, we have revised the manuscript accordingly. Specifically, the original title, 'FAN, the homolog of mammalian Apoptosis Antagonizing Transcription Factor AATF/Che-1 protein, is involved in safeguarding genome stability through the ATR-induced pathway in Arabidopsis,' has been changed to 'FAN, the Arabidopsis homolog of mammalian AATF/Che-1 protein, is involved in safeguarding genome stability.'
2. Abstract: "suggesting that FAN plays a role in DDR" . As stated	In response to comment 1 we softened Abstract L45-L51. We have rewritten the

above, there are MANY possible explanations for this phenotype. Given the homology to CHE1, it's fair to lean towards your proposed regulatory role, but not because of the phenotype described here. The fact that the authors push an "ATR inhibits FAN model" without ever determining whether ATR is required for inhibition of transcript, protein, or for phosphorylation (if phosphorylation is occurring at all) is pretty strange- I realize they want to put this into a more developed paper, but that second paper can be the one that draws these conclusions.

summary of the DNA damage treatment result and conclusion to read 'Under standard conditions, the fan mutant exhibits a cell death response and differentiation defects at the root tip. When treated with hydroxyurea (HU), the mutant demonstrates a heightened sensitivity compared to Col-0, indicating that FAN is integral to the DNA damage response (DDR). Additionally, the phenotypic analysis of the *atr-2;fan* double mutant reveals that FAN and ATM/RAD53-RELATED (ATR) operate within the same pathway. Collectively, these findings underscore the importance of FAN in meristem maintenance and the DNA damage response.'

In response to comment 1, we have therefore deleted 'FAN is regulated by ATR' in line 58. However, the phenotype of *atr-2;fan* double mutant treated with or without HU shows the genetic interaction between two genes and leads to the conclusion that they act in the same pathway.

The same case in L137 to 141, 'Based on the phenotypic changes, we propose that FAN, the Arabidopsis AATF/Che-1 homolog, is involved in root development and DNA damage response and is localized in the nucleolus. Our analysis of the *atr-2;fan* double mutant shows that FAN is responsible for genome integrity through the ATR pathway'. was changed to 'Based on the phenotypic changes, we propose that the nuclear-localized FAN, the Arabidopsis AATF/Che-1 homolog, is

	involved in root development and genome stability. Our analysis of atr-2;fan double mutant shows that FAN and ATR act in the same signaling pathway.'
3. L38 FAN isn't hypersensitive to HU. More must of the author's measurements (HU has zero effect on fan, but does have an effect on Col wt). Note the fold increase in cell death in fan with 2 mM HU is not significantly different from the fold increase in wt. There is an approx. 2x increase in root tip cell death in fan, but this is also true for Col.	As for us, we only have cell death result with treatment. But the statistical result about cell death area indicates there is significant differences between control and HU treatment in fan but not in Col-0 (Figure S6B). And we made some modification in the text: L52-L54 "In the fan mutant, under normal conditions, we detected DNA damage, and cell death response at the root tip, while hypersensitivity to exogenously applied hydroxyurea (HU) compared to Col-0, suggesting that FAN plays a role in the DNA damage response (DDR)" is changed to "Under standard conditions, the fan mutant exhibits a cell death response and differentiation defects at the root tip." And in L354-L357, the summary of the drug treatments was also changed to "By measuring the area of dead cells in the proximal meristem, we could conclude that after HU, MMC and zeocin treatments, the fan mutant was more sensitive to HU and MMC than Col-0 (Figure 5A, Figure S6A, B),".
4. A very easy measurement of DDR in plants is the induction of certain transcripts (BRCA1, RAD51 etc). These are induced very strongly in plants, in proportion to the DSBs and with dependence on ATM and SOG1.	The relative mRNA levels of RAD51, PARP2 and BRCA1 in Col-0 and fan with or without HU treatment were added in Figure 5F, 5G and 5H. The results showed that the relative mRNA level of these three genes was increased in fan

The authors measure a few transcript levels in the absence of exogenous DNA damaging agents, but suggest (not insist) that they also check to see if the mutant is defective in the induction of these transcripts in response to exogenous agents. The hyperexpression of these transcripts in the absence of exogenous damaging agents suggests fan is not defective in this pathway, and would justify their focus on ATR rather than ATM.	under HU treatment compared to no exogenous agent but less than Col-0. This suggests that FAN is required in response to DNA damage. This result was added in L358-L361 “Furthermore, the increased expression levels of DNA damage marker genes AtBRCA1, AtRAD51 and AtPARP2, induced by HU, were found to be lower in fan mutant compared to Col-0 (Figure 5F, G, H)”.
5. L107 and later... It’s a troubling that the group has only one mutant, and that mutation affects splicing. This mutant might be a partial loss of function or- a different issue- generate a variety of splice products, creating a variety of proteins. The authors need to tell us whether there is only a single splice product- or at least a strongly predominant product- in the mutant cell. Please also include both your wt and mutant cDNA sequence in the supplement. *I also feel strongly that the authors need to generate a KO and tell us whether that’s homozygous lethal or exhibits a phenotype equivalent to fan-1*. The allele described here could be a weak allele. Hopefully they are already in the process of getting this line.	Due to the current situation in which the first authors are not available in the laboratory, we regret to inform you that we are unable to generate a knockout mutant at this time. The wild-type and CDS sequences are provided in in Supplementary Data 1.
6. L145 add the gene number to the text here, and to figure 2.	It is confusing that we did not mention any gene in L145, so we have not changed

	anything in this line. But we have made some changes in Figure 2A: we have shown part of FAN wild type and mutant gDNA, CDS, and predicted protein sequence to clarify the mutation in fan.
7. L149 You haven't looked at the protein, so maybe say "resulting in a PREDICTED 290 aa truncation" ..	The changes were made in L236-237: "result in premature translation termination 870 bp from the translation start site (Figure 2A), likely producing a truncated protein of 290 amino acids that still retains the part of domain."
8. L159 *Please take the yeast and mammalian CHE1 sequences and look to see if there are additional homologs in the Arabidopsis genome*	We performed a BLAST analysis of the protein sequences of the AATF/Che-1 homologues AJU67199 (Saccharomyces cerevisiae) and NP_03627 (Homo sapiens) using TAIR BLAST 2.9.0+ (https://www.arabidopsis.org/tools/blast/). The results of blasting the NP_03627 sequence showed that AT5G61330 received the highest score of 131, with an expectation of 5e-33, while the second gene AT44G34412 received a score of 32.3 with an expectation of 0.74. For AJU67199, AT5G61330 received the highest score of 52.4 with an expectation of 8e-07, and AT5G11910 received the score of 30. with an expectation of 44.1. In conclusion, AT5G61330 (FAN) appears to be the only homolog of AATF/Che-1 in Arabidopsis.
9. Why is the gene named FAN? It would be really helpful to name it AtCHE1 so readers have one less homolog to memorize.	One of the authors assigned the name FAN to AT5G61330. This name 'FAN' is derived from a character in a novel. When the gene was first discovered, scientists gave it a specific name. While it is recognized that multiple names for a

	single gene can lead to confusion, we have used this name for a long time and other collaborators have also adopted it in their research. Changing the name is therefore a major challenge for us.
10. L170 I don't know- what IS the rate of columellar cell division vs say cortex stem cell division? Give us numbers? Maybe genome maintenance isn't important in columnellar (pre-root cap) cells but is in other cells? Expression is rather uniform through the cotyledons, which are not undergoing cell division. Why is expression high in veins? Does your result match published cell type specific transcriptome data? Also is this a description of a single transformed line, or a consensus of many?	We did not quantify the division frequency of columella cells or cortical stem cells.. However, the cell cycle marker CyclinB1 is highly expressed in the root meristem but less so in the columella cell (Culligan, K., Tissier, A., & Britt, A. (2004). ATR regulates a G2-phase cell-cycle checkpoint in Arabidopsis thaliana. (Plant Cell, 16(5), 1091 – 1104.). As we showed in Figure 3A FAN is widely expressed in leaves, roots and flowers etc. This study did not focus on all these tissues such as seeds, flowers and shoots. Perhaps, FAN has additional functions in these tissues. In addition, we have independently transformed lines of FAN_{pro}:GUS lines and analyzed a line of FAN_{pro}:FANg-GFP. Plant ontology annotations indicate that AT5G61330 is expressed in hypocotyl, root, shoot system, flower, stamen, carpel, sepal, petal, inflorescence meristem, flower pedicel, vascular leaf, leaf apex, petiole, stem, cauline leaf, shoot apex, seed, plant embryo, cotyledon, leaf lamina base, collective leaf structure and guard cell (Schmid, M., Davison, T. S., Henz, S. R., Pape, U. J., Demar, M., Vingron, M., Schölkopf, B., Weigel, D., & Lohmann, J. U. (2005). A gene expression map of Arabidopsis thaliana development. Nature Genetics, 37(5), 501 – 506.). In conclusion,

	AT5G61330 is widely expressed in different plant tissues.
11. L203 I'd say Fig 5A,F	L336, '(Figure 5A) 'was changed to '(Figure 5A and S6A).
12. L205 you mean "neutral" comet assay, I'm sure? Add that word.	L337, 'we first used the comet assay' was changed to 'we first used neutral comet assay'.
13. L213 I'd add the DDR transcript data to Figure 5	Figure 5F, 5G and 5H were added to Figure 5.
14. L229 say this suggests that FAN suppresses cell death- as in mammals, I think? I really would play up this parallel more. Could the entire phenotype be due to dysregulation of PCD? That would result in enhanced DSBs too.	In L374, the conclusion was revised to "Taken together, these results suggest that FAN may help to regulate root growth and suppress cell death". We acknowledge that our measurements focused only on cell death without assessing DNA damage levels and that cell death is an indirect evidence of DNA damage.
15. DSB-induced SOG1 dependent programmed cells death occurs only in the stem cell niche. I would have liked to see more on stem cell nice vs mitotic zone cell death ("mitotic cell death" doesn't seem to be programmed at all; it's death due to cell division in the absence of functional DDR).	Thank you for your professional advice. We have examined the PI staining images of the Col-0 and fan mutant roots with and without zeocin treatment. We found that cell death typically occurs in mitotic cells and stele initial rather than in the stem cell niche except stele initial (epidermis/lateral root cap initial, columella initial, cortex/endodermal initial and quiescent centre) in fan under control conditions SCN epidermis/lateral root cap initial, columella initial, cortex/endodermal initial and quiescent centre vs mitotic cells and stele initial: 0:3, 0:5 and 0:3, data are from three independent experiments). Under zeocin treatment, the ratio of cell death happened in SCN (except stele initial)

	root number vs total root number that has cell death is 6:14, 9:12, 6:15 in Col-0 and 2:16, 3:16, 5:25 in fan. The cell death in the stele initial cell of fan is DSB induced programmed cell death and programmed cell death in the stem cell niche (such as columella stem cells) requires FAN under zeocin treatment.
16. L238 reduced FUSION protein levels- you don't assay the wt protein	L385: "fusion" was added in the sentence "Meanwhile, FAN_{pro}:FANg-GFP confocal images showed that HU, MMC, and zeocin reduced FAN fusion protein levels (Figure S9C, S9D)".
17. L241 The preservation of root meristem structure under HU stress in atr-/- is a VERY nice phenotype. But are you sure it's not simply because fan-/- is growing (and replicating its DNA) so slowly that ATR isn't needed (as much) to slow down growth? I remember the original work demonstrating the existence of programmed cell cycle response to DNA damage, in yeast, showed that cells defective in damage induced cell cycle arrest could be rescued by reducing growth rate through other means. And again, as mentioned above under abstract, fan is not "hypersensitive to HU" - perhaps there's more cell death (though the fold change is like wt) but there's no effect on meristem organization. *I would like to know if there's an effect on root growth or rate of germination- please show us!*	Thank you very much for your valuable comment. DNA damage can induce cell cycle arrest and endoreduplication in plants. The fan mutant shows slower root growth than Col-0 under control conditions. Furthermore, under hydroxyurea (HU) treatment, there is no significant difference in the primary root length of the fan mutant compared to that observed under control conditions (Figure S10). Taken together with the cell death data, HU induces cell death in the fan mutant without affecting root growth. For the atr-2;fan and fan under HU treatment, the root length of atr-2;fan is intermediate to that of atr-2 and fan (Figure 7 and Figure S10).

18. L258 “CHE-1 is essential for both root development and DNA damage response.” The term essential is far too strong. Both root development (growth and differentiation) and DNA damage response are perhaps intact (to some degree induction of transcripts (in fan vs wt), though their inducibility by DNA damaging agents should be investigated, and cell death, though the cell death seen in fan hasn't been demonstrated to be programmed- to do this the authors would need to show that this requires regulatory factors like ATM and ATR, or more simply, SOG1).	L421-L424, “In contrast, both root length and meristematic cell number of the atr-2;fan were comparable to those of the fan mutant, suggesting that the FAN function is essential for the development of the growth inhibition induced by replication stress (Figure 6A-D)” was revised to read “In contrast, the root length and number of meristematic cells in the atr-2;fan double mutant were comparable to those in the fan mutant, suggesting that the loss of FAN partially alleviates the root growth inhibition observed in the atr-2 mutant under replication stress (Figure 7A-D).” And the cell death level of sog1-1;fan double mutant under control condition is shown in Figure 6. The result indicates that cell death in the fan mutant is not dependent on SOG1. Loss of SOG1 in fan even causes higher cell death level. In summary, the genetic analysis between ATR and FAN, SOG1 and FAN, indicate that FAN helps to keep cell alive under normal condition and plays a synergistic role with SOG1. ATR and FAN act in the same pathway in response to HU-induced DNA damage stress.
19. L281 Nicely put! Yes! Now apply that same caution to the rest of the paper, which reaches unjustified conclusions.	Thank you for your kind reminder.
20. L297 define root growth? *Please DO show the equivalent of figure 1E- root length, but for atr, HU etc. I would love to know how fan root growth in the presence of HU stacks up vs air*	The root lengths of Col-0, fan, atr-2, and atr-2;fan grown on 1/2 MS medium with or without HU are shown in Figure S10. However, we did not monitor the growth curve of fan under HU treatment from

	seed germination to 6 days after germination (DAG).
21. Figure S3- include gene numbers for each of the homologs described here.	The gene numbers of the AATF/Che-1 homologues are listed in Materials and methods Accession numbers.

Original comments from reviewer #1 are presented below:

Reviewer #3 (Remarks to the Author):

In this paper, Liu et al. describe protein FAN, the putative homolog of transcription factor AATF/Che-1. Although I like the biology presented in this paper and fact that it provides the functional characterization of yet undescribed protein in Arabidopsis, I think that authors should provide stronger support for their hypothesis and there are still some open questions and many points of technical character that need to be addressed. Overall, it is not clear what is the function of the FAN itself, all data are based on indirect evidence.

Apart from the points described below, it is not completely clear, why the mutant was named fan. „The mutant, named fan, which we isolated from the EMS mutagenized Arabidopsis Col-0 population“. The name is quite confusing for the DNA damage community, as it points to the Fanconi anemia pathway genes (FAN1 and FANCONI Anemia Complex components). I strongly suggest renaming the gene unless there is a clear relationship to existing Fanconi Anemia pathway genes.

The fan was found in the EMS mutagenesis screen. The correct mapping of the causal mutation was confirmed by complementation with the genomic construct. The description of causal mutation is a bit confusing. I suggest improving Figure 2A by alignment of WT and mutated sequences on both gDNA and protein levels. Regarding the protein domains in Figure 2B, it should be better integrated in the text, that the resulting truncated proteins still contain the AATF-Che1 domain.

I am surprised by the relatively low homology of the protein as stated on lanes 156 – 157: “41.7% sequence similarity with the human AATF/Che-1 protein, which encodes the Apoptosis Antagonizing Transcription Factor, with 26.5 % sequence identity”. Although the authors provide evidence of a sequence similarity, the presence of conserved domains, and phylogeny (Figure S3), I think this is only limited evidence. Further testing is needed to consider this protein a true AATF/Ce-1 homolog in plants. Add sequence alignment of protein domains and comparing the sequence of conserved residues would be helpful. Also, the accession numbers of proteins analyzed in Figure S3 should be

provided in the Materials and Methods section. The phylogeny should be complemented with the alignment of the protein sequences.

For true homolog of RNA polymerase binding protein II, one would expect interaction/colocalization of the two proteins. Could the authors provide colocalization analyses, BiFC, or any other evidence that the proteins are localized in the same compartment?

In Figure 3J, the localization of FAN-GFP and DAPI doesn't show the localization of FAN-GFP in the cell nucleus. In fact, it looks like the image is somehow manipulated (different positions of blue objects in DAPI and merged images). I am not convinced that the protein is nucleolar and I suggest performing additional experiments with nucleolar markers (e.g. crosses with fibrillar marker lines or simple immunolabeling with anti-GFP and anti-fibrillar antibodies).

The whole discussion should be improved. Is there any predicted function of FAN in Arabidopsis?

Minor points:

- Line 34 of the abstract At5G61330 should be changed either to AT5G61330 or At5g61330
- Abbreviations that needs explanation: AP2/PLT transcription factors, GRAS transcription factors; MAIN and MAIN Like family members; AtMMS21; WEE1;
- Line 50 this sentence is hard to follow „In addition to these important transcription factors, genome integrity in stem cells and their progeny cells is required for meristem maintenance.“
- Line 108 “atr-2;fan double mutant“
- In Figure NetPhos 3.1 prediction of the phosphorylation sites – this database is available only for a limited number of animal kinases. This should be carefully described in the text.
- Line 186 - „To analyze the auxin distribution and levels in fan, we followed the auxin marker DR5rev:GFP“ – the DR5 is used as a proxy for auxin accumulation, please rephrase
- Line 188 - „To follow the expression changes of the root master regulators, PLTs,“ – please rephrase to changes in protein localization/stability or something similar
- Line 221 „Based on these observations we concluded that FAN protects genome integrity and is involved in the DNA damage response upon replication stress and DNA cross-linking.“ – simple cell death assay is only indirect evidence of increased DNA damage

In conclusion, this is a nice story describing a new protein in Arabidopsis. The findings

are valuable for the community, the DNA damage assays are well performed. My main concern is the homology with the AATF/che1 which should be toned down or better supported.

Reply to reviewer #3

Table 3. Comments from reviewer #3 and author's responses

Comments	Responses
1. Why the mutant was named fan	One of the authors assigned the FAN name to AT5G61330. This 'FAN' designation is derived from a character in a novel. Initially, scientists provided a specific name for the gene upon its discovery. While it is acknowledged that multiple names for a single gene can lead to confusion, we have utilized this name for an extended period, and other collaborators have adopted it in their research as well. Consequently, changing the name poses significant challenges for us.
2. Improving Figure 2A by alignment of WT and mutated sequences on both gDNA and protein levels. Regarding the protein domains in Figure 2B, it should be better integrated in the text, that the resulting truncated proteins still contain the AATF-Che1 domain.	Figure 2A has been reorganized. Portions of wild-type gDNA, CDS, and mutant gDNA, along with the corresponding protein sequences, are presented in Figure 2A to elucidate the mutation observed in the fan mutant at both the DNA and protein levels. The complete gDNA and CDS sequences can be found in Supplementary Data 1. Additionally, the predicted truncated protein containing AATF/Che-1 is also mentioned in L210.
3. Further testing is needed to consider this protein a true AATF/Ce-1 homolog in plants. Add sequence alignment of protein domains and comparing the sequence of	The protein sequence alignment was shown in Figure S4. And the AATF and TRAUB domain and the conserved residues was labeled with lines and asterisks.

conserved residues would be helpful.	
4. Also, the accession numbers of proteins analyzed in Figure S3 should be provided in the Materials and Methods section. The phylogeny should be complemented with the alignment of the protein sequences.	The phylogenetic tree has been updated in Figure S3. The protein alignments include sequences from Arabidopsis thaliana, Glycine max, Homo sapiens, Mus musculus, Oryza sativa, Zea mays, and Saccharomyces cerevisiae. The corresponding protein accession numbers are provided in the Materials and Methods section, specifically accession numbers L742-L744. Additionally, the alignment of the protein sequences is presented in Figure S4.
5. For true homolog of RNA polymerase binding protein II, one would expect interaction/colocalization of the two proteins. Could the authors provide colocalization analyses, BiFC, or any other evidence that the proteins are localized in the same compartment?	That's a very thoughtful suggestion. The homologs of AATF/Che-1 in animals have a lot of interacting proteins such as Rb (Retinoblastoma) etc. And hRPB11 (A core subunit of RNA polymerase II is one of interacting protein that functions in transcriptional regulation (Fanciulli, M., Bruno, T., Di Padova, M., De Angelis, R., Iezzi, S., Iacobini, C., Floridi, A., & Passananti, C. (2000). Identification of a novel partner of RNA polymerase II subunit 11, Che-1, which interacts with and affects the growth suppression function of Rb. FASEB journal : official publication of the Federation of American Societies for Experimental Biology, 14(7), 904 - 912). However, unfortunately, at this time, we are unable to test whether FAN interacts with RNA polymerase.
6. In Figure 3J, the localization of FAN-GFP and DAPI doesn't show the localization of FAN-GFP in the cell	The subcellular localization of FAN-GFP presents a significant challenge for us. Initially, it was straightforward to observe

nucleus. In fact, it looks like the image is somehow manipulated (different positions of blue objects in DAPI and merged images). I am not convinced that the protein is nucleolar and I suggest performing additional experiments with nucleolar markers (e.g. crosses with fibrillarin marker lines or simple immunolabeling with anti-GFP and anti-fibrillarin antibodies).	FAN-GFP localizes in nucleus based on confocal images. Subsequently, we speculated that FAN-GFP locates in nucleolus because of the shapes of FAN-GFP on the image were solid circulars, which are different from looped structures observed through DAPI staining and H2B-tdTomato, predicted NoLS showed in figure 2B, and the nucleolus localization of homolog AATF/Che-1. As we mentioned in the text, the reference indicate that H2B can be a nucleoplasm marker and we do not have a fine nucleolus marker at that time. We used H2B-tdTomato to show the nucleolus position indirectly. Our results indicate that the peak values of FAN-GFP, H2B-tdtomato and DAPI are not at the same place and a small quantity of FAN-GFP was detected at the place where DAPI and H2B-tdTomato signals were at the peak. However, we acknowledge that without a definitive nucleolar marker such as fibrillarin, our conclusions remain tentative. And we agree to moderate our claim at this part. To clarify that FAN-GFP is showed in nucleus and rather than emphasizing its potential presence in the nucleolus, we have revised the title on L250 as 'FAN is ubiquitously expressed and highly expressed in nucleus'. Also in L287 to L288: 'Collectively, these observations demonstrate that FAN is predominantly localized in the nucleus of dividing cells'.
7. The whole discussion should be improved. Is there any predicted function of FAN in Arabidopsis?	We rewrite the discussion. And there is no annotation to a function of FAN in TAIR Gene Ontology because of no available literature.

8. Line 34 of the abstract At5G61330 should be changed either to AT5G61330 or At5g61330	The small letter 't' was changed to 'T' in L40: At5G61330 to AT5G61330.
9. Abbreviations that needs explanation: AP2/PLT transcription factors, GRAS transcription factors; MAIN and MAIN Like family members; AtMMS21; WEE1;	Thank you for your reminder. We add the full name of AP2/PLT transcription factors, GRAS transcription factors, MAIN, MAIN like family and AtMMS21 in L65-76. AP2/PLT: AP2 is short of APETALA2 transcription factors and PLT is short of PLETHORA. GRAS transcription factors: GRAS family was characterized by the three TFs GAI (gibberellic acid insensitive), RGA (repressor of GA1-3 mutant), and SCR (scarecrow). MAIN is short of gene name 'MAINTENANCE OF MERISTEMS' . And MIAN Like family has three homologs (MAIN-related genes MAIN-LIKE1 (MAIL1, AT2G25010), MAIN-LIKE2 (MAIL2, AT2G04865) and MAIN LIKE3 (MAIL3, AT1G48120). AtMMS21 is for methyl methanesulfonate sensitivity gene21. As for WEE1, it takes time to find the original name from research articles and there is an evidence that WEE1 derived its gene name for the 'wee' phenotype in fission yeast (Russell, P., & Nurse, P. (1987). Negative regulation of mitosis by wee1+, a gene encoding a protein kinase homolog. Cell, 49(4), 559 - 567). Therefore, the gene name WEE is not an abbreviation and we did not include a full name following it.

10. Line 50 this sentence is hard to follow „ In addition to these important transcription factors, genome integrity in stem cells and their progeny cells is required for meristem maintenance. “	L71-L73: “ In addition to these important transcription factors, genome integrity in stem cells and their progeny cells is required for meristem maintenance. Moreover, maintenance of meristem needs genome integrity.”is deleted. And “Besides, meristem maintenance requires genome integrity” was added in L72-L73.
11. Line 108 “atr-2;fan double mutant “	We are currently uncertain about the specific modifications needed regarding the 'atr-2;fan double mutant'.
12. In Figure NetPhos 3.1 prediction of the phosphorylation sites – this database is available only for a limited number of animal kinases. This should be carefully described in the text.	L1053-L1054: we changed the description of Blue letters from “Blue letters and numbers indicate the confirmed and one of the potential phosphorylation sites on serine and threonine residues in hAATF and FAN respectively” to “Blue letters and numbers indicate the potential phosphorylation sites on serine and threonine residues in hAATF and FAN respectively”..
13. Line 186 - „To analyze the auxin distribution and levels in fan, we followed the auxin marker DR5rev:GFP “ – the DR5 is used as a proxy for auxin accumulation, please rephrase	L296-L299: We made changes in this sentences “To analyze the auxin distribution and levels in fan, we followed the auxin marker DR5rev:GFP. The auxin response maximum in QC and the distribution in columella cells in fan did not differ from wild-type (Figure 4A)” to “To analyze auxin response in fan, we examined DR5rev:GFP reporter. The auxin response maxima in the quiescent center (QC) and the distribution in columella cells in fan mutant were comparable to those in Col-0 (Figure 4A)”.
14. Line 188 - „To follow the expression changes of the root master regulators, PLTs, “ – please rephrase to changes in protein	L308: We has revised “To follow the expression changes of the root master regulators, PLTs, we crossed the PLT1_{pro}:CFP and PLT2_{pro}:CFP marker lines

localization/stability or something similar	into the fan mutant background” to “We further assessed the transcriptional changes of root master regulators by crossing the PLT1_{pro}:CFP and PLT2_{pro}:CFP marker lines into the fan mutant background.”
15. Line 221 ,, Based on these observations we concluded that FAN protects genome integrity and is involved in the DNA damage response upon replication stress and DNA cross-linking. “ - simple cell death assay is only indirect evidence of increased DNA damage	Thank you for your reminder. However, in addition to the cell death indicated by PI staining, we also assessed DNA damage levels using comet assay, immunofluorescence staining, and qPCR. These results suggest that DNA damage was indeed present.

Figure 1. Primary root growth and stem cell niche (SCN) maintenance are inhibited in the *fan* mutant.

Figure 1. Primary root growth and stem cell niche (SCN) maintenance are inhibited in the *fan* mutant.

(A) Growth habit of the *fan* mutant compared to the control, Col-0 at 6 DAG under normal growth conditions. Scale bar = 1 cm. (B) Representative images of differential interference contrast (DIC) microscopy showing the root meristem of Col-0 and *fan* (4 and 6 DAG). Scale bar = 50 μ m; arrowheads point to QC (White) and first elongated cortical cell position (Green). (C) Meristem length was measured from the QC to the first elongating cortex cell and (D) the number of meristem cells in the cortex of Col-0 and *fan* at 4 and 6 DAG was counted. The values and error bars in (C) and (D) represent means and \pm SD, $n > 20$. Double asterisks indicate highly significant differences ($P < 0.01$) between *fan* and Col-0 analyzed by two-tailed Student's t tests. (E) The graph shows the root growth of Col-0 and *fan* between 0-8 days after germination. Data represent the mean with \pm SD; $n = 30$. (F) to (H) Representative confocal images of propidium iodide (PI)-stained Col-0 and *fan* mutant root tips (F) PI only, (G) showing expression of *WOX5_{pro}:GFP* while (H) of columella stem cell marker *J2341*. Scale bars = 50 μ m, arrowheads point to QC. (I) to (K) Col-0 and *fan* roots expressing *CO2_{pro}:H2B-YFP* (I), *CO3_{pro}:H2B-YFP* (J), and *SCR_{pro}:GFP* (K). The ratios indicate the number of seedlings showing abnormal expression patterns compared to the total number of seedlings examined. White arrowheads indicate to QC and green arrowheads point to abnormal cell division. Scale bars = 50 μ m. Magenta, PI staining; yellow, YFP and green, GFP.

Figure 2. *FAN* encodes AATF/CHE-1 in Arabidopsis.

Figure 2. *FAN* encodes AATF/CHE-1 in Arabidopsis.

(A) Sequence comparison of the wild type (*FAN*) and mutant (*fan*) DNA, CDS and predicted proteins. The red frame shows the last base of the eighth intron, in which G is mutated to A by EMS mutagenesis. The green frame and black frame show the missing sequence ‘AACGTTAG’ and the formation of stop codon by mis-splicing. The green arrow indicates the last amino acid of predicted *fan* mutant protein. The CDS sequences were obtained by reverse transcription-polymerase chain reaction (RT-PCR) using total RNA isolated from mutant and wild-type seedlings. (B) Comparison of the conserved protein boxes between the human AATF/Che-1 (hAATF) and *FAN*. Light blue box: AATF/Che-1 domain, lila box: TRAUB, orange box: Nuclear localization signal (NLS), yellow box: Nucleolar localization sequence (NoLS). Red arrowhead: position of the last amino acid of *fan*. Blue letters and numbers indicate the potential phosphorylation sites on serine and threonine residues in hAATF and *FAN* respectively.

Figure 3. FAN is widely expressed and localized in nucleus.

Figure 3. FAN is widely expressed and localized in nucleus.

(A) to (G) represent GUS expressional studies in (A) young seedling, (B) root, (C) young shoot, (D) mature rosette leaf, (E) enlarged image, (F) flowers and (G) silique. The white arrowhead in (A) points to the root apical meristem. (H) The representative confocal image illustrates the expression of FAN in the root meristem followed in the *FAN_{pro}:FAN_g-GFP* line (6 DAG). The enlarged section highlights its reduced level of expression in the QC compared

to the surrounding stem cells. Magenta: PI staining; green, GFP fluorescence. Arrowheads (White) point to QC and the first elongated cortex cell (Green). Scale bars =1000 μm in (A), (C), (D-G), scale bars = 50 μm in (B) and (H). (I) Confocal images of the nuclear localization of the FAN-GFP in the $FAN_{pro}:FAN_g-GFP$ line transformed with the $35s:H2B-tdTomato$ marker. Magenta: H2B-tdTomato; green, FAN-GFP. Scale bar = 5 μm . (J) DAPI-stained nuclei of the $FAN_{pro}:FAN_g-GFP$ line. Blue: DAPI; green, GFP. Scale bar = 5 μm . (K) and (L) Grey value distribution of the total number of pixels at the position of the white lines marked in (I) and (J).

Figure 4. Auxin signaling and expression pattern of genes related to root development are altered in *fan*.

Figure 4. Auxin signaling and expression patterns of root developmental regulator genes are altered in *fan*.

(A) represents the expression of *DR5_{rev}:GFP*. (B) of *PLT1_{pro}:CFP* and (C) *PLT2_{pro}:CFP* in the root tips of 5 DAG Col-0 and *fan*. (D) Expression pattern of *SCR_{pro}:GFP* and (E) *SHR_{pro}:SHR-GFP* in the root tips of 5 DAG Col-0 and *fan*. (A-E) In each image, scale bar = 50 μ m; arrowhead points to QC, Magenta, PI staining; Green, GFP, Cyan, CFP. (F) Relative mRNA levels of *PLT1* and *PLT2* quantified by qRT-PCR in Col-0 and *fan*. (G) *SCR* and *SHR* relative mRNA levels determined by qRT-PCR in Col-0 and *fan*. For the analysis, both in (F and G), root tips (2 mm long) were collected, values and error bars represent means and \pm SD from three independent biological replicates. Asterisks indicate significant differences compared with Col-0. (***P* value < 0.01, **P* value < 0.05, two-tailed Student's t test).

Figure 5. *fan* mutants develop DNA damage response

Figure 5. *fan* mutants develop DNA damage response

(A) Confocal images of PI-stained root tips of Col-0 and *fan* under normal growth conditions, treated with 2 mM HU treatment, 10 μ M MMC and 20 μ M zeocin. Seedlings (4 DAG) were transferred to 1/2 MS medium with or without DNA damaging agents. Scale bars = 50 μ m; arrowheads point to QC position. (B) Representative comet assay image of nuclei from Col-0 and *fan* (7 DAG). (C) Levels of DNA fragments measured by the percentage of DNA in the tail of comets in the comet assay. The values and error bars represent means and \pm SD, from three independent experiments. (D) γ H2AX accumulation in the root tips of 5 DAG Col-0 on 1/2 MS medium containing 10 μ M zeocin, Col-0 and *fan* grown under normal conditions measured by immunofluorescence staining. Blue, DAPI staining; green, γ H2AX. Scale bar = 5 μ m. (E) Quantification of γ H2AX foci in Col-0 treated with 10 μ M zeocin, Col-0 and *fan*. At least 100 nuclei were analyzed and grouped into 6 classes according to the number of γ H2AX foci/nuclei. The data are from three independent experiments. Error bars indicate the SD. (F-H) illustrate relative expression level of DNA damage response genes quantified by qRT-PCR. Seedlings, 4DAG, were grown in normal growth conditions and treated for 24 h with the relevant genotoxic agents. 2 mm Root tips (2mm long) were collected. The values and error bars in (F-H) represent means and \pm SD from three independent biological replicates. Columns with different letters indicate significantly differences at $P < 0.05$ (Duncan's multiple range means comparisons).

Figure 6 Cell death response in *fan* is independent on SOG1.

Figure 6 Cell death response in the *fan* mutant is enhanced in the lack of SOG1 function.

(A) Confocal images of PI-stained root tips of Col-0, *fan*, *sog1-1* and *sog1-1;fan* double mutants. Scale bars = 50 μm ; arrowheads point to QC. (B) Measurement of dead cell area and (C) percentage of roots showing dead cells. The values and error bars in (B) and (C) represent means and \pm SD of three independent experiments with at least 10 seedlings each.

Columns with different letters indicate significant differences at $P < 0.05$ (Duncan's multiple range means comparisons).

Figure 7. FAN participates in the ATR-induced DNA damage response.

Figure 7. FAN participates in the ATR-induced DNA damage response.

(A) and (B) Representative images of root meristems of 6 DAG Col-0, *fan*, *atr-2* and *atr-2;fan*. Seedlings grown on 1/2 MS medium with or without 1mM HU. Scale bars = 50 μ m; arrowheads point to QC. (C) Measurement of root meristem length and (D) root meristem cell number of 6 DAG Col-0, *fan*, *atr-2* and *atr-2;fan*. Seedlings grown on 1/2 MS medium with or without 1 mM HU. (E) and (F) Confocal images of PI-stained root tips of Col-0, *fan*, *atr-2* and *atr-2;fan* double mutants. 4 DAG seedlings were transferred and grown on 1/2 MS medium with or without 2 mM HU for 24 h. Scale bars = 50 μ m; arrowheads point to QC. (G) Measurement of the percentage of roots showing dead cells. (H) Measurement of dead cell area from Col-0, *fan*, *atr-2* and *atr-2;fan* PI-stained roots. The values and error bars in (C)-(D) and (G)-(H) represent means and \pm SD from three independent experiments with at least 10 seedlings each. Columns with different letters indicate significant differences at $P < 0.05$ (Duncan's multiple range mean comparisons).

Table S1. Primers used for genotyping

Primer names	Sequence (from 5' to 3')
atr2 : (SALK_032841)	
SALK_032841_LP	GCAGCAAAAATTTCTTGGTTG
SALK_032841_RP	ACTTCAAGGGTTCCGATGTTC
sog1-1	
sog1 Forward	CTATGTTGTTGTTGTAAACACAA
sog1 Reverse	CCAAATGGTATTGGTGCATCACCCAGTTGG
fan	
fan mutant F	TAAGCTATGCTCTTCTCCTCTTC
fan mutant R	CCTCTGGATTTGGTTCCAT

Table S2. Primers used for mapping

Markers	Name	Sequence from 5' to 3'
MSN2	457122_F1:	TAAATGGGCGAGACAACATCACAG
	457122_F2:	CTATAGTGGGACTGAAAAGCCC
	457122_R1:	GGAAGTTGGTGAAAGAGCGGC
	457122_R2:	GTAGCAAACATCATCGCGTTGGGG
MUB3	457274_F1:	GGGCATAGAATAAGAGCATATTCC
	457274_F2:	GAATAAGGCAGTGGCAACCATTG
	457274_R1:	GAGAATACAATATTCGTGAGAG
	457274_R2:	CCATCACTCAAATTTGAACGTTGC
K11J9	454021_F1:	CGTTTTACGAGCATGGTCTTGGC
	454021_F2:	CATGGTCTTGGCTAACATCCTCC
	454021_R1:	CCAAACTCCTCGTGTTTGGCTG
	454021_R2:	TGGCTGACCGGTTATGATGAGG
F15L12	449419_F1:	CCTGACTATCCTCACTTCTGAGG
	449419_F2:	TTGAGGTTTCATGTTTCTATGACG
	449419_R1:	GCCAAAGGGTAAGTGGGTGGGTGA
	449419_R2:	CAGAGACAGAGACGGTGAGTTAGAC
MTE17	457148_F1:	GGGTTACGTGAACATGACAC
	457148_F2:	GATCGATCGATAGGAAGTGTTGG
	457148_R1:	CTGAAGCTGTGTTGCTCGTTGAG
	457148_R2:	GCGGGGTGATGGAGAGATTCAC
MTH12	MTH12_F:	CGGCATCTGTTCATGCATTATA
	MTH12_R:	CTCAAGGCTAAGTAGTGTATGA

MMN10	MMN10_F:	CCCCATTGCCCCGCGGTAATAAGC
	MMN10_R:	CGAACCATCACCCTGGTGAGTG
F15L12	F15L12_F:	AGTAGGAATTGGAATGAGCAA
	F15L12_R:	TGTCATGTGAGGACAAGTCTGAA
MUP24	MUP24_F:	GTAGAATCAGAAATACCATAATC
	MUP24_R:	GGTGTCCAATCAAGTTTTTCGGTT
MAE1	MAE1_F:	GAACTGGACTATAATATAAATT
	MAE1_R:	GACTTGGGTCAGAGGACAAACG
MSL3	MSL3_F:	CTTACGTTTGTGAACACATATTA
	MSL3_R:	GACAAAATTGCAAACGCGGAGAT
MFB13	MFB13_F:	GACGACTGATTACATAACATAGT
	MFB13_R1:	GTGTGTATTGTCTATATATATTACG
	MFB13_R2:	AGCTGTCATGCGTGATGCTTG
MAC9	MAC9_F:	CACATGAGACTTGAGTGTTGTTC
	MAC9_R:	CGGATTTTCATACGAACTTGCTAA
K19B1	K19B1_F:	GGTTATCGAATATATAAAAATGTG
	K19B1_R:	GTCAATGTTGGGAATTTGCAGTCC
MQB2	MQB2_F:	GGCGACTACTAGCATAAAAAATA
	MQB2_R:	GATCTTGCCATTTATTTGGTCAA
MBK5	MBK5_F:	GGCCCATCTAGAGTATAACCATG
	MBK5_R:	CAGCTACTGCGTGCAAATAAAGAT
MGI19	MGI19_F:	CTAGAGAGACAAGATAAGACACC
	MGI19_R:	GTTATCGCCAACTTGACCCTTA

MUB3	MUB3_F:	GCCTGGCTGATATTATGAACTTTC
	MUB3_R:	GGCTATTATCACTTCCGAAGAGGTT
MUB3	457274_F3:	GCATTGAATAAGGCAGTGGCAACC
	457274_R3:	CGTGTGGCAAATCCAATAGTTAG
MAF19	MAF19_a_F:	CTCAACCAATCAAAGGCGGACACC
	MAF19_a_R:	TATTAAACGATAAATTCGCCGTTTGC
MAF19	MAF19_b_F:	GTCACTGTTGTCTTTCTAGAAACAGAG
	MAF19_b_R:	CTCTATCTCTCTCTGTGTCTCTCCA
10A10	10A10_F:	GGTCACAGGGATCAAGATGTGG
	10A10_R:	GCCTTATGGATTTTCTGGAGAAAG
EG7F2	EG7F2_F:	GCATAGAATTTGACGATAACGAGC
	EG7F2_R:	GATCTGTGTAGGACTACGAGAC

Table S3. The sequence information for the primers used for mapping

Name	Sequence from 5' to 3'
210_F1:	GGGTCGTTCTGAGTCGTCTCC
210_R1:	GTCTCTCAGGATAATATCAC
210_F2:	GTAGTAACCAATCCAAGTGTTC
210_R2:	CTCACACCAACTAATGTCCTCAG
210_F3:	GCCGTTGAAATCGACCATGATC
210_R3:	CTCCACTCTCTGATAAGCATC
210_F4:	GGGACTGAACTCAGCACCCAGAG
210_R4:	ACTGGAAAGATTCTTATCATG
200_F1:	GGTAATGGAGAGATTCTCATG
200_R1:	CTGGTTTATACAGCAGGAAGAGGAAC
190_F1:	TCGACGAGATAGTGAGAGGAGTA
190_R1:	ATTCCCCTGTGGCTCCACTGC
190_F2:	CCAGAGAAAAGCATGTTCCAAGAGG
190_R2:	CACTTGCATTACGCAGTGCAGTCA
190_F3:	GATATTATAGCTTTGGATTAGGTAC
190_R3:	GACAAAGTTTAGATGCAGCTGGTC
190_F4:	GGAGAAACTAAGCATTTCATGGGAGG
190_R4:	GATCGGTTTCTTCTCGGATTATCCTTG
190_F5:	CGAGGGGCACCAGAAGATAAAGTGG
190_R5:	CATCCATTGTTGACAACTTAAACATG

180_F1: ACAATCTCTAAACCCTGACCTCCC
180_R1: CGATTAGGGGGTCCTGCAGATGCGC

170_F1: AGTATCGCCGACGCCGCAGCACA
170_R1: GGAATTCTACTACTGGTTTTTCATC

160_F1: TTAACTCCATCTGATACTCAGCTG
160_R1: GGGTCACCACTACGTGCCTTCACCATG

160_F2: CACGCGCTGAGCTTCACTTGTCAA
160_R2: AGGGGGAAATAGATTCCTGATCATG

150_F1: CGTCGGAGCTCAGATCACGAGCCG
150_F1: GTTCATCCTGCTCAACGTCATTCCGAAC

150_F2: GAGGTTCAAGTTGCACAGTCAGACG
150_R2: GCATCAGAAGTACTAGTAATAGGC

150_F3: ACATATGTAAGTGAACACTTCTTC
150_R3: CATCAAGAGCATCTTCCAAGTAGCC

150_F4: ACAAAGCTGAGTCAGGCGAGGGAG
150_R4: AACGAGTCTAATTCCAGACTACAT

Table S4. Primers for vector construction

Primer names	Sequence (from 5' to 3')
Promoter reporter analysis with GUS	
Forw: 164-6:	GGGGCGATCGGCCGCCGCGGAGACAGAGATATCATATC CATAGCTC
Rev: 164-7:	GGGGGGCGCGCCAAATCAGGTAATTGATCATCTTCATT GTCGCT
Complementation analysis of mutant plant lines	
Forw: 164-35:	GGGGGCCTGCAGGGGAGACAGAGATATCATATCCATAG CTC
Rev: 164-11:	GGGGGAGCTCGTGAAAGAGACGCAGAAGATGTGAAAC C
Subcellular localization	
Forw: 164-35:	GGGGGCCTGCAGGGGAGACAGAGATATCATATCCATAG CTC
Rev: 164-12:	GGGGGGCGCGCCAAGCTTCAGACTGAACGTTTCTGGTCT TG
Overexpression of CDS	
Forw: 164-15:	GGGGACTAGTATGGCTGGGGGGTCAAAGAGGTCT
Rev: 164-14:	GGGGGAGCTCTTAAAGATCCTCCTCAGAAATCAACTTTT GCTC AGCTTCAGACTGAACGTTTCTGGTCTTG

Table S5. Primers for qRT-PCR

Gene names	Forward Primers (from 5' to 3')	Revers Primers (from 5' to 3')
PLT1	TTGCAGCAACAGTCGAGCCAGA	TCGGTCGATCCAACAGTCGAGC
PLT2	GGCTGAGGAAGAGTTTCCAGCCG	TCCATACCCTACCTTGCCTCGC
BRCA1	TCATGGGAGATTTTCGAGCTT	ATTTAGCCAAGGCTTCAGCA
RAD51	TTGCTGGTCCCAATTTAAG	CAAACATGGCGAGCTTATCA
SCR	AGCAACAACCGTGGTCCTCCT	ATACAGTAGCCGCCGCGTAA
SHR	GCGAACGATGCTACCGAACCAT	GCCTCTCCGTCTACTGCTTCCA
PARP2	ATGGCGTTCTGCTCCTCTGC	GGTGCTGTTTTCCCCACACC
FAN	GAAGAATCAGAAGGCTCTTTGGG	CTCAAACAAAGCCTCTTGCAAC
UBQ10	CCTGCGTCTTCGTGGTGGTT	GTCGAGTCACTTTGCAGGCGT

Figure S1. Abnormal organisation of stem cell niche (SCN) in the *fan* mutant.

Differential Interference Contrast (DIC) images of Lugol stained roots Col-0 and *fan* (6 DAG) showing the expression of *QC25:GUS* in. Scale bar = 50 μ m, arrowheads point to QC.

Figure S2. *FAN* genomic DNA fragment can complement the *fan* mutation.

Figure S2. *FAN* genomic DNA fragment can complement the *fan* phenotype.

(A) Complementation of the *fan* mutation with the *FAN* genomic DNA region expressed under its own promoter. Root growth phenotype of Col-0, *fan* and the complementation lines *FAN_{pro}-FAN_g-FAN_{3-UTR}* (FAN C1 and FAN C2, 6 DAG). Scale bar = 1 cm. (B) Primary root length (6 DAG, cm). (C) Root meristem of Col-0, *fan* and complementation lines. Scale bar = 50 μm; white arrowheads point to QC, while the green one to the first elongated cortical cell. (D) Number of the amplifying cell in the cortex cell line and (E) Root meristem length. The value and error bars in (B), (D) and (E) represent means and ±SD, n>20 in each. Columns with

different letters indicate significant differences at $P < 0.05$ (Duncan's multiple range means comparisons).

Figure S3. FAN is the plant homolog of AATF/Che-1.

Phylogenetic analysis of AATF/Che-1 from different species. Mm, *Mus musculus*; Hs, *Homo sapiens*; Os, *Oryza sativa*; Zm, *Zea mays*; At, *Arabidopsis thaliana*; Sc, *Sacharomyces cerevisiae*. NCBI blast, Mega 5.05, align by clustal W, construct/test neighbor joining tree. Bootstrap test using 1000 repetitions.

Figure S4 The alignment of the AATF/Che-1 protein sequences

Saccharomyces	--meksladqisdi-aikpvnkdfdi--edeenasifqhnekngesdlsdygnsntteetk	55
Homo	magpqp1alqlleqlnprpseadpeadpeeataarvidrf-dege-----dg	46
Mus	maapqp1alqlleqlnprpseadpeadpeeatrarvidrf-dege-----ee	46
Arabidopsis	-----	0
Glycine	-----	0
Oryza	-----	0
Zea	-----	0
Saccharomyces	kahyleveksklraekglelndpkytgvkgsrqalyee---ase-----	96
Homo	egdf1lvvgsirk1lasas1ldtdkrycgkttstrkawnedhweqtlpgssdeei-----	98
Mus	kdd-lavssirk1lapvs1ldtdkrysgkttstrkawkedhweqalpsssdnea-----	97
Arabidopsis	-mrlsialkvvmaggs-----krskrarlds---esedisqenlkaesdne	44
Glycine	-----m---glsa	5
Oryza	-----mapgt-----tlapkrkkae---aspsp-spspm---gds	29
Zea	-----	0
Saccharomyces	-----nedee-----eeeeeeeeegeeeeeg-----	123
Homo	--sdeegsgdedseglgleeyd--eddlgaeeqecgdhreskkrsrshsaktpgfsvqsi	154
Mus	--sdeggsedgdseglgleeis--edvdedlednkisde-----	132
Arabidopsis	ddqlpdgieddevdsmmeddegeseeddegdteeddegdseed-----	86
Glycine	kksrkrkgkrdsdsdeydnmeye-----evddyd-----	34
Oryza	d---ggysdsdlhdaeesfysar---sgseddrqvsssndd-----	64
Zea	-----	0
Saccharomyces	-----eeeked-alsfrtdsedeeveideeesdadggeteeaaqqkrh	165
Homo	sdfekftkgmddlgseeeedeesgmee--gdda-edsggese--edrag-----	199
Mus	-----ggsedgdseglgleefsedve-edlegede--edree-----	166
Arabidopsis	-----degenkedgedesedfedgndkesesgdegnddnkdaqme---elek	130
Glycine	-----ddgeeed-e---eehgeevtddedgtgehgewkndeme---qlek	73
Oryza	-----ddseeeqeeremdeeedeeddddeemneeddedegeemn---elek	108
Zea	-----	0

Continue of Figure S4

Saccharomyces	alskliqgetkqainklsqsvqrdsakgysilqgqtklfidniidlrilqlgkaviaanklpl	225
Homo	---dnrseddgvvmtfssvkvseevkgravknqialwdqlllegrikqlgkallittnqlpq	256
Mus	---dnrseddgvvaaafssvkvseevkgravknqialwdqlllegrikqlgkallittnqlpq	223
Arabidopsis	evkelsrqeqd- ilknlkrdkgedavkqgavknqkaldkilefrflkqkafdrsrlpq	189
Glycine	eyrdlhhqeld-tlknkhhkdedllkqgavksqkalwykilelrlfllkqkpfsssnrlpq	132
Oryza	eyrtlqtnqqn-iletlkqhrdddvskggavknqkvlwdkalemrfllkqkafstsnklpk	167
Zea	-----mmnehreedalrggavknqkaiwdktlemrfllkqkvfstsanklpq	45
	. : :* :: .* :: : : * : *** . :* : **	
Saccharomyces	tteswee-----akmddseetkrrllkeneklfnnlfnrlnifrikfqlgdhitqn	275
Homo	pdvfpfkdkggpefssalknshkalkallrslvlgqeellfgydpdtrylvdgtpknags	316
Mus	pdvfpvfkdkggpefasalknshkalkallrslvdlqeellfgydpdtrhivngakpntes	283
Arabidopsis	epvkslfcsede-dvstaytldlvtsskktltdldelelqeaifeknpsvdqgvnatase---	245
Glycine	esikslfcetde-tvrvaysdlmtsskcttdldelelqeaifeknpsitqaivgsegsskd	191
Oryza	epirsmfcdhnq-eieqayldllnsskqtlgsmmelqeaalernratkdvtdtd-----	220
Zea	esirtrfcihdk-qieqayddllnsskhtlssmmelqeaallesnqatkaneii-----	97
	* : * * * * * : *	
Saccharomyces	eevakh-----k--lskkrslkelygetnsldselkeyrtavlnkwstkvss	320
Homo	eeisseddelveekkkqrrrvpakrklemedypsfmakrfadftvyrnrtlqkwhdktkl	376
Mus	eeisseddelvgekkgqr--kappkrklemedypsfmakrfadftiyrnhtlqkwhdktkl	342
Arabidopsis	--snksd-----aedsewqrisdlqkrmsvfrnkavdkwqrktqv	285
Glycine	levykhld -----gnldqewsqisqmhsitsfrdksinkwqrvtqv	233
Oryza	--nsse-l-----ngeddewsevqlqktritpfrnseidkwqrktqv	259
Zea	--psa-s-----ngdndewsevqlrarittfrnteidkwqrkiqv	135
	. : . : .* : ** .	
Saccharomyces	asgnaalssnkfkainlpadvqvenqlsdmsrlmkrtklnrrnitplyfqkdcangrlpe	380
Homo	asgk--lg-kgfgafersiltqidhilmdkerllrrtqtkrsvyrvl-----gkpep	425
Mus	asgk--lg-kgfgafersiltqidhimdkerllrrtqtkrsayrvl-----gkpep	391
Arabidopsis	ttgaaaik-gklhafnqnvsqvasymrdpsrmikqmqqrstvfvf-----gtvpq	336
Glycine	ttgaaaik-gklhafnqdisnqvaaymrdpsrlkqmvrrsdvnif-----lsvpe	284
Oryza	ttgaaaik-gklhafnqnisdqvtsymrdpsrminrmhlrkstlgvf-----gee--	308
Zea	ttgaaaik-gklhafnqnisdqvagymrdpsrminrmyltnsavrvf-----gkd--	184
	: : * : * : * . * : : . :	
Saccharomyces	lispvvksvddnensddglidipknydprkdnnaiditenpyvfddedfyrvllndlid	440
Homo	aaqpvpeslpgepeilpq-----apanahlkdldeeifdddgyfhyqlirelie	473
Mus	vpepvaetlpgepetlpq-----gpanahlrldideefdddgyfhyqlirelie	439
Arabidopsis	e-----amepn-----peekqeeqdpelvedaefyqllkefle	370
Glycine	v-----vgepk-----eaetctdgdpellddsefyqllkeffe	318
Oryza	-----vgehen-----nkeenntegdpelvddeefyqllkefle	343
Zea	-----vgepgt-----aeeghivegdpeliddsefyqllkefle	219
	: : : : * : * * : * : ** : * : * : *	
Saccharomyces	kkisnahnsesaaait-itstnarsnnklknidtkaskgrklnysvqdpianyeapitsg	499
Homo	rktssldpndqvamgrqwlaigqlrskirkkvdrraskgrklrfhvlskllsfmapidht	533
Mus	rktssldpndqvamgrqwlaigqlrskirkkvdrraskgrklrfhvlskllsfmapidht	499
Arabidopsis	tid ----- passeeafyemkkfgtqkrkvdrraskrkiyvnvhekivnfmappak	423
Glycine	tvd-----psseeafyalkrmpkkrkivdrraskrkiyvnvhekivnfmappan	371
Oryza	scd-----agasesafyalkkqhkkrklvdrraskrkiyvhvhekiannfmappmv	396
Zea	scd-----rgasesafyslkqkvkkrklvdrpaskrkiyvhvhekintfmappmv	272
	* * * * * * * * * * * * * * * * * * * *	
Saccharomyces	ykwsddqiddeffagllgqrvnfenedeeghariendeelavknnddiqifg	551
Homo	tm-nddartelyrslfgqlhppdeghgd-----	560
Mus	am-sddartelfrslfgqlnppdadrgk-----	526
Arabidopsis	ip---pntadllknlfglkrtrnvqsea-----	447
Glycine	vp---pmapklfenlfglkrtrnvqsea-----	396
Oryza	ip---pmapklfenlfgmgngkseta-----	419
Zea	lp---pmapklfenlfgnss-----	289
	. : : * : *	

Figure S4. The alignment of the AATF/Che-1 protein sequences

Alignment of AATF/Che-1 via clustalw2 (<http://www.ebi.ac.uk/Tools/msa/clustalw2/>). Black line indicates AATF domain and green line indicates TRAUB domain of Arabidopsis.

Figure S5. Expression levels of *PLT1*, *PLT2*, *SHR* and *SCR* are reduced in *fan* compared to Col-0.

Figure S5. Expression levels of *PLT1*, *PLT2*, *SHR* and *SCR* are reduced in *fan* compared to Col-0.

Mean gray value of *PLT1_{pro}:CFP* (A), *PLT2_{pro}:CFP* (B) *SHR_{pro}:SHR-GFP* (C) and *SCR_{pro}:GFP* (D) in Col-0 and *fan*. The values and error bars in (A) - (D) represent means and \pm SD. Asterisks indicate significant differences compared to Col-0. (***P* value < 0.01, two-tailed Student's *t* test, *n*>10).

Figure S6. Response to DNA damage treats in *fan* compared to Col-0.

Figure S6. Response to DNA damage treats in *fan* compared to Col-0..

(A) The proportion of roots with cell death and (B) the mean area of dead cells after treatment compared to the control. Data in (A) and (B) are the means and \pm SD from at least three independent experiments with at least 10 seedlings each. Columns with different letters indicate significant differences, $P < 0.05$ (Duncan's multiple range means comparisons).

Figure S7. Root meristem phenotype of *FAN* overexpression lines.

Figure S7. Root meristem phenotype of *FAN* overexpression lines.

(A) Representative images of the root meristem of Col-0, *FAN* overexpression lines *FAN*^{OE2}, and *FAN*^{OE3} (6 DAG). Scale bar = 50 µm; white arrowheads point to QC, while the green one to the first elongated cortex cell. (B) Relative mRNA levels of *FAN* in Col-0, *FAN*^{OE2} and *FAN*^{OE3} (5 DAG). The values and error bars represent means and ±SD. Data from three independent biological replicates. (C) Quantification of root meristem length of Col-0, *FAN*^{OE2} and *FAN*^{OE3} (6 DAG). (D) Quantification of number of root meristem cells of Col-0, *FAN*^{OE2}, and *FAN*^{OE3} (6 DAG). The value and error bars in (C) and (D) represent means and

SD. Asterisks indicate significant difference compared to Col-0 (**P value < 0.01 two-tailed Student's t test, n>10).

Figure S8. *FAN* overexpression results in reduced level of cell death level upon HU treatment.

Figure S8. Decreased cell death level of *FAN* overexpression lines under HU treatment.

(A) and (B) PI-stained root tips of Col-0, *FAN^{OE2}* and *FAN^{OE3}* under control conditions (A) and 2 mM HU treatment (B). 4 DAG seedlings were transferred to 1/2 MS medium with or without 2 mM HU and grown for 1 day. Scale bars = 50 μm ; arrowheads point to QC. (C) Ratio of roots (%) with cell death response and (D) dead cell area (μm^2) from PI-stained roots of Col-0 and *FAN* overexpression lines. The values and error bars in (C) and (D) represent means and \pm SD from three independent experiments with at least 10 seedlings each. Columns with different letters are significantly differences at $P < 0.05$ (Duncan's

multiple range means comparisons).

Figure S9. Treatment with genotoxic toxic agents affect *FAN* fusion protein level

Figure S9. *FAN* fusion protein level is effected by DNA damage reagent treatment.

(A) GUS-stained images of *FAN* promoter fused to *GUS* reporter gene (*FAN_{pro}:GUS*) seedlings. 5 DAG seedlings were transferred and grown for 1 day on 1/2 MS medium with either no stress, 2 mM HU, 10 μ M MMC, or 20 μ M zeocin respectively. Scale bar = 50 μ m.

(B) Relative mRNA levels of *FAN* under DNA damage stress treatments were measured by

qRT-PCR. 4 DAG seedlings were transferred to 1/2 MS liquid medium without stress, 2 mM HU, 10 μ M MMC, or 20 μ M zeocin and grown for 24 h. After that, 5 DAG seedlings were collected as samples. The values and error bars represent mean and \pm SD from three independent experiments. Columns with different letters are significantly difference at $P<0.05$ (Duncan's multiple range means comparisons). (C) Confocal images of PI-stained of *FAN_{pro}:FAN_g-GFP* root tips. 5 DAG seedlings were transferred to 1/2 MS plates with no stress, 2 mM HU, 10 μ M MMC or 20 μ M zeocin and grown for 24 h. Magenta, PI staining and green, GFP. Scale bar = 50 μ m; arrowheads point to QC. (D) Gray value of *FAN_{pro}:FAN_g-GFP* under DNA damage treatment. The values and error bars represent mean and \pm SD, $n>10$. Columns with different letters are significantly difference at $P<0.05$ (Duncan's multiple range means comparisons).

Figure S10. Primary root length of the *atr-2* mutant is partly rescued by *fan* in the double mutant upon HU treatment.

Figure S10. Primary root length of *atr-2;fan* double mutant is partially rescued compared to *atr-2* mutant upon HU treatment.

(A) to (D) Primary root phenotype of 6 DAG Col-0 (A), *fan* (B), *atr-2* (C) and *atr-2;fan* (D) seedlings grown on 1/2 MS medium containing 0 (control condition) or 1 mM HU. Scale bars = 1 cm. (E) Measurement of primary root length of Col-0, *fan*, *atr-2* and *atr-2;fan* (6 DAG). The value and error bars represent the means and \pm SD from three independent experiments with at least 10 seedlings each. Columns with different letters are significantly differences at $P < 0.05$ (Duncan's multiple range means comparisons).

Supplement data 1 genomic and CDS sequence of wild type *FAN* and *fan* mutant

FAN (AT5G61330) genomic sequence from ATG to TGA:

ATGGCTGGGGGTCAAAGAGGTCTAAAAGAGCAAGACTTGACAGTGAATCGGAAGACATAAGCGACCAAGAAAACC
TTAAGgtatgtttatgttgcttttgcctcgtttaagttatntagatggtataggtgaatttatgttgagggtt
ctctatattatagGCCGAAAGCGACAATGAAGATGATCAATTACCTGATGGGATAGAGGATGATGAAGTAGATAGC
ATGGAAGATGATGAGGGAGAGAGTGAGGAAGACGATGAAGGAGATACCGAGGAAGATGATGAAGGAGATAGCGAGG
AAGATGATGAAGGAGAGAAACAAGGAAGACGAGGATGGAGAAAGCGAGGACTTTGAAGATGGAAATGATAAAGAGAG
TGAGAGTGGCGATGAAGGTAATGATGACAATAAAGATGCTCAGATGGAAGAGCTTGAGAAAGAGGTCAAGGAGCTT
CGTTCACAAGAGCAgtaagatttcatttgaactaatcatttttaataactgttatttctaagatagtgaattccttg
tatgtgtttccttctagaaacggtagatagtcgtattaaagtttcaactgtttaatctagatcagcaaggatgttg
gctagatttaCaTATGGCTTTCGATAGAAACGGTGTCAATAGTAGTATTACAACACACCACAGAAAAAGGGACGAT
CTATTACAGCGTTTCAAATTTGTGGTTTGTGGATATTAAGCTTCAGACTGAACGTTTCTGGTCTTGAGACCGAAG
AGATTCTTAAGCAAATCCGCCGTGTTTGGAGGAATCTTCGCTGGTCTGGAGCCATAAAGTTGACAATCTTCTCAT
GAACATTATACCTGTTTCGATTTTTTACAAGAAAATGATTCAAAAGAGAGAATTATGGGAAGAGAAAACCTCCAAG
AGGTGAACTATGTGTGTTTTATTTACCTGATCTTCTGCTCTTTGAGGCACGTCGATCCACAACCTTCTCTTCT
TCGCTCGGAACCTTCTTCATCTCGTAGAAAGCCGCCTCTGCCAATGGCAAATATTGAGATTTCAAGAAATATAAAAAAG
ACTCTTTACACTTGAGTTGTTGGACATAACAATCGCTAGGGGGGGTTTCATATCTAATGATTTCTAGGATGTTTTTC
TTTACTATACCTGAGGAAGCTGGATCGATAGTCTCGAGAACTCCTTTAGTAACTGCCATATAGAACTCAGCATCTT
CCACAAGTTCAGGGTCTCCTTCTTCTGTTTTTCTGGACAAGAAAGAGTACACATGTTAGAGAATGGGTATCAAG
ATAATCAATGAGATTGATTAATGTTTATACTCTTTATGGTTAGAGAAAACCGTGATTTTACCTCTGGATTTGGTT
CCATGGCTTCTGATATGGCTTTCGATAGAAACGGTGTCAATAGTAGTATTACAACACACCACAGAAAAAGGGACG
ATCTATTACAGCGTTTCAAATTTGTGGTTTGTGGATATTAAGCTTCAGACTGAACGTTTCTGGTCTTGAGACCGA
AGAGATTTCTTAAGCAAATCCGCCGTGTTTGGAGGAATCTTCGCTGGTCTGGAGCCATAAAGTTGACAATCTTCTC
ATGAACATATACCTGTTTCGATTTTTTACAAGAAAATGATTCAAAAGAGAGAATTATGGGAAGAGAAAACCTCCA
AGAGGTGAACTATGTGTGTTTTATTTACCTGATCTTCTGCTCTTTGAGGCACGTCGATCCACAACCTTCTCTT
CTTCGCTCGGAACCTTCTTCATCTCGTAGAAAGCCGCCTCTGCCAATGGCAAATATTGAGATTTCAAGAAATATAAAAA
AGACTCTTTACACTTGAGTTGTTGGACATAACAATCGCTAGGGGGGGTTTCATATCTAATGATTTCTAGGATGTTTT
TCTTTACTATACCTGAGGAAGCTGGATCGATAGTCTCGAGAACTCCTTTAGTAACTGCCATATAGAACTCAGCATC
TTCCACAAGTTCAGGGTCTCCTTCTTCTGTTTTTCTGGACAAGAAAGAGTACACATGTTAGAGAATGGGTATCA
AGATAATCAATGAGATTGATTAATGTTTATACTCTTTATGGTTAGAGAAAACCGTGATTTTACCTCTGGATTTGG
TTCCATGGCTTCTGAGGGACCTGTGGATGTTTtagaagtaggttcgaactcattatagcaattggatttatcgtg
attagtagaaaatatatacgttttaaaggaaaagaacagacttttgccaaaagttttgtcagatactgtcttaat
tcggttgcatcatttaattttTACAAGAAAATGATTCAAAAGAGAGAATTATGGGAAGAGAAAACCTCCAAGAGG
TGAAACTATGTGTGTTTTaaatgaacgTGTC AATAGTAGTATTACAACACACCACAGAAAAAGGGACGATCTATTA
CAGCGTTTCAAATattgagaggcaaaagaatgttctttgtgtaatatgctgttctggtatagGGATATATTGA
AGAACTTGAAGCGTGATAAGGGTGAAGATGCTGTTAAAGGTCAAGCGGTGAAGAATCAGAAAGgtatgacctgttct
tgcggtttgcaagtagtctaactagaaatcagttgcagttgtatattcctattttatgttaatgaagttgatcta
gagcgaatttttgacaattccttctttgtgctctatgtaaagGCTCTTTGGGATAAGATTCTGGAGTTTCAGATTCT
TACTTCAGAAAGCATTTGATCGTTCAAACAGATTACCACAGgtaagctatgctcttctcctcttctGtgGAAGCAA
CCTGTTCACTAACGTTTtGGTTTTGAAAATTAATTTAAAGTTAGATTCTGATTGAtCCCAGAtAAGAGCattta
tggtgtgtttgttcatcatactttatgacagaaATATATTTTCTACTAATCACGAATAAATCCAATTGCTATAATG
AGTTCGAACCTACTTCTAAATCTAGCCAACATCTTGTCTGATCTAGATTAAACAGTTGAACTTTAATACGACTAT

CTACCGTTTCTAGAAAGAAACACATACAAGAATTCACATCTTAGAAATAACAGTATTTAAAAATGATTAGTTCAA
ATGAAATCTTACTGCTCTTGTGAACGAAGCTCCTTGACCTttTTTACAAGAAAATGATTCAAAGAGAGAATTATG
GGAAGAGAAAACCTCCAAGAGGTGAAACTATGTGTGTTTTATTTACCTGATCTTTCTGCTCTTTGAGGCACGTGCA
TCCACAACCTTTCTCTTCTTCGTCTGGAACCTTTCATCTCGTAGAAAGCCGCTCTGCCAATGGCAAATATTGAG
ATTTTCAGAATATAAAAAAGACTCTTTTACACTTGAGTTGTTGGACATAACAATCGCTAGGGGGGTTTCATATCTAA
TGATTCTAGGATGTTTTTCTTTACTATACCTGAGGAAGCTGGATCGATAGTCTCGAGAACTCCTTTAGTAACTGC
CTATAGAACTCAGCATCTTCCACAAGTTCAGGGTCTCCTTCTTCTTGTTTTTCTGGACAAGAAAGAGTACACATG
TTAGAGAATGGGTATCAAGATAATCAATGAGATTGATTAATGTTTATACTCTTTATGGTTAGAGAAAACCGTGAT
TTTACCTCTGGATTGGTTCCATGGCTTCTGAGGGACCTGTGGATGTTTAAATTTAAAGTCAGGAAAAATTGAAG
AGAAGCTGATCTATTGACCACCAAAGGCCAAATTCCTTAGATGGTTTATGCTTTTTATTATCAAAGCAAAAACCTTACA
GTTCCAAAAACAGCAACAGTAGATCTTGATTGTTGCATCTGTTAATCATTCTACTTGGATCCCTCATGtgGAAGC
AACCTGTTCACTAACGTTTtGGTTTTGAAAATTAATtAAAGTTAGATTCCTGATTGAtCCCAGAtAAGAGCCTT
TCTCAAGCTCTTCCATCTGAGCATCTTTATTGTTCATCTTACCTTCATCGCCACTCTCACTCTCTTTATCATTTCC
ATCTTCAAAGTCTCGCTTCTCCATCCTGACAGATAGAAAAAGAAAGGAgatGtTATATTGACCCTCGCTTATAG
TGATGATAATTATGTTATCTGTTAATATGGCATAACATTCACTTGTGATCAACCGAGGGTCTTCTCAAACAAA
GCCTGTAGAAATCCATATTTAGATAATAATAAAAAGTCAACAAACGTGGCAGAGCGACAAAACACTAAGAGACA
GCATACAATTACTGAATCAAGCCTATGCATACATACCTCTTGCAACTCCAAGAGGAATCTAATGTCTTCTTAGAT
GAAGTAACTAGATCTGTATATGCTGTTGAGACATCCTCATCTTCTGAACAAAATAACGATTTTACAGGCTCCTGAA
AAGGTACCTTCTGTCAAAAGTATGATGAAACAAACAACCATAAATAGAAGAGGAGAAGAGCATAGCTTACCTGT
GGTAATCTGTTTGAACGATCAAATGCTTCTGAAGTAAGAATCTGAACTCCAGAATCTTATCCCAAAGAGCCTTTA
CATAGAGCACAAGAAAGAATTGTCAAAAATTCGCTCTAGATCAACTTCATTAACATAAAAATAGGAATATACAAC
GCAACTGATTTCTAGTTAGACTACTTCGCAAACCGCAGAACAAAGGTCATACCTTCTGATTCTTACCGCTTGACct
tCGTCTTCCTTGTTCTCTCCTTCATCATCTTCTCGTATCTCCTTCATCATCTTCTCGGTATCTCCTTCATCGT
CTTCTCACTCTCTCCCTCATCATCTTCCATGCTATCTACTTCATCATCTCTATCCCATCAGGTAATTGATCATC
TTCATTGTCGCTTTTCGGCTATAATATAGAGAACCCTCAACCAAATAAATTCACCTATACCATCTAAATAACTTAA
AcgaGacaAAAGCAACAtaACATACCTTaagGTTTTCTTggtCGCTTATGTCTTCCgATTCACTGggtaccctttt
cagGAGCCTGTGAAATCGTTATTTTTGTTTCAAGATGAGGATGTCTCAACAGCATATACAGATCTAGTTACTTCAT
CTAAGAAGACATTAGATTCCTCTTGGAGTTGCAAGAGgtatgtagcataggcttgattcagtaattgtagctg
tctcttagtgTTTTgtcgctctgccagtttggtgacttttattattattatctaaatatggatttctacagGCTT
TGTTTGAGAAGAACCCTCGGTTGATCAACAAGTGAATGgtatgccatattaacagataacataattatcatcact
ataagcgagggtcaatataacatctcttttcttttctatctgtcagCAACAGCTAGTGAAGAATCCAATAAATCA
GATGCAGAAGATAGCGATGAATGGCAGCGAATATCTGACTTGCAGAAAAGgtattcaagactaaatggtcaactgt
ccagctttaagttctgtccaatcaaaagaattctgtttagtaaagcatgattgacttcatttggagaccagtctc
ataaccatttatgtttcatcgttaacacaaatttggtatttgctatgtagAATGTCTGTGTCCGAAACAAGGCTG
TGGACAAATGGCAGAGAAAAACACAAGTCACAACTGGTGCAGCTGCTATTAAAGGAAAGCTCCACGCCTTTAACCA
Ggtatagaaagcaaactccgaagatgattttgctcttatctgggatcaatcaggaatctaactttaattaattt
tcaaaaccaAACGTTAGTGAACAGGTTGCTTCTTACATGAGGGATCCAAGTAGAATGATTAACAGATGCAACAA
TCAAGATCTACTGTTGCTGTTTTTGGAACTgtaagttttgctttgataataaaagcataaaccatctaagaattt
ggcctttggtggtcaatagatcagcttctcttcaatttttctgactttaaatttaaacatccacagGTCCCTCAG
GAAGCCATGGAACCAAATCCAGAGGtataaatcacggttttctctaaccataaagagtataaacatttaaatcaatct
cattgattatcttgataaccattctctaacatggtgactctttctgtccagGAAAAACAAGAAGAAGGAGACCT
GAAC TTGTGGAAGATGCTGAGTTCTATAGGCAGTTACTAAAGGAGTTTCTCGAGACTATCGATCCAGCTTCTCAG
gtatagtaaagaaaaacatcctagaatcattagatatgaaacccccctagcgattgttatgtccaacaactcaag
tgtaaagagtctttttatattctgaaatctcaatatttgcattggcagAGGCGGCTTCTACGAGATGAAGAAG

TTCCAGACGAAGAAGAGGAAAAGTTGTGGATCGACGTGCCCTCAAAGAGCAGAAAAGATCAGGttaaataaaacacacat
agtttcacctccttgaggttttctcttcccataattctctcttttgaatcattttcttgtaaaaaatcgaaacagG
TATAATGTTTCATGATCCCTATACCAGGAACACGCATATTACACAAAAGAACATTCTTTTGCCGCGAAGATTCTCTCA
AACACGGCGGATTTGCTTAAGAATCTCTTCGGTCTCAAGACCAGAAAACGTTTCAGTCTGAAGCTTAA

In *fan* mutant g changed to a (highlighted with green), which leads to RNA mis-splicing:

ATGGCTGGGGGTCAAAGAGGTCTAAAAGAGCAAGACTTGACAGTGAATCGGAAGACATAAGCGACCAAGAAAACC
TTAAGgtatggtttatgttgcttttctcgtttaagttattagatggtataggtgaatttatttggttgagggtt
ctctatattatagGCCGAAAGCGACAATGAAGATGATCAATTACCTGATGGGATAGAGGATGATGAAGTAGATAGC
ATGGAAGATGATGAGGGAGAGAGTGAGGAAGACGATGAAGGAGATACCGAGGAAGATGATGAAGGAGATAGCGAGG
AAGATGATGAAGGAGAGAAACAAGGAAGACGAGGATGGAGAAAAGCGAGGACTTTGAAGATGGAAATGATAAAGAGAG
TGAGAGTGGCGATGAAGGTAATGATGACAATAAAGATGCTCAGATGGAAGAGCTTGAGAAAAGAGGTCAAGGAGCTT
CGTTCACAAGAGCAGtaagatttcatttgaactaatcttttaataactgttatttctaagatagtgatttcttg
tatgtgtttcttcttagaaaacggtagatagtcgtattaaagtttcaactgtttaatctagatcagcaaggatggtg
gctagatttaCaTATGGCTTTTCGATAGAAACGGTGTCAATAGTAGTATTACAACACACCACAGAAAAAGGGACGAT
CTATTACAGCGTTTCAAATTTGTGGTTTGTGTTGATATTAAGCTTCAGACTGAACGTTTCTGGTCTTGAGACCGAAG
AGATTCTTAAGCAAATCCGCCGTGTTTGGAGGAATCTTCGCTGGTCTGGAGCCATAAAGTTGACAATCTTCTCAT
GAACATTATACCTGTTTCGATTTTTTACAAGAAAATGATTCAAAAGAGAGAATTATGGGAAGAGAAAACCTCCAAG
AGGTGAAACTATGTGTGTTTTATTTACCTGATCTTCTGCTCTTTGAGGCACGTCGATCCACAACCTTCTCTTCT
TCGCTCTGGAACCTTCTTCATCTCGTAGAAAGCCGCCTCTGCCAATGGCAAATATTGAGATTTCAGAATATAAAAAAG
ACTCTTTACACTTGAGTTGTTGGACATAACAATCGCTAGGGGGGTTTCATATCTAATGATTTCTAGGATGTTTTTC
TTTACTATACCTGAGGAAGCTGGATCGATAGTCTCGAGAACTCCTTTAGTAACTGCCATAGAACTCAGCATCTT
CCACAAGTTCAGGGTCTCCTTCTTCTGTTTTTCTGGACAAGAAAGAGTACACATGTTAGAGAATGGGTATCAAG
ATAATCAATGAGATTGATTAATGTTTATACTCTTTATGGTTAGAGAAAACCGTGATTTTACCTCTGGATTTGGTT
CCATGGCTTCTGATATGGCTTTTCGATAGAAACGGTGTCAATAGTAGTATTACAACACACCACAGAAAAGGGACG
ATCTATTACAGCGTTTCAAATTTGTGGTTTGTGTTGATATTAAGCTTCAGACTGAACGTTTCTGGTCTTGAGACCGA
AGAGATTTCTTAAGCAAATCCGCCGTGTTTGGAGGAATCTTCGCTGGTCTGGAGCCATAAAGTTGACAATCTTCTC
ATGAACATTATACCTGTTTCGATTTTTTACAAGAAAATGATTCAAAAGAGAGAATTATGGGAAGAGAAAACCTCCA
AGAGGTGAAACTATGTGTGTTTTATTTACCTGATCTTCTGCTCTTTGAGGCACGTCGATCCACAACCTTCTCTT
CTTCGCTCTGGAACCTTCTTCATCTCGTAGAAAGCCGCCTCTGCCAATGGCAAATATTGAGATTTCAGAATATAAAAA
AGACTCTTTACACTTGAGTTGTTGGACATAACAATCGCTAGGGGGGTTTCATATCTAATGATTTCTAGGATGTTTT
TCTTTACTATACCTGAGGAAGCTGGATCGATAGTCTCGAGAACTCCTTTAGTAACTGCCATAGAACTCAGCATC
TTCCACAAGTTCAGGGTCTCCTTCTTCTGTTTTTCTGGACAAGAAAGAGTACACATGTTAGAGAATGGGTATCA
AGATAATCAATGAGATTGATTAATGTTTATACTCTTTATGGTTAGAGAAAACCGTGATTTTACCTCTGGATTTGG
TTCCATGGCTTCTGAGGGACCTGTGGATGTTTtagaagtaggttcgaactcattatagcaattggatttattcgtg
attagtagaaaatataacgtttaaaaggaaaagaacagacttttgccaaaagttttgtcagatactgtcttaat
tcggttgcacatthtaattttTACAAGAAAATGATTCAAAAGAGAGAATTATGGGAAGAGAAAACCTCCAAGAGG
TGAAACTATGTGTGTTttaaatagaacgTGTC AATAGTAGTATTACAACACACCACAGAAAAGGGACGATCTATTA
CAGCGTTTCAAATattgagaggcaaaagaatggttcttcttgtaatatgctgttctggtatagGGATATATTGA
AGAACTTGAAGCGTGATAAGGGTGAAGATGCTGTTAAAGGTCAAGCGGTGAAGAATCAGAAGGgtatgacctgttc
tgcggtttgcaagtagtctaactagaaatcagttgcagttgtatattcctattttatgtaataagttgatcta
gagcgaatttttgacaattcttcttcttctgtctctatgtaaagGCTCTTTGGGATAAGATTTCTGGAGTTTCAGATTCT
TACTTCAGAAAAGCATTGATCGTTCAAACAGATTACCACAGgtaagctatgctcttctcctcttctGtgGAAGCAA

CCTGTTCACTAACGTTTtGGTTTTGAAAATTAATTTAAAGTTAGATTCCCTGATTGAtCCCCAGAtAAGAGCattta
tggttggttggtttcatcatactttatgacagaaATATATTTTCTACTAATCACGAATAAATCCAATTGCTATAATG
AGTTCGAACCTACTTCTAAATCTAGCCAACATCCTTGCTGATCTAGATTTAAACAGTTGAACTTTAATACGACTAT
CTACCGTTTCTAGAAAGAAACACATACAAGAATTCACATCTTAGAAATAACAGTATTTAAAAATGATTAGTTCAA
ATGAAATCTTACTGCTCTTGTGAACGAAGCTCCTTGACCTtTTTACAAGAAAATGATTCAAAAGAGAGAATTATG
GGAAGAGAAAACCTCCAAGAGGTGAAACTATGTGTGTTTTATTTACCTGATCTTTCTGCTCTTTGAGGCACGTGCA
TCCACAACCTTTCTCTTCTTCTGCTGGAACCTTCTTCATCTCGTAGAAAGCCGCTCTGCCAATGGCAAATATTGAG
ATTTCAGAATATAAAAAAGACTCTTTACACTTGAGTTGTTGGACATAACAATCGCTAGGGGGGTTTCATATCTAA
TGATTCTAGGATGTTTTCTTTACTATACCTGAGGAAGCTGGATCGATAGTCTCGAGAACTCCTTTAGTAACTGC
CTATAGAACTCAGCATCTTCCACAAGTTCAGGGTCTCCTTCTTCTTGTTTTTCTGACAAAGAGTACACATG
TTAGAGAATGGGTATCAAGATAATCAATGAGATTGATTTAAATGTTTATACTCTTTATGGTTAGAGAAAACCGTGAT
TTTACCTCTGGATTTGGTTCATGGCTTCTGAGGGACCTGTGGATGTTTAAATTTAAAGTCAGGAAAAATTGAAG
AGAAGTGATCTATTGACCACAAAGGCCAAATTCCTTAGATGGTTTTATGCTTTTATTATCAAAGCAAAAACCTTACA
GTTCCAAAAACAGCAACAGTAGATCTTGATTGTTGCATCTGTTAATCATCTACTTGGATCCCTCATGtGGAAGC
AACCTGTTCACTAACGTTTtGGTTTTGAAAATTAATTTAAAGTTAGATTCCCTGATTGAtCCCCAGAtAAGAGCCTT
TCTCAAGCTCTTCCATCTGAGCATCTTTATGTGATCATTACCTTCATCGCCACTCTCACTCTCTTTATCATTTCC
ATCTTCAAAGTCTCGCTTCTCCATCTGACAGATAGAAAAGAAAGGAgatGtTATATTGACCCTCGCTTATAG
TGATGATAATTATGTTATCTGTTAATATGGCATAACCATTCACTTGTGATCAACCGAGGGGTTCTTCTCAAACAAA
GCCTGTAGAAATCCATATTTAGATAATAATAAAAAGTCAACAAACGTGGCAGAGCGACAAAACACTAAGAGACA
GCATACAATTACTGAATCAAGCCTATGCATACATACCTCTTGCAACTCCAAGAGGGAATCTAATGTCTTCTTAGAT
GAAGTAACTAGATCTGTATATGCTGTTGAGACATCTCATCTTCTGAACAAAATAACGATTTACAGGCTCCTGAA
AAGGGTACCTTCTGTCAAAAGTATGATGAAACAAACAACCATAAATAGAAGAGGAGAAGAGCATAGCTTACCTGT
GGTAATCTGTTTGAACGATCAAATGCTTCTGAAAGTAAGAACTGAACTCCAGAATCTTATCCCAAAGAGCCTTTA
CATAGAGCACAAGAAAGAATTGTCAAAAATTCGCTCTAGATCAACTTCATTAACATAAAAATAGGAATATACAACT
GCAACTGATTTCTAGTTAGACTACTTCGCAAACCGCAGAACAAGGTCATACCTTCTGATTTCTCACCGCTTGACct
tCGTCTCCTTGTCTCTCCTTCATCATCTTCTCGCTATCTCCTTCATCATCTTCTCGGTATCTCCTTCATCGT
CTTCTCACTCTCTCCCTCATCATCTTCCATGCTATCTACTTCATCATCTTCTATCCCATCAGGTAATTGATCATC
TTCATTGTGCTTTTCGGCTATAATATAGAGAACCCTCAACCAAATAAATTCACCTATACCATCTAAATAACTTAA
AcgaGacaAAAGCAACAtaACATACCTTaaGTTTTCTTggtCGCTTATGTCTTCCgATTCACTGggtaccctttt
cagGAGCCTGTGAAATCGTTATTTTGTTCAGAAGATGAGGATGTCTCAACAGCATATACAGATCTAGTTACTTTCAT
CTAAGAAGACATTAGATTCCTCTTGGAGTTGCAAGAGgtatgtatgcataggcttgattcagtaattgtatgctg
tctcttagtgtttgtcgctctgccacgtttggtgacttttattattattatctaaatatggatttctacagGCTT
TGTTTGAGAAGAACCCCTCGGTTGATCAACAAGTGAATGgtatgccatattaacagataacataattatcatcact
ataagcgagggtcaatataacatctcctttcttttctatctgtcagCAACAGCTAGTGAAGAATCCAATAAATCA
GATGCAGAAGATAGCGATGAATGGCAGCGAATATCTGACTTGCAAGAAAGgtattcaagactaaatgttcaactgt
ccagctttaagttctgtccaatcaaaagaattctgtttagtaaagcatgattgacttcatttggagaccagtctc
ataaccatttatgtttcatcgtaacacaaatttggatattgctatgtagAATGTCTGTGTCCGAAACAAGGCTG
TGGACAAATGGCAGAGAAAAACACAAGTCACAACTGGTGCAGCTGCTATTTAAAGGAAAGCTCCACGCCTTTAACCA
Ggtatagaagcaaaactccgaagatgattttgctcttatctgggatcaatcaggaatctaactttaattaattt
tcaaaaccaAACGTTAGTGAACAGGTTGCTTCTACATGAGGGATCCAAGTAGAATGATTTAAACAGATGCAACAA
TCAAGATCTACTGTTGCTGTTTTTGGAACTgtaagtttttgctttgataataaaagcataaaccatctaagaattt
ggcctttggtggtcaatagatcagcttctcttcaatttttctgactttaaatttaacatccacagGTCCCTCAG
GAAGCCATGGAACCAAATTCAGAGgtaaaatcacggttttcttaaccataaagagtataaacatttaaatcaatct
cattgattatcttgataaccattcttaacatgtgtactctttctgtccagGAAAACAAGAAGAAGGAGACCTT

GAACTTGTGGAAGATGCTGAGTTCTATAGGCAGTTACTAAAAGGAGTTTCTCGAGACTATCGATCCAGCTTCCTCAG
gtatagtaaagaaaaacatcctagaatcattagatatgaaacccccctagcgattgttatgtccaacaactcaag
tgtaaagagtcttttttatattctgaaatctcaatatttgcattggcagAGGCGGCTTCTACGAGATGAAGAAG
TTCCAGACGAAGAAGAGGAAAAGTTGTGGATCGACGTGCCTCAAAGAGCAGAAAAGATCAGGtaaataaaacacacat
agtttcacctcttggaggttttctcttccataattctctcttttgaatcattttcttgtaaaaaatcgaaacagG
TATAATGTTTCATGATCCCTATACCAGGAACACGCATATTACACAAAAGAACATTCTTTTGCCGCGAAGATTCTCCA
AACACGGCGGATTTGCTTAAGAATCTCTCGGTCTCAAGACCAGAAAACGTTTCAGTCTGAAGCTTAA

FAN wild type (AT5G61330) CDS:

ATGGCTGGGGGGTCAAAGAGGTCTAAAAGAGCAAGACTTGACAGTGAATCGGAAGACATAAGCGACCAAGAAAACC
TTAAGGCCGAAAGCGACAATGAAGATGATCAATTACCTGATGGGATAGAGGATGATGAAGTAGATAGCATGGAAGA
TGATGAGGGAGAGAGTGAGGAAGACGATGAAGGAGATACCGAGGAAGATGATGAAGGAGATAGCGAGGAAGATGAT
GAAGGAGAGAACAAGGAAGACGAGGATGGAGAAAAGCGAGGACTTTGAAGATGGAAATGATAAAGAGAGTGAGAGTG
GCGATGAAGGTAATGATGACAATAAAGATGCTCAGATGGAAGAGCTTGAGAAAAGAGGTCAAGGAGCTTCGTTTACA
AGAGCAGGATATATTGAAGAACTTGAAGCGTGATAAGGGTGAAGATGCTGTTAAAGGTCAAGCGGTGAAGAATCAG
AAGGCTCTTTGGGATAAGATTCTGGAGTTCAGATTCTTACTTCAGAAAGCATTGATCGTTCAAACAGATTACCAC
AGGAGCCTGTGAAATCGTTATTTTGTTCAGAAGATGAGGATGTCTCAACAGCATATACAGATCTAGTTACTTCATC
TAAGAAGACATTAGATTCCCTCTGGAGTTGCAAGAGGCTTTGTTTGAGAAGAACCCTCGGTTGATCAACAAGTG
AATGCAACAGCTAGTGAAGAATCCAATAAATCAGATGCAGAAGATAGCGATGAATGGCAGCGAATATCTGACTTGC
AGAAAAGAATGTCTGTGTTCCGAAACAAGGCTGTGGACAAATGGCAGAGAAAAACACAAGTCACAACTGGTGCAGC
TGCTATTAAGGAAAGCTCCACGCCTTTAACCAGAACGTTAGAACAGGTTGCTTCTTACATGAGGGATCCAAGT
AGAATGATTAACAGATGCAACAATCAAGATCTACTGTTGCTGTTTTTGGAACTGTCCCTCAGGAAGCCATGGAAC
CAAATCCAGAGGAAAAACAAGAAGAAGGAGACCCTGAACTTGTGGAAGATGCTGAGTTCTATAGGCAGTTACTAAA
GGAGTTTCTCGAGACTATCGATCCAGCTTCTCAGAGGCGGCTTTCTACGAGATGAAGAAGTTCAGACGAAGAAG
AGGAAAGTTGTGGATCGACGTGCCTCAAAGAGCAGAAAAGATCAGGTATAATGTTTCATGAGAAGATTGTCAACTTTA
TGGCTCCACGACCAGCGAAGATTCTCCAAACACGGCGGATTTGCTTAAGAATCTCTTCGGTCTCAAGACCAGAAA
CGTTCAGTCTGAAGCTTAA

In *fan* mutant CDS, the yellow highlighted fragment in *FAN* wild type ‘AACGTTAG’ is missing.

ATGGCTGGGGGGTCAAAGAGGTCTAAAAGAGCAAGACTTGACAGTGAATCGGAAGACATAAGCGACCAAGAAAACC
TTAAGGCCGAAAGCGACAATGAAGATGATCAATTACCTGATGGGATAGAGGATGATGAAGTAGATAGCATGGAAGA
TGATGAGGGAGAGAGTGAGGAAGACGATGAAGGAGATACCGAGGAAGATGATGAAGGAGATAGCGAGGAAGATGAT
GAAGGAGAGAACAAGGAAGACGAGGATGGAGAAAAGCGAGGACTTTGAAGATGGAAATGATAAAGAGAGTGAGAGTG
GCGATGAAGGTAATGATGACAATAAAGATGCTCAGATGGAAGAGCTTGAGAAAAGAGGTCAAGGAGCTTCGTTTACA
AGAGCAGGATATATTGAAGAACTTGAAGCGTGATAAGGGTGAAGATGCTGTTAAAGGTCAAGCGGTGAAGAATCAG
AAGGCTCTTTGGGATAAGATTCTGGAGTTCAGATTCTTACTTCAGAAAGCATTGATCGTTCAAACAGATTACCAC
AGGAGCCTGTGAAATCGTTATTTTGTTCAGAAGATGAGGATGTCTCAACAGCATATACAGATCTAGTTACTTCATC
TAAGAAGACATTAGATTCCCTCTGGAGTTGCAAGAGGCTTTGTTTGAGAAGAACCCTCGGTTGATCAACAAGTG
AATGCAACAGCTAGTGAAGAATCCAATAAATCAGATGCAGAAGATAGCGATGAATGGCAGCGAATATCTGACTTGC
AGAAAAGAATGTCTGTGTTCCGAAACAAGGCTGTGGACAAATGGCAGAGAAAAACACAAGTCACAACTGGTGCAGC

TGCTATTAAAGGAAAGCTCCACGCCTTTAACCAGTGAACAGGTTGCTTCCTACATGAGGGATCCAAGTAGAATGAT
TAAACAGATGCAACAATCAAGATCTACTGTTGCTGTTTTTGGAACTGTCCCTCAGGAAGCCATGGAACCAAATCCA
GAGGAAAAACAAGAAGAAGGAGACCCTGAACTTGTGGAAGATGCTGAGTTCTATAGGCAGTTACTAAAGGAGTTTC
TCGAGACTATCGATCCAGCTTCCTCAGAGGCGGCTTTCTACGAGATGAAGAAGTTCCAGACGAAGAAGAGGAAAGT
TGTGGATCGACGTGCCTCAAAGAGCAGAAAGATCAGGTATAATGTTTCATGAGAAGATTGTCAACTTTATGGCTCCA
CGACCAGCGAAGATTCCTCCAAACACGGCGGATTTGCTTAAGAATCTCTTCGGTCTCAAGACCAGAAACGTTTCAGT
CTGAAGCTTAA

Dear Reviewers,

Thank you for your valuable comments and professional guidance. We are grateful for the opportunity to submit this revised manuscript, which we believe has been significantly improved based on your insightful suggestions.

In response to your feedback, we have made the necessary corrections to enhance the quality of our work. Specifically, we have changed the gene of interest from "*FAN*" to "*AtCHE1*".

The additional reviewers' comments and our detailed responses are organized numerically in the table below for your convenience.

Thank you once again for your attention and support. We look forward to your feedback at your earliest convenience.

Yours sincerely,
Prof. Dr. Klaus Palme

Reviewers' comments:

Reviewer #2 (Remarks to the Author):

The is a review of a revised ms. The authors have identified a mutant with defect root tip meristem organization and slow growth, which they have named FAN. Map-based cloning reveals that this may be a homolog of the mammalian CHE-1 gene (which performs a wide range of functions) and it is the closest Arabidopsis homolog. The authors characterize many aspects of the early seedling phenotype (similar to some other genome maintenance-defective mutants, such as teb), and come to a more specific conclusion about the function of FAN than I think is justifiable. I do appreciate discovering new-(ish) genes via EMS mutagenesis, and I do think this report will be useful, I just think the function of FAN, and there may be many functions (if it's a CHE-1 homolog), is still unclear. I think the phenotype described is consistent with some defect in genome maintenance- but as I said before, this could be due to defects in metabolism, repair, replication, or regulation of damage response. I don't agree that their results conclusively indicate that FAN operates upstream of ATR in a regulatory pathway. I think their results are fairly described in the Discussion, but over interpreted in the title and abstract.

L118 I might mention here that abbreviation of the mitotic zone is often observed in roots undergoing genomic stress.

L150 tell us briefly how you defined the promoter (maybe just everything up to the next predicted gene?)

Fig 2A Did you accidentally switch the labels for the mutant vs wt gDNA sequences? Didn't wt have a G that EMS turned into an A?

L173 Does GFP localize to the nucleus on its own?

L181 Fig 3 are you sure you have the right DAPI image in the middle of fig. 3J? Maybe the right image but it was flipped?

L215 if FAN is required for the prevention of DSBs under routine growth conditions (and it has a phenotype under routine growth conditions), we'd expect an increase in the expression of these genes. 5FGH assays very responsive DDR genes in the absence of exogenously applied DNA damaging agents. But your statistics in Fig. 5FGH indicate (a little surprisingly!) that they are not significantly different from the level of expression in wt (both columns are labeled a). So your "increased expression is not significant. Or is there an error in the figure?

L217 Be a little more specific here? Does CHE1 play a role in the induction or the suppression of the DDR in mammals.

L225 again, your figure 5F doesn't support the notion that PARP is induced to a level significantly different from wt. This "does transcriptional DDR occur" experiment is a really important experiment if you want to make a case for a role in the induction of damage response, rather than a role in preventing damage occurrence. Also, when you say "relative expression" on the Y axis, I

assume you mean vs some control transcript, like actin. Not vs the uninduced level? Just clarify that in the figure legend.

L233 it's not clear that the level of expression between the two OE lines is significantly different, and the two lines don't seem to have the same phenotypes. I'd drop the whole OE section.

L244 The shrinkage of GUS expression is I'm sure due to the shrinkage of the mitotic zone with zeo. The diminishing of GFP expression may be due to the extensive cell death.

L258 this is a nice result indicating that the effects of sog- and fan- are synergistic- that they independently protect the genome via different pathways. The term "work together" is a little vague and suggests a more intimate collaboration, I'd drop that.

L269 again, I suggest that the restoration of HU resistance by FAN may simply because the fan defect is slowing growth down by a lot. One of ATR's roles is to arrest the cell cycle in response to the depletion of dNTPs induced by HU. Any mutation that drastically reduces cell division rate should enhance HU resistance. If FAN was truly upstream of ATR- required to induce ATR function in the presence of replicative stress- then its phenotype would be similar to that of the atr mutant. If it were instead upstream and required to suppress ATR function in the absence of replicative stress... then, hmmm, that might explain the constitutive slow growth. Would it also explain other phenotypes?

L262 this paragraph seems to ignore the wt phenotypes, just comparing the mutants. It just needs some corrections- for example, not all seedlings display cell death with HU.

Response to Reviewer #2

Table 1. Comments from reviewer #2 and author's response

Comments	Responses
1. I don't agree that their results conclusively indicate that FAN operates upstream of ATR in a regulatory pathway. I think their results are fairly described in the Discussion, but over interpreted in the title and abstract.	Thank you for valuable comment. We have continued to tone down in abstract and discussion part this time, because of the lack of evidence about AtCHE1 involving in ATR pathway. But we believe that the title "AtCHE1, the Arabidopsis homolog of mammalian AATF/Che-1 protein, is involved in safeguarding genome stability." is already toning down compared to the last version "FAN, the homolog of mammalian Apoptosis Antagonizing Transcription Factor AATF/Che-1 protein, is involved in safeguarding genome through the ATR induced pathway in Arabidopsis." In abstract: line 37-42: "Under standard

	conditions, the fan mutant exhibits a cell death response and differentiation defects at the root tip. Additionally, the phenotypic analysis of the atr-2;fan double mutant reveals that FAN and ATM/RAD53-RELATED (ATR) operate within the same pathway. Collectively, these findings underscore the importance of FAN in meristem maintenance and the DNA damage response.” has been changed to “Under standard conditions, the che1 mutant exhibits an accumulation of damaged DNA, cell death and differentiation defects at the root tip. Collectively, these findings underscore the importance of AtChe-1 in meristem maintenance and the genome stability.” In Result: line 253:” FAN plays a synergistic role with SOG1 and is partially involved in the ATR-induced DNA damage response” has been changed to “AtCHE1 plays a synergistic role with SOG1”. In Discussion: line 326-331:”These findings suggest that FAN is essential for triggering the inhibition of root development in the ATR induce pathway during replication stress. ” has been deleted. And “This phenomenon could be explained by the followings: either, ATR and AtCHE1 act in the same pathway, ATR is upstream or downstream of AtCHE1 or the retarded growth of the che1 mutant partly restore the function of ATR arresting the cell cycle in response to HU treatment. Additional experimentation is required to provide evidence to verify these options.” has been added.
2. L118 I might mention here that abbreviation of the mitotic zone is often observed in roots undergoing genomic stress.	Thank you for your suggestion. In line 111, we have revised the text: “While Col-0 showed a slight increase in cell numbers at 6 DAG, the che1 mutant continued to exhibit a decline (Figure 1B, D), indicating a progressive retardation of the meristem, and that shortening of the

	mitotic zone is often observed in roots undergoing genomic stress (Ühlken et al., 2014)". Additionally, we cited a research article that describes a mutant with reduced meristem size under DNA damage stress.
3. L150 tell us briefly how you defined the promoter (maybe just everything up to the next predicted gene?)	We cloned a 2197bp region upstream of first exon of AtCHE1. This 2197bp DNA fragment encompasses the upstream DNA including 5'UTR of AT5G61330 (AtCHE1), part of exon and 3'UTR of the neighboring gene AT5G61340, with the purpose of containing any potential distal control elements. To clarify our definition of the promoter, we add supplementary information in parentheses to the following sentence "To confirm that the mutation in the AtCHE1 gene was solely responsible for the growth defects observed in the che1 mutants, we transformed homozygous che1 plants with the full-length genomic region of AtCHE1, expressed under its own promoter. (The sequence spanning 2197 bp between AtCHE1 and the adjacent gene AT5G61340, along with portions of both exon and the 3'UTR of AT5G61340. This design aims to encompass any potential distal control elements.) and including its 3' -UTR (AtCHE1_{pro}-AtCHE1_g-AtCHE1_{3'-UTR})." Line 144 to Line 148.
4. Fig 2A Did you accidentally switch the labels for the mutant vs wt gDNA sequences? Didn't wt have a G that EMS turned into an A?	Thank you for your careful revision. We accidentally switched the labels in Figure 2A, where 'fan gDNA' should be labeled as 'FAN gDNA. As, we changed the gene name from 'FAN' to 'AtCHE1', the labels of sequences from top to bottom should be 'che1 gDNA' 'AtCHE1 gDNA' 'che1 CDS' and 'AtCHE1 CDS'.
5. L173 Does GFP localize to the nucleus on its own?	This is an important question that prompts us to consider the subcellular localization of AtCHE1 with greater care. GFP (fluorescent protein) and its variants are used to study the subcellular localization of proteins by analyzing fusion proteins with limitations.

	Because of GFP's low molecular weight of 27kDa, it can move through the nuclear pores and translocate to the nucleus on its own with no NLS (nuclear localization signals) (Seibel, N. M., Eljouni, J., Nalaskowski, M. M., & Hampe, W. (2007). Nuclear localization of enhanced green fluorescent protein homomultimers. Analytical biochemistry, 368(1), 95 - 99.). Considering that a minority of nuclear proteins with molecular mass less than approximately 50kDa diffuse passively through the nuclear pores (I.G. Macara, Transport into and out of the nucleus, Microbiol. Mol. Biol. Rev. 65 (2001) 570 - 594), it is reasonable to conclude that AtCHE1 is located within the nucleus.
6. L181 Fig 3 are you sure you have the right DAPI image in the middle of fig. 3J? Maybe the right image but it was flipped?	We apologize for the error in Figure 3J, erroneously the wrong cells were cropped; This has now been corrected.
7. L215 if FAN is required for the prevention of DSBs under routine growth conditions (and it has a phenotype under routine growth conditions), we'd expect an increase in the expression of these genes. 5FGH assays very responsive DDR genes in the absence of exogenously applied DNA damaging agents. But your statistics in Fig. 5FGH indicate (a little surprisingly!) that they are not significantly different from the level of expression in wt (both columns are labeled a). So your "increased expression is not significant. Or is there an error in the figure?	Thanks for your observation. There is no error in Figure 5. As this issue is consistent with comment #9 we respond to this comment in details at #9.
8. L217 Be a little more	CHE1/AATF plays a role in DNA damage

specific here? Does CHE1 play a role in the induction or the suppression of the DDR in mammals.	response though different pathways and exhibits different function depending on the level of DNA damage. In response to DNA damage, CHE1/AATF can be phosphorylated by ATM which contributes to cell cycle arrest. On the other hand, when DNA damage exceeds repair capacity, cells undergo apoptosis. This occurs due to the degradation of CHE1/AATF, resulting in its inability to repress apoptotic processes. To clarify this point further, the sentence has been reorganized as follows "In mammals, AATF/Che1 plays an essential role in DNA damage response pathway, contributes to cell cycle arrest and apoptosis depending on the DNA damage level (Passananti and Fanciulli, 2007)." Line 218-220.
9. L225 again, your figure 5F doesn't support the notion that PARP is induced to a level significantly different from wt. This "does transcriptional DDR occur" experiment is a really important experiment if you want to make a case for a role in the induction of damage response, rather than a role in preventing damage occurrence. Also, when you say "relative expression" on the Y axis, I assume you mean vs some control transcript, like actin. Not vs the uninduced level? Just clarify that in the figure legend.	Thanks for your comment. Under control condition, these results confirmed our observations upon PI staining, comet assay or immunofluorescence staining. Although PARP2 is not induced in which mutant, but BRCA1 and RAD51 are induced. In contrast, under stress, though the transcription of BRCA1, RAD51 and PARP2 is induced in the which mutant at lower than in the wildtype. We assume either that the transcription of these genes is affected by AtCHE1 or the accumulation of cell death in the mutant is due to the lack of BRCA1, RAD51 and PARP2 involved in DNA damage repair. We added this assumption in Discussion Line 319 to 322:" It is noteworthy that the transcription levels of BRCA1, RAD51, and PARP2 are elevated in mutants but remain lower than those observed in wild-type. We assume either that the transcription of these genes is affected by AtCHE1 or the accumulation of cell death in the mutant is due to the lack of BRCA1, RAD51 and PARP2 involved in DNA damage repair"

	We have clarified: "relative expression level" means gene vs reference gene in the Figure legend.
10.L233 it's not clear that the level of expression between the two OE lines is significantly different, and the two lines don't seem to have the same phenotypes. I'd drop the whole OE section.	As the data obtained from the experimentation with the overexpression lines do not lead to significant conclusions we have removed the entire section on overexpression from the article.
11. L244 The shrinkage of GUS expression is I'm sure due to the shrinkage of the mitotic zone with zeo. The diminishing of GFP expression may be due to the extensive cell death.	We agree with your conclusion that the root growth is inhibited by zeocin treatment. We added a sentence to make it clearly in Line 251 to 252. "It is worth to note that under zeocin treatment, the reduction in GFP and GUS expression is likely due to extensive cell death and shrinkage of the mitotic zone. "
12. L258 this is a nice result indicating that the effects of sog- and fan- are synergistic- that they independently protect the genome via different pathways. The term "work together" is a little vague and suggests a more intimate collaboration, I'd drop that.	We removed the phrase "work together" from this sentence to enhance its precision. In L258 to L260, the sentence was revised from "This heightened response indicates a synergistic interaction, with FAN and SOG1 working together to maintain meristematic cell integrity" to "This elevated response indicates a synergistic interaction, with AtCHE1 and SOG1 maintaining meristematic cell integrity"
13.L269 again, I suggest that the restoration of HU resistance by FAN may simply because the fan defect is slowing growth down by a lot. One of ATR's roles is to arrest the cell cycle in response to the depletion of dNTPs induced by HU. Any mutation that drastically reduces cell division rate should enhance HU resistance. If FAN was truly upstream of ATR-	Thank for your suggestion. The restored phenotype of the double mutant compared to the atr-2 mutant under HU treatment has perplexed us. We assume that the loss of AtCHE1 may partially restore growth defects under HU treatment, however, we do not have any evidence related to the underlying mechanisms. What is the relationship between ATR and AtCHE1: is AtCHE1 regulated by ATR or vice versa? To clarify this interaction, further experimentation is required. Nevertheless, the observations of short root meristems and enlarged cells

required to induce ATR function in the presence of replicative stress- then its phenotype would be similar to that of the atr mutant. If it were instead upstream and required to suppress ATR function in the absence of replicative stress... then, hmmm, that might explain the constitutive slow growth. Would it also explain other phenotypes?	indicate that cell arrest and potentially endocycle events occur in the che1 mutant. This would support your initial hypothesis that a reduction in cell division rate enhances HU resistance in the atr-2;che1 double mutant. We have tone down our conclusions regarding the atr-2;che1 double mutant in Line39-40. The sentence "Additionally, the phenotype analysis of the atr;che1 double mutant reveals that FAN and ATR/RAD-RELATED (ATR) operate within the same pathway." has been removed from Abstract and Line 100-101, the last paragraph of Introduction, "Our analysis of atr-2;fan double mutant shows that FAN and ATR act in the same signaling pathway" has been replaced by "Our analysis of sog1-1;che1 double mutant shows that AtCHE1 and SOG1 plays a synergistic role in genome safeguarding". Line 276-279, the conclusion is corrected as "These findings suggest that the loss of AtCHE1 partially rescues impaired root development caused by the deficiency of ATR under HU stress." in the Result and Discussion Line 326-329, the sentence "This phenomenon could be explained by the followings: either, ATR and AtCHE1 act in the same pathway, ATR is upstream or downstream of AtCHE1 or the retarded growth of the che1 mutant partly restore the function of ATR arresting the cell cycle in response to HU treatment. Additional experimentation is required to provide evidence to verify these options." was added. "These findings suggest that FAN is essential for triggering the inhibition of root development in the ATR-induced pathway during replication stress" was removed in lines 329-331.
14.L262 this paragraph seems to ignore the wt phenotypes, just comparing	In line 256-267, we have reformulated the sentence from "The introgressed sog1-1;che1 double mutant exhibited an enhanced

the mutants. It just needs some corrections- for example, not all seedlings display cell death with HU.	level of cell death under control conditions compared to either single mutant, che1 or sog1-1 , which also develops cell death (Figure 6).” to “The introgressed sog1-1;che1 double mutant showed enhanced level of cell death under control conditions compared to either single mutant, che1 and sog1-1 , as well as Col-0 (Figure 6).
---	---

Reviewer #3 (Remarks to the Author):

The authors significantly improved the text of the manuscript and sufficiently answered all my concerns and questions. I would still prefer to simply call the gene AtCHE1 instead of FAN, but I understand the personal reasons why the authors doesn't want to change it. Anyhow, I recommend the article for publication.

Reply to reviewer #3

Tabel 2. Comments from reviewer #3 and author's response

Comment	response
I would still prefer to simply call the gene AtCHE1 instead of FAN, but I understand the personal reasons why the authors doesn't want to change it.	Thank you for your valuable advice. We have reconsidered your suggestion and agree to change the gene name from “ FAN ” to “ AtCHE1, ” as well as the corresponding mutant name from “ fan ” to “ che1, ” in order to avoid any potential confusion with Fanconi anemia pathway genes for readers.

Other modifications:

1. After reviewing the images shown in the earlier version, we noticed that some of the original images were missing and found better quality images. Therefore, we have replaced Figure 1A and 1F with better quality images from our experiments. Figure S1 has been removed. These changes do not have any impact on our conclusions of the study.
The error in Figure 3J where incorrect cells were cropped has now been corrected. Furthermore, the image of *che1* root in Figure 6A has been replaced with a more representative one.
2. Figure 1E and Figure 5E have been converted to dot-plot graphs to show the data.
3. In the revised Figure S1A and Figure S7, seedlings from the different mutants are delineated by white lines for clarify.

4. The phylogenetic tree (Figure S2) has been replaced. The gene names from each species have been replaced by their corresponding species names.
5. In Figures S4, S5, and S6 we have indicated the mutant with its revised name.
6. Upon Reviewer's 2 comment, we have removed the data related to the overexpress lines (previously Figure S7 and S8)
7. We have transferred the gDNA and CDS sequences of both, the wildtype and *che1* mutant in Supplement data1.
8. We have incorporated the sections titled "Statistics and reproducibility" and "Data availability" to the main body of the article, in lines 484 to 488 and lines 505 to 508.
9. We have changed the title of Figure 7 "FAN participates in ATR induced DNA damage response" to "Loss of AtCHE1 partially rescues impaired root development of *atr-2* under HU treatment".

1 **Figure 1. Primary root growth and stem cell niche (SCN) maintenance are inhibited in the *che1***
 2 **mutant.**

3

4

5

6 **Figure 1. Primary root growth and SCN maintenance are inhibited in the *che1* mutant.**

7 (A) Growth habit of the *che1* mutant compared to the control, Col-0 at 6 DAG under normal growth
8 conditions. Scale bar = 1 cm. (B) Representative images of differential interference contrast (DIC)
9 microscopy showing the root meristem of Col-0 and *che1* (4 and 6 DAG). Scale bar = 50 μ m;
10 arrowheads point to QC (White) and first elongated cortical cell position (Green). (C) Meristem length
11 was measured from the QC to the first elongating cortex cell and (D) the number of meristem cells in
12 the cortex of Col-0 and *che1* at 4 and 6 DAG was counted. The values and error bars in (C) and (D)
13 represent means and \pm SD, n>20. Double asterisks indicate highly significant differences ($P<0.01$)
14 between *che1* and Col-0 analyzed by two-tailed Student's t tests. (E) The graph shows the root growth
15 of Col-0 and *che1* between 0-8 days after germination. Data represent the mean with \pm SD; n=30. (F)
16 to (H) Representative confocal images of propidium iodide (PI)-stained Col-0 and *che1* mutant root
17 tips (F) PI only, (G) showing expression of *WOX5_{pro}:GFP* while (H) of columella stem cell marker
18 *J2341*. Scale bars = 50 μ m, arrowheads point to QC. (I) to (K) Col-0 and *che1* roots expressing
19 *CO2_{pro}:H2B-YFP* (I), *CO3_{pro}:H2B-YFP* (J), and *SCR_{pro}:GFP* (K). The ratios indicate the number of
20 seedlings showing abnormal expression patterns compared to the total number of seedlings examined.
21 White arrowheads indicate to QC and green arrowheads point to abnormal cell division. Scale bars =
22 50 μ m. Magenta, PI staining; yellow, YFP and green, GFP.

23

24 **Figure 2. *AtCHE1* encodes AATF/CHE-1 in Arabidopsis.**

25

26

27 **Figure 2. *AtCHE1* encodes AATF/CHE-1 in Arabidopsis.**

28 (A) Sequence comparison of the wild type (*AtCHE1*) and mutant (*che1*) DNA, CDS and predicted
 29 proteins. The red frame shows the last base of the eighth intron, in which G is mutated to A by EMS
 30 mutagenesis. The green frame and black frame show the missing sequence ‘AACGTTAG’ and the
 31 formation of stop codon by mis-splicing. The green arrow indicates the last amino acid of predicted
 32 *che1* mutant protein. The CDS sequences were obtained by reverse transcription-polymerase chain
 33 reaction (RT-PCR) using total RNA isolated from mutant and wild-type seedlings. (B) Comparison
 34 of the conserved protein boxes between the human AATF/Che-1 (hAATF) and AtCHE1. Light blue
 35 box: AATF/Che-1 domain, lila box: TRAUB, orange box: Nuclear localization signal (NLS), yellow
 36 box: Nucleolar localization sequence (NoLS). Red arrowhead: position of the last amino acid of *che1*.
 37 Blue letters and numbers indicate the potential phosphorylation sites on serine and threonine residues
 38 in hAATF and AtCHE1 respectively.

39
40

Figure 3. AtCHE1 is widely expressed and localized in nucleus.

41

42

Figure 3. AtCHE1 is widely expressed and localized in nucleus.

43

(A) to (G) represent GUS expression studies in (A) young seedling, (B) root, (C) young shoot, (D)

44

mature rosette leaf, (E) enlarged image, (F) flowers and (G) silique. The white arrowhead in (A)

45

points to the root apical meristem. (H) The representative confocal image illustrates the expression of

46

AtCHE1 in the root meristem followed in the *AtCHE1_{pro}:AtCHE1_g-GFP* line (6 DAG). The enlarged

47

section highlights its reduced level of expression in the QC compared to the surrounding stem cells.

48

Magenta: PI staining; green, GFP fluorescence. Arrowheads (White) point to QC and the first

49

elongated cortex cell (Green). Scale bars =1000 µm in (A), (C), (D-G), scale bars = 50 µm in (B) and

50

(H). (I) Confocal images of the nuclear localization of the AtCHE1-GFP in the *AtCHE1_{pro}:AtCHE1_g-*

51 *GFP* line transformed with the *35s:H2B-tdTomato* marker. Magenta: H2B-tdTomato; green,
52 *AtCHE1*-GFP. Scale bar = 5 μm . **(J)** DAPI-stained nuclei of the *AtCHE1_{pro}:AtCHE1_g-GFP* line. Blue:
53 DAPI; green, GFP. Scale bar = 5 μm . **(K)** and **(L)** Gray value distribution of the total number of pixels
54 at the position of the white lines marked in **(I)** and **(J)**.
55

56 **Figure 4. Auxin signaling and expression pattern of genes related to root development are**
 57 **altered in *che1*.**
 58

59
 60 **Figure 4. Auxin signaling and expression patterns of root developmental regulator genes are**
 61 **altered in *che1*.**

62 (A) represents the expression of *DR5_{rev}:GFP*. (B) of *PLT1_{pro}:CFP* and (C) *PLT2_{pro}:CFP* in the root
63 tips of 5 DAG Col-0 and *che1*. (D) Expression pattern of *SCR_{pro}:GFP* and (E) *SHR_{pro}:SHR-GFP* in the
64 root tips of 5 DAG Col-0 and *che1*. (A-E) In each image, scale bar = 50 μ m; arrowhead points to QC,
65 Magenta, PI staining; Green, GFP, Cyan, CFP. (F) Relative mRNA levels of *PLT1* and *PLT2* quantified
66 by qRT-PCR in Col-0 and *che1*. (G) *SCR* and *SHR* relative mRNA levels (gene vs reference gene)
67 determined by qRT-PCR in Col-0 and *che1*. For the analysis, both in (F and G), root tips (2 mm long)
68 were collected, values and error bars represent means and \pm SD from three independent biological
69 replicates. Asterisks indicate significant differences compared with Col-0. (***P* value < 0.01, **P* value
70 < 0.05, two-tailed Student's *t* test).

71

72

Figure 5. *che1* mutants develop DNA damage response

73

74

75 Figure 5. *che1* mutants develop DNA damage response

76 (A) Confocal images of PI-stained root tips of Col-0 and *che1* under normal growth conditions, treated
77 with 2 mM HU treatment, 10 μ M MMC and 20 μ M zeocin. Seedlings (4 DAG) were transferred to 1/2
78 MS medium with or without DNA damaging agents. Scale bars = 50 μ m; arrowheads point to QC
79 position. (B) Representative comet assay image of nuclei from Col-0 and *che1* (7 DAG). (C) Levels of
80 DNA fragments measured by the percentage of DNA in the tail of comets in the comet assay. The values
81 and error bars represent means and \pm SD, from three independent experiments. (D) γ H2AX accumulation
82 in the root tips of 5 DAG Col-0 on 1/2 MS medium containing 10 μ M zeocin, Col-0 and *che1* grown
83 under normal conditions measured by immunofluorescence staining. Blue, DAPI staining; green,
84 γ H2AX. Scale bar = 5 μ m. (E) Quantification of γ H2AX foci in Col-0 treated with 10 μ M zeocin, Col-
85 0 and *che1*. At least 100 nuclei were analyzed and grouped into 5 classes according to the number of
86 γ H2AX foci/nuclei. The data are from three independent experiments. Error bars indicate the SD. (F-
87 H) illustrate relative expression level (gene vs reference gene) of DNA damage response genes
88 quantified by qRT-PCR. Seedlings, 4DAG, were grown in normal growth conditions and treated for 24
89 h with the relevant genotoxic agents. 2 mm Root tips (2mm long) were collected. The values and error
90 bars in (F-H) represent means and \pm SD from three independent biological replicates. Columns with
91 different letters indicate significantly differences at $P < 0.05$ (Duncan's multiple range means
92 comparisons).
93

94

Figure 6 Cell death response in *che1* is independent on SOG1.

95

96

Figure 6 Cell death response in the *che1* mutant is enhanced in the lack of SOG1 function.

97

98

99

100

101

102

(A) Confocal images of PI-stained root tips of Col-0, *che1*, *sog1-1* and *sog1-1;che1* double mutants. Scale bars = 50 μm ; arrowheads point to QC. (B) Measurement of dead cell area and (C) percentage of roots showing dead cells. The values and error bars in (B) and (C) represent means and \pm SD of three independent experiments with at least 6 seedlings each. Columns with different letters indicate significant differences at $P < 0.05$ (Duncan's multiple range means comparisons).

103 **Figure 7. Loss of AtCHE1 partially rescues impaired root development of *atr-2* under HU**
104 **treatment.**
105

106

107 **Figure 7. Loss of AtCHE1 partially rescues impaired root development of *atr-2* under HU**
 108 **treatment.**

109 (A) and (B) Representative images of root meristems of 6 DAG Col-0, *che1*, *atr-2* and *atr-2;che1*.
110 Seedlings grown on 1/2 MS medium with or without 1mM HU. Scale bars = 50 μ m; arrowheads point
111 to QC. (C) Measurement of root meristem length and (D) root meristem cell number of 6 DAG Col-0,
112 *che1*, *atr-2* and *atr-2;che1*. Seedlings grown on 1/2 MS medium with or without 1 mM HU. (E) and (F)
113 Confocal images of PI-stained root tips of Col-0, *che1*, *atr-2* and *atr-2;che1* double mutants. 4 DAG
114 seedlings were transferred and grown on 1/2 MS medium with or without 2 mM HU for 24 h. Scale bars
115 = 50 μ m; arrowheads point to QC. (G) Measurement of the percentage of roots showing dead cells. (H)
116 Measurement of dead cell area from Col-0, *che1*, *atr-2* and *atr-2;che1* PI-stained roots. The values and
117 error bars in (C)-(D) and (G)-(H) represent means and \pm SD from three independent experiments with
118 at least 8 seedlings each. Columns with different letters indicate significant differences at $P < 0.05$
119 (Duncan's multiple range mean comparisons).

120

121

122 **Figure S1. *AtCHE1* genomic DNA fragment can complement the *che1* mutation.**

123

124

125 **Figure S1. *AtCHE1* genomic DNA fragment can complement the *che1* phenotype.**

126 (A) Complementation of the *che1* mutation with the *AtCHE1* genomic DNA region expressed under
127 its own promoter. Root growth phenotype of Col-0, *che1* and the complementation lines *AtCHE1_{pro}-*
128 *AtCHE1_g-AtCHE1_{3'-UTR}* (*AtCHE1* C1 and *AtCHE1* C2, 6 DAG). Scale bar = 1 cm. (B) Primary root
129 length (6 DAG, cm). (C) Root meristem of Col-0, *che1* and complementation lines. Scale bar = 50
130 μm; white arrowheads point to QC, while the green one to the first elongated cortical cell. (D) Number
131 of the amplifying cell in the cortex cell line and (E) Root meristem length. The value and error bars in
132 (B), (D) and (E) represent means and ±SD, n>20 in each. Columns with different letters indicate
133 significant differences at $P<0.05$ (Duncan's multiple range means comparisons).

134

135

136 **Figure S2. AtCHE1 is the plant homolog of AATF/Che-1.**

137

138

139 Phylogenetic analysis of AATF/Che-1 from different species. NCBI blast, Mega 5.05, align by clustal

140 W, construct/test neighbor joining tree. Bootstrap test using 1000 repetitions.

141

142 **Figure S3 The alignment of the AATF/Che-1 protein sequences**

Saccharomyces	--meksladqisdi-aikpvnkdfdi--edeenaslfqhnekngesdlsdygnsnsteetk	55
Homo	magppqplalqlleqllnprpseadpeadpeeataarvidrf-dege-----dg	46
Mus	maapqplalqlleqllnprpseadpeadpeeatararvidrf-dege-----ee	46
Arabidopsis	-----	0
Glycine	-----	0
Oryza	-----	0
Zea	-----	0
Saccharomyces	kahyleveksklraekglelndpkytgvkgsrqalyee---ase-----	96
Homo	egdflvvgisirklasadltdtkrycgkttsrkawnehweqtlpgssdeei-----	98
Mus	kdd-lavssirklapvsltdtkrysgkttsrkawnehweqalpsssdna-----	97
Arabidopsis	-mrlsialkvvmaggs-----krskrarlids---esedisqenlkaesdne	44
Glycine	-----m---glsa	5
Oryza	-----mapgt-----tlapkrkkae---aspsp-spspm---gdss	29
Zea	-----	0
Saccharomyces	-----nedee-----eeeeeeeeeeeeeeeeegeeee-----	123
Homo	--sdeegsgdedseglgleeyd--eddlgaeeqecgdhreskksrshsaktpgfsvqsi	154
Mus	--sdeggsedgdseglgleeis--edvdedlednkisde-----	132
Arabidopsis	ddqlpdgieddevdsmmeddegeseeddegdteeddegdseed-----	86
Glycine	kksrkrkgkrdsdsdeydnmeye-----eveddyd-----	34
Oryza	d---ggysdsdlhdaeesfysar---sgseddrqvsssndd-----	64
Zea	-----	0
Saccharomyces	-----eeekeed-alsfrtdsedeeveideeesdadggeteeaaqqrh	165
Homo	sdfekftkgmddlgsseeeedeesgme--gdda-edsqgese--edrag-----	199
Mus	-----ggsedgdseglgleefsedve-edlegede--edree-----	166
Arabidopsis	-----degenkededgedsedfedgndkesesgdegnddnkdaqme---elek	130
Glycine	-----ddgeeed-e-----eehgeevtddedgtgehgewkndeme---qlek	73
Oryza	-----ddseeeeqeremdeeedeeddddeemneeededegemn---elek	108
Zea	-----	0

143

144

Saccharomyces	alskliqqetkqainklsgsvqrdaskgysilqgtklfdniidriklqkaviaaanklpl	225
Homo	---drnseddgvvmtfssvkvseevkgravknqialwdqlllegrikqlkallttnglpq	256
Mus	---drnseddgvvaafssvkvseevkgravknqialwdqlllegrikqlkallttnglpq	223
Arabidopsis	evkelrsqeqd-ilknlkrkdgedavkkgavknqkalwdkileffllgkafdrsnrlpg	189
Glycine	eyrdlhhqeld-tlknkhhkdedllkgqavksqkalwykilelrfllgkpfsssnrlpg	132
Oryza	eyrtlqtnqgn-iletlkqhrdddsvkqgavknqkvlwdkalemrllgkafstsnklpk	167
Zea	-----mmnehreedalrgqavknqkaiwdktlemrllgkvfstsnklpg	45
	. : * : : * : : : : * : * * . : * : *	
Saccharomyces	tteswee-----akmddseetkrllkeneklfnlfnrlinfrifqlgdhitqn	275
Homo	pdvfplfkdkggpffssalknshkalkallrslvlgqeellfgyptdrylvdgkpnags	316
Mus	pdvfplfkdkggpffsasalknshkalkallrslvdlqeellfgyptdrhivngakpntes	283
Arabidopsis	epvkslfcsede-dvstayatdlvtsskktldsllelqealfeknpvddgqvnatase---	245
Glycine	esikslfcetde-tvrvaysdlmtssketldsilqlqealfaknpsitqalvsgsskd	191
Oryza	epirsmfcdhnq-eieqayldllnsskqtlgsmmelqeallernratkdvttdt-----	220
Zea	esirtrfcihk-qieqaydllnsskhtlssmmelqeallesngatkdanei-----	97
	* : * * . * : * :	
Saccharomyces	eevakh-----k--lskkrslkelyqetnsldselkeyrtavlnkwtkvss	320
Homo	eeisseddelveekqqr-rvpakrklemedypsfmakrfadftvyrnrtlqkwhdtkl	376
Mus	eeisseddelvgekkqr-kappkrklemedypsfmakrfadftiyrnhtlqkwhdtkl	342
Arabidopsis	--esnksd-----aedsdewqrisdlqkrmsvfrnkavdkwqrktqv	285
Glycine	levykhld-----gnldqewsqisqmhksitsfrdksinkwqrvtqv	233
Oryza	--nsse-l-----ngeddewevqklqkrirtpfnseidkwrktqv	259
Zea	---psa-s-----ngdndewevqrlqarittfrnteidkwhrkiqv	135
	. : . : * : * *	
Saccharomyces	asgnaalssnkfkainlpadvqvenqisdmsrlmkrtklrnrnitplyfkdccangrlpe	380
Homo	asgk--lg-kgfgafersiltqldhilmdkerllrrtqtkrsvyrvl-----gkpep	425
Mus	asgk--lg-kgfgafersiltqldhimmdkerllrrtqtkrsayrvl-----gkpep	391
Arabidopsis	ttgaaaik-gklhafnqnvseqvasyrmdpsrmikqmqqrstvf-----gtvpq	336
Glycine	ttgaaaik-gklhafnqdisnqvaayrmdpsrlikqmrvrnsdvnif-----lsvpe	284
Oryza	ttgaaaik-gklhafnqnsdqvtsyrmrpsrminrmhkrkstlgvf-----gee--	308
Zea	ttgaaaik-gklhafnqnsdqvagymrpsrminrmyltnsavrvf-----gkd--	184
	::* : : * : * : * : * * : * : *	
Saccharomyces	lispvvkdsvddnensddgldipknydprrrkdnaiditenpyvfdededfyrvllndlid	440
Homo	aaqpvpeislpgepeilpq-----apanahlkdldeeffdddfyhqllrelie	473
Mus	vpepvaetlpgepetlpq-----gpanahlrdldeeffdddfyhqllrelie	439
Arabidopsis	e-----amepn-----peekqegdpelvedaeifyqllkefle	370
Glycine	v-----vgepk-----eaetctdgdpeliddsefyqllkefle	318
Oryza	-----vgehen-----nkeenntegdpelvdsefyqllkefle	343
Zea	-----vgepgt-----aeeghivegdpeliddsefyqllkefle	219
	: : * * * : * : * : *	
Saccharomyces	kkisnahnsesaait-itstnarsnnkkidtkaskgrklnsvqdpianyeapitsg	499
Homo	rktssldpndqvamgrqwlaiqlrskihkvdrraskgrklrfhvlskllsfmapidht	533
Mus	rktssldpndqvamgrqwlaiqlrskirkkvdrraskgrklrfhvlskllsfmapidht	499
Arabidopsis	tid-----passeaafvemkfgtkkrkvdraskrkiyrnvhekivnfmaprpak	423
Glycine	tvd-----pssekafoyalkrmqpkkrkivdrraskrkiyrnvhekivnfmaplpn	371
Oryza	scd-----agasesafyalkkqghkkrklvdrraskrkiyrnvhekianfmapvpmv	396
Zea	scd-----rgasesafyalkkqvkkkrklvdrraskrkiyrnvhekivnfmapvpmv	272
	. * * * * *	
Saccharomyces	ykwsddqideffagllgqrvnfnenedeeqhariendeeleavknddiqifg	551
Homo	tm-nddartelyrsifgqlhppdeghgd-----	560
Mus	am-sddartelfrsifgqlhppdadrgk-----	526
Arabidopsis	ip---pntadilknlfglktvrvqsea-----	447
Glycine	vp---pmapklfenlfglktqrssaas-----	396
Oryza	ip---pmapklfenlfgmgngkstta-----	419
Zea	lp---pmapklfenlfgnss-----	289
	: : * : *	

146

147 Figure S3. The alignment of the AATF/Che-1 protein sequences

148 Alignment of AATF/Che-1 via clustalw2 (<http://www.ebi.ac.uk/Tools/msa/clustalw2/>). Black line
149 indicates AATF domain and green line indicates TRAUB domain of Arabidopsis.

150 **Figure S4. Expression levels of *PLT1*, *PLT2*, *SHR* and *SCR* are reduced in *che1* compared to**
151 **Col-0.**

152

153

154

155 **Figure S4. Expression levels of *PLT1*, *PLT2*, *SHR* and *SCR* are reduced in *che1* compared to**
156 **Col-0.**

157 Mean gray value of *PLT1_{pro}:CFP* (A), *PLT2_{pro}:CFP* (B) *SHR_{pro}:SHR-GFP* (C) and *SCR_{pro}:GFP* (D) in
158 Col-0 and *che1*. The values and error bars in (A) - (D) represent means and \pm SD, n>10. Asterisks
159 indicate significant differences compared to Col-0. (**P value < 0.01, two-tailed Student's t test,
160 n>10).

161

162 **Figure S5. Response to DNA damage treats in *che1* compared to Col-0.**

163

164

165 **Figure S5. Response to DNA damage treats in *che1* compared to Col-0..**

166 (A) The proportion of roots with cell death and (B) the mean area of dead cells after treatment
167 compared to the control. Data in (A) and (B) are the means and \pm SD from at least three independent
168 experiments with at least 30 seedlings in total. Columns with different letters indicate significant
169 differences, $P < 0.05$ (Duncan's multiple range means comparisons).

170

171

172

173

174

175

176

177

178

179

180 **Figure S6. Treatment with genotoxic toxic agents affect *AtCHE1* fusion protein level**

181

182

183 **Figure S6. *AtCHE1* fusion protein level is effected by DNA damage reagent treatment.**

184 (A) GUS-stained images of *AtCHE1* promoter fused to *GUS* reporter gene (*AtCHE1_{pro}:GUS*)
185 seedlings. 5 DAG seedlings were transferred and grown for 1 day on 1/2 MS medium with either no
186 stress, 2 mM HU, 10 μM MMC, or 20 μM zeocin respectively. Scale bar = 50 μm. (B) Relative
187 mRNA levels (gene vs reference gene) of *AtCHE1* under DNA damage stress treatments were
188 measured by qRT-PCR. 4 DAG seedlings were transferred to 1/2 MS liquid medium without stress,
189 2 mM HU, 10 μM MMC, or 20 μM zeocin and grown for 24 h. After that, 5 DAG seedlings were
190 collected as samples. The values and error bars represent mean and ±SD from three independent
191 experiments. Columns with different letters are significantly difference at $P < 0.05$ (Duncan's
192 multiple range means comparisons). (C) Confocal images of PI-stained of *AtCHE1_{pro}:AtCHE1_g-GFP*

193 root tips. 5 DAG seedlings were transferred to 1/2 MS plates with no stress, 2 mM HU, 10 μ M
194 MMC or 20 μ M zeocin and grown for 24 h. Magenta, PI staining and green, GFP. Scale bar = 50
195 μ m; arrowheads point to QC. **(D)** Gray value of *AtCHE1_{pro}:AtCHE1_g-GFP* under DNA damage
196 treatment. The values and error bars represent mean and \pm SD, n>8. Columns with different letters
197 are significantly difference at $P<0.05$ (Duncan's multiple range means comparisons).

198

199

200 **Figure S7. Primary root length of the *atr-2* mutant is partly rescued by *che1* in the double**
 201 **mutant upon HU treatment.**

202

203

204 **Figure S7. Primary root length of *atr-2;che1* double mutant is partially rescued compared to**
 205 ***atr-2* mutant upon HU treatment.**

206 (A) to (D) Primary root phenotype of 6 DAG Col-0 (A), *che1* (B), *atr-2* (C) and *atr-2;che1* (D)
 207 seedlings grown on 1/2 MS medium containing 0 (control condition) or 1 mM HU. Scale bars = 1
 208 cm. (E) Measurement of primary root length of Col-0, *che1*, *atr-2* and *atr-2;che1* (6 DAG). The
 209 value and error bars represent the means and \pm SD from three independent experiments with at least
 210 10 seedlings each. Columns with different letters are significantly differences at $P < 0.05$ (Duncan's
 211 multiple range means comparisons).